# On the Tension Between Optimality and Adversarial Robustness in Policy Optimization

**Haoran Li** [*]
School of Mathematical Sciences
University of Chinese Academy of Sciences
Beijing, China
lihaoran21@mails.ucas.ac.cn

**Jiayu Lv** [*]
School of Advanced Interdisciplinary Sciences
University of Chinese Academy of Sciences
Beijing, China
lvjiayu24@mails.ucas.ac.cn

**Congying Han**
School of Mathematical Sciences
University of Chinese Academy of Sciences
Beijing, China
hancy@ucas.ac.cn

**Zicheng Zhang**
JD.com
Beijing, China
zhangzicheng6@jd.com

**Anqi Li** [†]
School of Mathematical Sciences
Nankai University
Tianjin, China
anqili@nankai.edu.cn

**Yan Liu**
School of Statistics and Data Science
Nankai University
Tianjin, China
liuyan23@nankai.edu.cn

**Tiande Guo**
School of Mathematical Sciences
University of Chinese Academy of Sciences
Beijing, China
tdguo@ucas.ac.cn

**Nan Jiang**
Siebel School of Computing and Data Science
University of Illinois Urbana-Champaign
Urbana, IL 61801, USA
nanjiang@illinois.edu

## Abstract

Achieving optimality and adversarial robustness in deep reinforcement learning has long been regarded as conflicting goals. Nonetheless, recent theoretical insights presented in CAR (Li et al., 2024) suggest a potential alignment, raising the important question of how to realize this in practice. This paper first identifies a key gap between theory and practice by comparing standard policy optimization (SPO) and adversarially robust policy optimization (ARPO). Although they share theoretical consistency, *a fundamental tension between robustness and optimality arises in practical policy gradient methods*. SPO tends toward convergence to vulnerable first-order stationary policies (FOSPs) with strong natural performance, whereas ARPO typically favors more robust FOSPs at the expense of reduced returns. Furthermore, we attribute this tradeoff to the *reshaping effect of the strongest adversaries* in ARPO, which significantly complicates the global landscape by inducing *deceptive sticky FOSPs*. This improves robustness but makes navigation more challenging. To alleviate this, we develop the *BARPO*, a bilevel framework unifying SPO and ARPO by modulating adversary strength, thereby facilitating navigability while preserving global optima. Extensive empirical results demonstrate that BARPO consistently outperforms vanilla ARPO, providing a practical approach to reconcile theoretical and empirical performance. The code is available at https://github.com/RyanHaoranLi/BARPO.

---

[*]Equal contribution
[†]Corresponding author

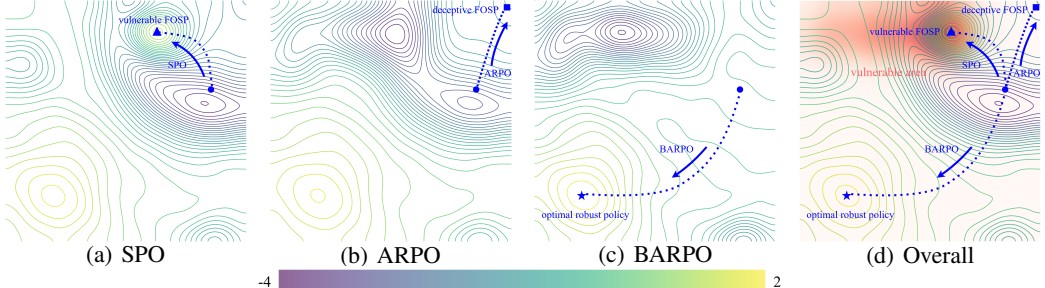

Figure 1: Schematic illustration of the optimization landscapes under SPO, ARPO, and BARPO. (a) SPO ascents along fragile directions, leading to vulnerable FOSPs with high natural value. (b) ARPO becomes trapped in robust regions but is limited to low-return solutions. (c) BARPO reshapes the landscape by lifting robust but low-return regions, enabling convergence to robust FOSPs with high returns. (d) Overall comparison of the three paradigms: contour lines represent natural returns, while background color indicates robustness, with darker red denoting lower robustness.

# 1 INTRODUCTION

Deep reinforcement learning (DRL) has demonstrated remarkable success across a variety of complex tasks (Mnih et al., 2015; Lillicrap et al., 2016; Silver et al., 2016) and real-world applications, including robotics (Ibarz et al., 2021; Liu et al., 2022; Tang et al., 2025), autonomous driving (Kiran et al., 2021; Chen et al., 2022; 2024), and large model training (Xin et al., 2025; Kumar et al., 2025; Guo et al., 2025). Despite such substantial progress, DRL agents remain highly vulnerable to imperceptible attacks on their state observations, leading to severe performance degradation and even complete system failure (Huang et al., 2017; Behzadan & Munir, 2017a; Lin et al., 2017; Kos & Song, 2017; Pattanaik et al., 2018; Weng et al., 2019; Inkawhich et al., 2020; Ilahi et al., 2021). This vulnerability threatens the reliable deployment of DRL in safety-critical environments, highlighting the urgent need for robust agents capable of withstanding subtle perturbations and malicious attacks.

Motivated by these challenges, Zhang et al. (2020c) introduced the state-adversarial Markov decision process (SA-MDP) framework, providing a theoretical foundation for adversarial robustness in reinforcement learning. Building on this framework, they demonstrated that an optimal robust policy (ORP) may not always exist, indicating a potential conflict between the objectives of optimality and robustness. Recently, Li et al. (2024; 2025) further explored the concept of ORP in practical settings and proposed the ISA-MDP formulation, a refinement that formally establishes the existence of an ORP that coincides with the Bellman optimality policy. These theoretical results suggest that the objectives of adversarial robustness and optimality are, in principle, aligned in most practical tasks.

This insight drives the pursuit of DRL agents that not only maintain strong performance in clean environments but also exhibit resilience to adversarial attacks. This theoretical consistency is critical for deploying DRL agents in real-world applications, where strong malicious attacks are comparatively rare. However, in practice, achieving this alignment remains challenging. Despite sharing this theoretical consistency, conventional policy optimization methods struggle under adversarial settings (Zhang et al., 2021; Sun et al., 2022; 2024). It remains an open problem whether this theoretical alignment can be technically realized, leaving a critical gap between theory and practice.

In this paper, we delve into the optimization behavior and learning dynamics of two training paradigms: *standard policy optimization (SPO)* and *adversarially robust policy optimization (ARPO)*. SPO aims to maximize the standard value function, i.e., $\max_\pi V^\pi(s), \ \forall \ s$, while ARPO optimizes the worst-case adversarial value function, i.e., $\max_\pi \min_\nu V^{\pi \circ \nu}(s), \ \forall \ s$. By conducting a systematic comparison, we reveal a novel optimality-robustness tension in policy gradient methods.

In our analysis, we first prove that both SPO and ARPO converge to first-order stationary policies (FOSPs) rather than global optima. Next, by examining the learning behaviors around these FOSPs, we uncover a key discrepancy: ARPO consistently produces policies with markedly higher robustness to state perturbations, whereas SPO-derived FOSPs remain vulnerable. These findings help explain the empirical vulnerability of SPO, despite the theoretical existence of an ORP, and emphasize the importance of ARPO in building robust DRL agents. Finally, we reveal that this robustness

gain comes at a significant cost: ARPO-trained policies exhibit substantially lower natural returns compared to SPO. Altogether, these results expose a fundamental robustness-performance tradeoff in policy-gradient methods, highlighting a practical challenge for closing the theory-practice gap.

To understand this tradeoff, we analyze the optimization landscape and value geometry induced by SPO and ARPO. We attribute the tension to the *reshaping effect* induced by the strongest adversaries: although they help guide the agent toward robust FOSPs, they often impede optimization by compromising the navigability of the landscape. As illustrated in Figure 1, ARPO creates robust peaks but introduces additional valleys, distorting the global landscape to a more rugged terrain. These new valleys act as traps for the optimization path, hindering gradient-based methods from ascending towards the global or better local peaks. Instead, the optimization process stalls at these *deceptive sticky FOSPs*, which often result in lower natural returns compared to those achieved by SPO. This reshaping effect presents a core barrier to learning both optimality and robustness.

To address this challenge, we propose a bilevel optimization framework that bridges SPO and ARPO, which adjusts the adversary strength to promote traversable optimization paths while maintaining robustness. This modulation prevents the optimization landscape from becoming overly rugged, helping to avoid local traps while still encouraging robustness. Importantly, it preserves the globally optimal robust policy shared by both SPO and ARPO. To enhance convergence and efficiency, we incorporate SPO dynamics into the bilevel framework, further smoothing the optimization landscape. The resulting approach, *Bilevel ARPO (BARPO)*, consistently achieves superior natural and robust returns, offering a promising approach to bridging the gap between theoretical alignment and practical performance. To summarize, this paper makes the following cohesive contributions:

- We explain the nature of vulnerability in standard policy optimization (SPO) within the theoretically aligned framework. While an optimal robust policy exists, SPO typically converges to fragile first-order stationary policies (FOSPs), which offer strong natural performance but lack robustness.
- We uncover a new form of optimality-robustness tension arising in policy optimization. Unlike prior notions of conflicting objectives, adversarially robust policy optimization (ARPO) shares the same global optima with SPO but prefers more robust FOSPs that typically yield lower returns.
- We elucidate the underlying mechanism behind this trade-off. The strongest adversaries in ARPO reshape the optimization landscape by introducing numerous sticky FOSPs, which improve robustness but hinder effective policy improvement by making the landscape more difficult to navigate.
- We propose a unified bilevel framework that connects SPO and ARPO, and instantiate it as Bilevel ARPO (BARPO), a novel method that reshapes the landscape to reduce the prevalence of poor sticky FOSPs while retaining access to robust global optima. Empirical results demonstrate that BARPO significantly improves practical performance, narrowing the gap between theory and practice.

## 2 PRELIMINARIES AND NOTATIONS

**MDP and SPO.** A *Markov Decision Process (MDP)* is formulated as a tuple $(\mathcal{S}, \mathcal{A}, r, \mathbb{P}, \gamma, \mu_0)$, where $\mathcal{S}$ is the state space. $\mathcal{A}$ is the action space. The reward function $r : \mathcal{S} \times \mathcal{A} \to \mathbb{R}$ assigns a scalar reward to each state-action pair. The transition dynamics $\mathbb{P} : \mathcal{S} \times \mathcal{A} \to \Delta(\mathcal{S})$ define the probability distribution over next states, and $\mu_0 \in \Delta(\mathcal{S})$ denotes the initial state distribution. The discount factor $\gamma \in [0, 1)$ controls the trade-off between immediate and future rewards. A stationary policy $\pi : \mathcal{S} \to \Delta(\mathcal{A})$ maps each state to a probability space over actions. Under a given policy $\pi$, the state value function is $V^\pi(s) = \mathbb{E}_{\pi, \mathbb{P}}[\sum_{t=0}^\infty \gamma^t r(s_t, a_t)|s_0 = s]$, and the action-value function (or $Q$-function) is $Q^\pi(s, a) = \mathbb{E}_{\pi, \mathbb{P}}[\sum_{t=0}^\infty \gamma^t r(s_t, a_t)|s_0 = s, a_0 = a]$. A fundamental property of MDPs is the existence of a stationary and deterministic optimal policy $\pi^*$, termed *Bellman optimality policy (BOP)*, that maximizes all $V^\pi(s)$ and $Q^\pi(s, a)$ for all states and actions. In practice, particularly for large-scale state and action spaces, *standard policy optimization (SPO)* methods are widely used for their efficiency (Silver et al., 2014; Schulman et al., 2016; 2015; Lillicrap et al., 2016; Mnih et al., 2016; Schulman et al., 2017; Haarnoja et al., 2018). For the parameterized $\pi_\theta$, SPO aims to solve:

$$\max_{\theta : \pi_\theta \in \Pi} V^{\pi_\theta}(s), \ \forall s \in \mathcal{S}, \text{ where } \Pi := \{\pi_\theta | \pi_\theta(\cdot|s) \in \Delta(\mathcal{A}), \ \forall s \in \mathcal{S}\}. \quad \text{(SPO)}$$

To simplify notations, we take the expectation over the distribution $\mu_0$: $V^{\pi_\theta}(\mu_0) = \mathbb{E}_{s \sim \mu_0}[V^{\pi_\theta}(s)]$.

**ISA-MDP and ARPO.** An *Intrinsic State-adversarial Markov Decision Process (ISA-MDP)* is defined by the tuple $(\mathcal{S}, \mathcal{A}, r, \mathbb{P}, \gamma, \mu_0, B^*)$. This formulation extends the standard MDP by incor-

porating structured neighborhoods of allowable state perturbations. For each state $s \in \mathcal{S}$, the set-value function $B^*(s) \subseteq \mathcal{S}$ specifies admissible perturbations which cannot alter the optimal action. An adversary is represented by a mapping $\nu : \mathcal{S} \to \mathcal{S}$ that perturbs the observed state $s$ to a nearby state $s_\nu := \nu(s) \in B^*(s)$. Under this adversary, the policy operates on the perturbed state, denoted by $(\pi \circ \nu)(a|s) := \pi(a|\nu(s))$. Under this setting, the adversarial value function is given by $V^{\pi \circ \nu}(s) = \mathbb{E}_{\pi \circ \nu, \mathbb{P}}[\sum_{t=0}^{\infty} \gamma^t r(s_t, a_t)|s_0 = s]$, and the corresponding adversarial $Q$-function is $Q^{\pi \circ \nu}(s, a) = \mathbb{E}_{\pi \circ \nu, \mathbb{P}}[\sum_{t=0}^{\infty} \gamma^t r(s_t, a_t)|s_0 = s, a_0 = a]$. For any policy $\pi$, there exists the strongest adversary $\nu^*(\pi)$ that minimizes the value function across all states, defined by $\nu^*(\pi) = \arg\min_\nu V^{\pi \circ \nu}(s), \forall s$. Within this framework, an *optimal robust policy (ORP)* $\pi^*$ exists, which simultaneously maximizes performance against its strongest adversary for all states, i.e., $V^{\pi^* \circ \nu^*(\pi^*)}(s) = \max_\pi V^{\pi \circ \nu^*(\pi)}(s)$ for all $s \in \mathcal{S}$ and $\pi^*$ aligns with Bellman optimality policy.

We refer to the robust optimization problem in ISA-MDP as *Adversarial Robust Policy Optimization (ARPO)*. Despite its centrality to robust decision-making, ARPO has not been thoroughly explored. In this work, we adopt the direct parameterized adversary, resulting in the following formulation:

$$\max_{\theta:\pi_\theta \in \Pi} \min_{\vartheta:\nu_\vartheta \in \Psi} V^{\pi_\theta \circ \nu_\vartheta}(s), \; \forall s \in \mathcal{S}, \text{ where } \Psi := \{\nu_\vartheta|\nu_\vartheta(s) = s + \vartheta_s \in B^*(s), \; \forall s\}. \quad \text{(ARPO)}$$

Similar to SPO, we consider the expected objective: $V^{\pi_\theta \circ \nu_\vartheta}(\mu_0) = \mathbb{E}_{s \sim \mu_0}[V^{\pi_\theta \circ \nu_\vartheta}(s)]$. In particular, SPO corresponds to solving the outer maximization while fixing the inner solution to $\vartheta \equiv 0$.

**FOSP** refers to a stationary point in the optimization sense, where the policy gradient vanishes. Formally, we define $\pi_\theta$ (or $\theta$) as a *first-order stationary policy (FOSP)* for SPO if it satisfies:

$$\nabla_\theta V^{\pi_\theta}(\mu_0) = 0, \quad \nabla_{\theta\theta}^2 V^{\pi_\theta}(\mu_0) \preceq 0. \quad \text{(FOSP in SPO)}$$

Similarly, $\pi_\theta$ (or $\theta$) is a FOSP for ARPO if it satisfies the following conditions:

$$\nabla_\theta V^{\pi_\theta \circ \nu^*(\pi_\theta)}(\mu_0) = 0, \quad \nabla_{\theta\theta}^2 V^{\pi_\theta \circ \nu^*(\pi_\theta)}(\mu_0) \preceq 0. \quad \text{(FOSP in ARPO)}$$

We further define $(\pi_\theta, \nu^*(\pi_\theta))$ (or $(\theta, \vartheta^*(\theta))$) as a first-order stationary policy-adversary for ARPO.

**Terminology Clarification.** To avoid ambiguity, we define the following terms used throughout the paper: 1) *Natural returns* (or *performance*): the estimate of standard value function $V^\pi(\mu_0)$. 2) *Robust returns* (against $\nu$): the approximation of adversarial value function $V^{\pi \circ \nu}(\mu_0)$. 3) *Robustness* (against $\nu$): an estimation of the relative degradation in performance $\frac{V^{\pi \circ \nu}(\mu_0) - V^\pi(\mu_0)}{V^\pi(\mu_0)} \leq 0$.

## 3 OPTIMALITY-ROBUSTNESS TENSION IN POLICY OPTIMIZATION

This section identifies the inherent tradeoff between optimality and robustness in policy gradient methods. Although SPO and ARPO share the consistent globally optimal robust policy, their convergence behaviors diverge significantly. In practice, SPO often tends to vulnerable policies with high natural performance, while ARPO prefers more robust policies, typically at the cost of reduced returns. Notably, we explore the value geometry and optimization landscape, attributing this divergence to the adversarial reshaping of ARPO. The strongest adversary distorts the global terrain by introducing numerous local minima, which trap the navigation in robust but suboptimal policies.

### 3.1 ROBUST CONVERGENCE BEHAVIOR OF ARPO

We begin with a rigorous convergence analysis of ARPO, showing that it generally converges to a first-order stationary policy (FOSP) rather than the global optimum. To better understand this, we examine the nature of FOSPs in both ARPO and SPO by analyzing empirical behaviors and local learning dynamics. Our findings reveal that ARPO promotes stronger robustness near convergence, even without reaching the globally optimal policy. In contrast, SPO achieves high natural returns but lacks robustness against perturbations. Furthermore, this analysis uncovers a connection between robustness and generalization, suggesting that sufficient robustness may improve generalization.

#### 3.1.1 CONVERGENCE TO FIRST-ORDER STATIONARY POLICIES

To enable a formal gradient-based analysis of ARPO, we first derive the policy gradient expression for the adversary. This formulation facilitates sample-based estimation and numerical computation within the inner optimization. The complete derivation and proof are provided in Appendix C.1.

Table 1: Natural and robust performance of SPO and ARPO for continuous control tasks in MuJoCo.

| Environment | Method | Natural Return | Return under Attack | | | | | | | Worst-case Robustness |
|---|---|---|---|---|---|---|---|---|---|---|
| | | | Random | Critic | MAD | RS | SA-RL | PA-AD | Worst | |
| **Hopper** | SPO | $3081 \pm 638$ | $2923 \pm 767$ | $2035 \pm 1035$ | $1763 \pm 619$ | $756 \pm 36$ | $79 \pm 2$ | $823 \pm 182$ | $79 \pm 2$ | -0.974 |
| ($\epsilon = 0.075$) | ARPO | $2101 \pm 588 \downarrow 32\%$ | $2058 \pm 559$ | $3156 \pm 528$ | $2225 \pm 641$ | $1001 \pm 30$ | $1032 \pm 47$ | $1799 \pm 547$ | $1001 \pm 30$ | -0.524 $\uparrow 46\%$ |
| **Walker2d** | SPO | $4662 \pm 22$ | $4628 \pm 21$ | $4584 \pm 15$ | $4507 \pm 675$ | $1062 \pm 150$ | $719 \pm 1079$ | $336 \pm 252$ | $336 \pm 252$ | -0.928 |
| ($\epsilon = 0.05$) | ARPO | $2095 \pm 983 \downarrow 56\%$ | $1815 \pm 930$ | $2360 \pm 1021$ | $1944 \pm 976$ | $1360 \pm 835$ | $1270 \pm 502$ | $1460 \pm 739$ | $1270 \pm 502$ | -0.394 $\uparrow 58\%$ |
| **Halfcheetah** | SPO | $5048 \pm 526$ | $4463 \pm 650$ | $3281 \pm 1101$ | $918 \pm 541$ | $1049 \pm 50$ | $-213 \pm 103$ | $-69 \pm 22$ | $-213 \pm 103$ | -1.042 |
| ($\epsilon = 0.15$) | ARPO | $1412 \pm 99 \downarrow 72\%$ | $1363 \pm 285$ | $1359 \pm 59$ | $1402 \pm 64$ | $1230 \pm 75$ | $1079 \pm 48$ | $1216 \pm 458$ | $1079 \pm 48$ | -0.236 $\uparrow 77\%$ |
| **Ant** | SPO | $5381 \pm 1308$ | $5329 \pm 976$ | $4696 \pm 1015$ | $1768 \pm 929$ | $1097 \pm 633$ | $-1398 \pm 318$ | $-3107 \pm 1071$ | $-3107 \pm 1071$ | -1.577 |
| ($\epsilon = 0.15$) | ARPO | $1709 \pm 564 \downarrow 68\%$ | $2026 \pm 38$ | $1976 \pm 131$ | $1839 \pm 350$ | $1661 \pm 593$ | $1648 \pm 666$ | $1675 \pm 573$ | $1648 \pm 666$ | -0.034 $\uparrow 98\%$ |

**Theorem 3.1** (Policy Gradient for Adversary). *Given a policy $\pi_\theta$, for all state $s \in \mathcal{S}$, consider the direct parameterization representation for adversary $\nu_\vartheta : \mathcal{S} \to \mathcal{S}$, $s \mapsto s + \vartheta_s \in B(s)$. Then, for any state $s_i \in \mathcal{S}$, we have the state-wise policy gradient for the adversary as follows:*

$$\nabla_{\vartheta_{s_i}} V^{\pi_\theta \circ \nu_\vartheta}(s) = \frac{1}{1-\gamma} \mathbb{E}_{(s',a') \sim d^{\pi_\theta \circ \nu_\vartheta}} \left[ Q^{\pi_\theta \circ \nu_\vartheta}(s',a') \nabla_{\vartheta_{s'}} \log \pi_\theta(a'|s' + \vartheta_{s'}) \cdot \mathbb{I}(s' = s_i) \right],$$

*where $d^{\pi \circ \nu}$ is the state-action visitation distribution under $\pi \circ \nu$, and $\mathbb{I}(\cdot)$ is the indicator function.*

Based on this analytical expression, we further characterize the convergence behavior of ARPO. The detailed analysis and proofs are provided in Appendix C.2. Justifications and potential relaxations of assumptions, and the connection with core contributions are further discussed in Appendix H.1.

**Theorem 3.2** (Convergence of ARPO). *Denote $\Delta := \max_\theta \min_\vartheta V^{\pi_\theta \circ \nu_\vartheta}(\mu_0) - \min_\vartheta V^{\pi_{\theta_0} \circ \nu_\vartheta}(\mu_0)$. Assume that the sampled policy gradient is Lipschitz continuous and bounded by $M_{\hat{v}}$, the sampled estimation of the value function is locally $\mu$-strongly convex, the state-action visitation distribution is Lipschitz continuous, and the variance of the stochastic gradient is bounded by $\sigma^2$. Set the step size as $\eta_k = \sqrt{\Delta / (\sigma^2 L K)}$. Then, the ARPO with $\delta$-approximate adversary and $K \geq \frac{\Delta L}{\sigma^2}$ satisfies:*

$$\frac{1}{K} \sum_{k=0}^{K-1} \mathbb{E} \left[ \left\| \nabla_\theta V^{\pi_{\theta_k} \circ \nu^*(\pi_{\theta_k})}(\mu_0) \right\|_2^2 \right] \leq 4\sigma \sqrt{\frac{\Delta L}{K}} + \frac{2L_{\theta\vartheta}^2 \delta}{\mu},$$

*where $L_{\theta\vartheta}$ and $L = \frac{L_{\theta\vartheta} L_{\theta\theta}}{\mu} + L_{\theta\theta} + M_{\hat{v}} \left( \frac{L_{d\vartheta} L_{\vartheta\theta}}{\mu} + L_{d\theta} \right)$ are the Lipschitz constants.*

Theorem 3.2 shows that, instead of converging to the globally optimal robust policy, ARPO approximates an FOSP of the value function under the strongest adversary. The convergence rate is $O(K^{-1/2})$, and the approximation error depends on the strength of the adversary. Similarly, SPO achieves convergence to the FOSP of the standard value function at the same rate (Agarwal et al., 2019; 2021). These underscore the gap between theoretical optimum and practical convergence.

### 3.1.2 LEARNING BEHAVIORS NEAR FIRST-ORDER STATIONARY POLICIES

Based on the above convergence results, we further explore the characteristics of SPO and ARPO around their FOSPs, highlighting the critical difference between robustness and natural performance.

**Empirical Behaviors Near FOSPs.** We empirically examine the properties of FOSPs obtained by ARPO across both simple and complex environments. Specifically, we analyze a toy ISA-MDP with two states and two actions using directly parameterized policies, alongside continuous control tasks in MuJoCo employing neural network policies. In both settings, we observe that *FOSPs discovered by ARPO tend to exhibit robustness but generally achieve substantially reduced returns than those achieved by SPO.* The detailed analysis and complete proofs are provided in Appendix D.

**Proposition 3.1.** *There exists an ISA-MDP such that the following statement holds. Let $\pi_S$ be a FOSP under SPO, and let $\pi_A$ be a FOSP under ARPO. Then, $\pi_A$ is a robust policy with $V^{\pi_A}(\mu_0) - V^-(\mu_0) < \frac{1}{2}(V^{\pi_S}(\mu_0) - V^-(\mu_0))$, where $V^-(\mu_0) = \min_\pi V^\pi(\mu_0)$ denotes the worst-case value.*

Furthermore, as shown in Table 1, in complex high-dimensional tasks, SPO achieves strong natural performance but suffers substantial degradation under adversarial attacks. In contrast, ARPO exhibits substantially greater robustness, particularly in more challenging environments such as

HalfCheetah and Ant. However, its natural performance remains consistently and noticeably lower, often reaching less than half the returns achieved by SPO. These empirical results underscore the practical disparity between the FOSPs obtained by SPO and ARPO, revealing an inherent tradeoff between robustness and optimality in policy gradient methods.

**Learning Dynamics Near FOSPs.** To better understand the practical performance of SPO and ARPO, we analyze their learning dynamics around their FOSPs. This sheds light on how these paradigms behave near convergence and reveals critical insights into how robustness impacts generalization. Specifically, we characterize the Hessian structures at the FOSPs for both SPO and ARPO, which reflect the local curvature of the value landscape and provide a measure of policy flatness.

**Theorem 3.3** (Flatness Bound of FOSPs). *Let $(\theta^*, \vartheta^*)$ be a first-order stationary policy-adversary in ARPO, satisfying the second-order optimality condition. Denote $\lambda_{\min}(\cdot)$ and $\lambda_{\max}(\cdot)$ as the smallest and largest eigenvalues, and let $d$ be the state space dimension. Suppose $(\theta^*, \vartheta^*)$ is locally stable, the initialization $(\theta_0, \vartheta_0)$ satisfies $V^{\pi_{\theta_0} \circ \nu_{\vartheta_0}}(\mu_0) < V^{\pi_{\theta^*} \circ \nu_{\vartheta^*}}(\mu_0)$, the policy gradient noise is coercive with coefficient $\kappa$. Then, for single-loop ARPO with step size $\eta \geq 2/\lambda_{\min}(-\nabla_{\theta\theta}^2 V^{\pi_{\theta^*} \circ \nu_{\vartheta^*}}(\mu_0))$, attack budget $\epsilon \leq 2/\lambda_{\max}(\nabla_{\vartheta\vartheta}^2 V^{\pi_{\theta^*} \circ \nu_{\vartheta^*}}(\mu_0))$, and batch size $B \geq d\epsilon^2 \kappa_\vartheta \lambda_{\max}(\nabla_{\vartheta\vartheta}^2 V^{\pi_{\theta^*} \circ \nu_{\vartheta^*}}(\mu_0))^2$, the following inequality between Hessians holds:*

$$\left( \|\nabla_{\theta\theta}^2 V^{\pi_{\theta^*} \circ \nu_{\vartheta^*}}(\mu_0)\|_F^2 + \frac{B}{\kappa_\theta \eta^2} \right) \left( 1 - \frac{\epsilon^2 \kappa_\vartheta}{B} \|\nabla_{\vartheta\vartheta}^2 V^{\pi_{\theta^*} \circ \nu_{\vartheta^*}}(\mu_0)\|_F^2 \right) \leq \frac{B}{\kappa_\theta \eta^2}. \tag{1}$$

*Similarly, for a locally stable FOSP $\theta^*$ in SPO, if the initialization $\theta_0$ satisfies $V^{\pi_{\theta_0}}(\mu_0) < V^{\pi_{\theta^*}}(\mu_0)$, and the policy gradient noise is coercive with $\kappa_\theta$, then $\|\nabla_{\theta\theta}^2 V^{\pi_{\theta^*}}(\mu_0)\|_F^2 \leq \frac{B}{\kappa_\theta \eta^2}$.*

The detailed derivations are in Appendix C.3. From inequality (1), we observe that for FOSP in ARPO, if the adversary-side curvature is bounded as $\|\nabla_{\vartheta\vartheta}^2 V^{\pi_{\theta^*} \circ \nu_{\vartheta^*}}(\mu_0)\|_F^2 \leq B/(2\kappa_\vartheta \epsilon^2)$, then the policy-side curvature in ARPO matches that of SPO, i.e., $\|\nabla_{\theta\theta}^2 V^{\pi_{\theta^*} \circ \nu_{\vartheta^*}}(\mu_0)\|_F^2 \leq B/(\kappa_\theta \eta^2)$. This implies that if ARPO achieves sufficiently strong robustness, i.e., flat adversarial curvature, it can preserve generalization comparable to SPO. Moreover, greater robustness may further enhance generalization. However, if robustness is insufficient, generalization may be significantly degraded.

## 3.2 RESHAPING EFFECT OF STRONGEST ADVERSARIES HINDERING POLICY IMPROVEMENT

In this subsection, we investigate how the strongest adversary in ARPO reshapes the optimization landscape and value space geometry, shedding light on a key factor behind the optimality-robustness tension. We show that although the strongest adversary promotes convergence to robust FOSPs, it also severely distorts the global landscape, thereby impeding effective policy improvement.

**Adversarial Reshaping Promotes Robust FOSPs.** As illustrated in Figure 2, we compare ARPO with SPO. It follows directly from the definition that the robust objective $V^{\pi_\theta \circ \nu^*(\pi_\theta)}(\mu_0)$ is always less than or equal to the standard objective $V^{\pi_\theta}(\mu_0)$. In blue regions where $\pi_\theta$ exhibits strong robustness, the gap $V^{\pi_\theta}(\mu_0) - V^{\pi_\theta \circ \nu^*(\pi_\theta)}(\mu_0)$ remains small, leading to only a slight shift in the location of

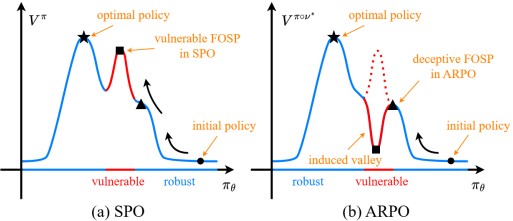

Figure 2: Intuitive examples of reshaping.

$\theta$ within the landscape relative to that under SPO. In contrast, in red regions where $\pi_\theta$ is more vulnerable, the gap becomes substantial, resulting in a sharp decline in the objective value at the corresponding location. This adversarial reshaping significantly increases the ruggedness of the optimization landscape, thereby steering the learning process toward robust policies as FOSPs.

**Distorted Landscape Impairs Policy Navigation.** However, the robust policy space typically exhibits vulnerable connectivity. Adversarial reshaping introduces deep valleys that fragment the terrain, isolate different FOSPs, and obstruct gradient-based traversal. The proof is in Appendix D.

**Proposition 3.2** (Vulnerable Connectivity Leading to Separated FOSPs). *There exists an ISA-MDP where the robust policy space $\Pi_{Robust}$ contains a cut point; that is, there exists a policy $\pi$ such that $\Pi_{Robust} \setminus \{\pi\}$ is disconnected. Therefore, ARPO tends to yield more isolated FOSPs than SPO.*

As illustrated in Figure 4(b), the valleys introduced by the adversary create barriers that obstruct ARPO from reaching better FOSPs or the globally optimal robust policy. This fragmentation increases the difficulty of learning and often traps the agent in suboptimal FOSPs. As further shown in Figure 3, even in a simple ISA-MDP, approximately one-third of initial policies converge to a deceptive FOSP with low value, highlighting the stickiness of these deceptive solutions and how the reshaped terrain can mislead the learning dynamics. These results

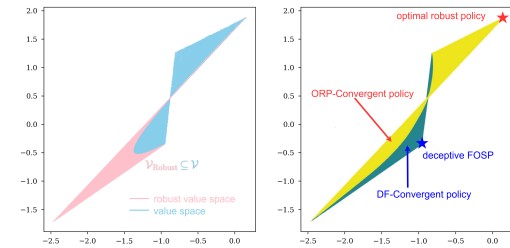

Figure 3: Adversarially robust value geometry.

reveal a core optimization challenge in adversarially robust DRL: adversarial reshaping not only fosters robustness but also introduces deceptive FOSPs that impede consistent policy improvement.

## 4 BRIDGING OPTIMALITY AND ROBUSTNESS VIA A BILEVEL FRAMEWORK

The above analysis highlights the strong influence of worst-case adversaries on the optimization landscape. In this section, we seek to mitigate the distorting effects of such adversarial reshaping while preserving strong robustness. To this end, we propose a bilevel optimization framework that interpolates between standard and adversarially robust policy optimization by modulating the strength of the adversary. Furthermore, we derive a surrogate for the strongest adversary to construct a practical algorithm that smooths the distorted landscape while maintaining robust performance.

### 4.1 UNIFYING STANDARD AND ADVERSARIALLY ROBUST POLICY OPTIMIZATION

To bridge standard and adversarially robust policy training, we introduce the following bilevel optimization framework that relaxes the inner optimization to allow non-strongest adversaries. This relaxation enables a better balance between adversarial robustness and landscape smoothness:

$$\max_{\theta} V^{\pi_\theta \circ \nu^\diamond(\theta)}(\mu_0), \quad \text{s.t. } \nu^\diamond(\theta) = \arg\max_{\vartheta} G(\pi_\theta, \nu_\vartheta). \quad \text{(General Bilevel ARPO)}$$

Here, the inner objective $G(\pi, \nu)$ defines the adversary behavior, extending robustness beyond worst-case scenarios. This formulation generalizes both SPO and ARPO. Specifically, when $\nu^\diamond(s; \theta) \equiv s$, this framework reduces to SPO; and when $G(\pi, \nu) = -V^{\pi \circ \nu}(\mu_0)$, it recovers ARPO. Importantly, within the ISA-MDP setting, this unified framework preserves the same optimal robust policy as both SPO and ARPO, preserving the theoretical alignment of optimality and robustness.

### 4.2 BILEVEL ADVERSARIALLY ROBUST POLICY OPTIMIZATION WITH KL-SURROGATE

To make the proposed bilevel framework applicable in practice, we need to specify a suitable inner objective $G$ that both promotes policy learning and maintains strong robustness. To this end, we consider adversaries practically derived via policy gradient methods, and identify the KL divergence between the original and perturbed policies as an effective surrogate for the strongest adversary.

**Theorem 4.1** (Surrogate Adversary). *For any policy $\pi_\theta$, state $s \in \mathcal{S}$, and $K > 0$, let the adversary $\vartheta$ be computed via $K$-step gradient descent on the adversarial value function, i.e., $\vartheta_{s,k} = \vartheta_{s,k-1} - \eta_{s,k-1} \nabla_{\vartheta_s} V^{\pi_\theta \circ \nu_\vartheta}(s)|_{\vartheta_s = \vartheta_{s,k-1}}, 1 \leq k \leq K$. Assume that step sizes $\eta_{s,k} \leq 1/(\lambda_s + \delta)$ for some $\delta > 0$, and $G(s; \pi, \nu) := \mathcal{D}_{\mathrm{KL}}(\pi(\cdot|s) \| (\pi \circ \nu)(\cdot|s)) \ll 1$. Then, we have the lower bound of robustness:*

$$V^{\pi_\theta}(s) - V^{\pi_\theta \circ \nu_\vartheta}(s) \geq \frac{2\delta}{\lambda_{\max}(F_{s,\theta})K} G(s; \pi_\theta, \nu_\vartheta) + O\left(G(s; \pi_\theta, \nu_\vartheta)^{\frac{3}{2}}\right),$$

*where $F_{s,\theta}$ is the Fisher information matrix of $\log(\pi_\theta \circ \nu_\vartheta)(a|s)$ with respect to $\vartheta_s = 0$, defined as $F_{s,\theta} := F(\theta, s) = \mathbb{E}_{a \sim \pi_\theta(\cdot|s)} \left[ (\nabla_{\vartheta_s} \log \pi_\theta(a|s + \vartheta_s)|_{\vartheta_s=0}) (\nabla_{\vartheta_s} \log \pi_\theta(a|s + \vartheta_s)|_{\vartheta_s=0})^T \right], \lambda_s = \max_{s + \vartheta_s \in B(s)} \lambda_{\max}(\nabla^2_{\vartheta\vartheta} V^{\pi_\theta \circ \nu_\vartheta}(s)|_{\vartheta = \vartheta_s})$ and $\lambda_{\max}(\cdot)$ is the largest eigenvalue of matrix.*

The detailed proof is provided in Appendix E. Theorem 4.1 indicates that, for gradient-based strong adversaries, the KL divergence serves as a valid first-order surrogate for minimizing the adversarial

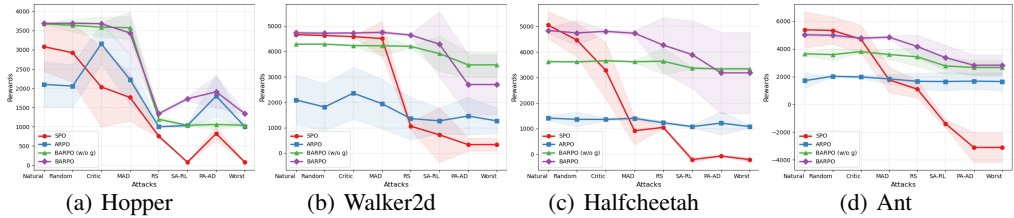

Figure 5: Natural and robust performance of SPO, ARPO, BARPO without guidance, and BARPO for four continuous control tasks in MuJoCo. BARPO (w/o g) consistently outperforms ARPO.

value. This surrogate is both appropriate and reliable from an optimization perspective. Motivated by this insight, we propose the following instantiation of our framework, termed *Bilevel ARPO*:

$$\max_{\theta} V^{\pi_\theta \circ \nu^\diamond(\theta)}(\mu_0), \quad \text{s.t. } \nu^\diamond(s;\theta) = \arg\max_{\vartheta} \mathcal{D}_{\mathrm{KL}}(\pi_\theta(\cdot|s)\|(\pi_\theta \circ \nu_\vartheta)(\cdot|s)), \ \forall \ s \in \mathcal{S}. \tag{BARPO}$$

As illustrated in Figures 1 and 4, BARPO effectively reshapes the landscape by elevating low-value but robust regions, thereby creating smoother paths toward better FOSPs and even global optima. This restructuring facilitates learning navigability while promoting updates along robust trajectories, ensuring simultaneous improvements of optimality and robustness. In contrast, SPO tends to follow vulnerable di-

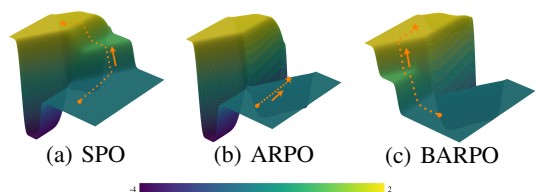

Figure 4: Illustrative Optimization Landscapes

rections, frequently ascending fragile peaks. Meanwhile, ARPO strictly follows the most robust path but often gets stuck in valleys from overly strong adversaries, hindering escape from poor FOSPs. We further delve into the theoretical understanding of BARPO in Appendices H.3 and H.4.

## 5 EXPERIMENTS

In this section, we conduct comprehensive comparison experiments and ablation studies to empirically substantiate our theoretical insights and evaluate the practical effectiveness of BARPO.

### 5.1 IMPLEMENTATION DETAILS

**Environments and Baselines.** Following baselines (Zhang et al., 2020c; Oikarinen et al., 2021; Liang et al., 2022), we perform experiments on four challenging MuJoCo benchmarks (Todorov et al., 2012) with continuous action spaces: Hopper, Walker2d, HalfCheetah, and Ant. We compare BARPO against several state-of-the-art robust training methods. SA-PPO (Zhang et al., 2020c) incorporates a KL-based regularization and solves the inner maximization problem using PGD (Madry et al., 2018) and CROWN-IBP (Zhang et al., 2020b), respectively. RADIAL-PPO (Oikarinen et al., 2021) incorporates adversarial regularization guided by robustness verification bounds derived from IBP (Gowal et al., 2019). WocaR-PPO (Liang et al., 2022) estimates worst-case values and also employs KL-based regularization. We use the official implementations of these baselines to train 17 agents per method under identical settings. Regularization coefficients are tuned appropriately for each method. To mitigate the effects of high variance in reinforcement learning training, we report the median performance across 17 runs to ensure reproducibility. More details are in Appendix F.

**Evaluations.** We evaluate the robustness of agents on MuJoCo tasks using six adversarial attack methods: (1) Random Attack: adds uniform random noise to state observations; (2) Critic Attack (Pattanaik et al., 2018): perturbs states based on the action-value function; (3) MAD Attack (Zhang et al., 2020c): maximizes the discrepancy between policies in clean and perturbed states; (4) RS Attack (Zhang et al., 2020c): first learns a robust action-value function, then conducts critic-based attacks guided by it; (5) SA-RL (Zhang et al., 2021): employs a learned adversary

Table 2: Average returns ($\pm$ std) over 50 episodes for baselines and BAR-PPO on MuJoCo tasks. Results include natural return, returns under six attacks, the worst return and robustness under these attacks. Bold and underlined values indicate the top and second performances, respectively.

| Environment | Method | Natural Return | Return under Attack | | | | | | | Worst-case Robustness |
|---|---|---|---|---|---|---|---|---|---|---|
| | | | Random | Critic | MAD | RS | SA-RL | PA-AD | Worst | |
| **Hopper** ($\epsilon = 0.075$) | PPO | 3081 $\pm$ 638 | 2923 $\pm$ 767 | 2035 $\pm$ 1035 | 1763 $\pm$ 619 | 756 $\pm$ 36 | 79 $\pm$ 2 | 823 $\pm$ 182 | 79 $\pm$ 2 | -0.974 |
| | SA | 3518 $\pm$ 272 | 2835 $\pm$ 866 | 3662 $\pm$ 6 | 3045 $\pm$ 797 | **1407** $\pm$ 36 | 1476 $\pm$ 255 | 1286 $\pm$ 282 | 1286 $\pm$ 282 | **-0.634** |
| | RADIAL | 3254 $\pm$ 714 | 3170 $\pm$ 754 | **3706** $\pm$ 11 | 2558 $\pm$ 888 | 1307 $\pm$ 420 | 993 $\pm$ 717 | 1696 $\pm$ 574 | 993 $\pm$ 717 | -0.718 |
| | WocaR | 3629 $\pm$ 35 | 3637 $\pm$ 29 | 3657 $\pm$ 30 | 3150 $\pm$ 737 | 1171 $\pm$ 1013 | 1452 $\pm$ 66 | **2124** $\pm$ 872 | 1171 $\pm$ 1013 | -0.677 |
| | BAR | **3684** $\pm$ 20 | **3692** $\pm$ 24 | 3678 $\pm$ 19 | **3437** $\pm$ 555 | 1340 $\pm$ 42 | **1728** $\pm$ 62 | 1908 $\pm$ 410 | **1340** $\pm$ 42 | -0.636 |
| **Walker2d** ($\epsilon = 0.05$) | PPO | 4662 $\pm$ 22 | 4628 $\pm$ 21 | 4584 $\pm$ 15 | 4507 $\pm$ 675 | 1062 $\pm$ 150 | 719 $\pm$ 1079 | 336 $\pm$ 252 | 336 $\pm$ 252 | -0.928 |
| | SA | **4875** $\pm$ 278 | **4907** $\pm$ 187 | **5029** $\pm$ 74 | **4833** $\pm$ 132 | 2775 $\pm$ 1308 | 3356 $\pm$ 1433 | 997 $\pm$ 1152 | 997 $\pm$ 1152 | -0.795 |
| | RADIAL | 2531 $\pm$ 1089 | 2170 $\pm$ 1102 | 2063 $\pm$ 1068 | 2316 $\pm$ 1171 | 1239 $\pm$ 800 | 426 $\pm$ 22 | 1353 $\pm$ 921 | 426 $\pm$ 22 | -0.832 |
| | WocaR | 4226 $\pm$ 938 | 4347 $\pm$ 892 | 4342 $\pm$ 875 | 4373 $\pm$ 850 | 3358 $\pm$ 1242 | 2385 $\pm$ 840 | 1064 $\pm$ 999 | 1064 $\pm$ 999 | -0.752 |
| | BAR | 4732 $\pm$ 86 | 4718 $\pm$ 59 | 4727 $\pm$ 36 | 4750 $\pm$ 65 | **4646** $\pm$ 53 | **4285** $\pm$ 1282 | **2699** $\pm$ 1192 | **2699** $\pm$ 1192 | **-0.436** |
| **Halfcheetah** ($\epsilon = 0.15$) | PPO | **5048** $\pm$ 526 | 4463 $\pm$ 650 | 3281 $\pm$ 1101 | 918 $\pm$ 541 | 1049 $\pm$ 50 | -213 $\pm$ 103 | -69 $\pm$ 22 | -213 $\pm$ 103 | -1.042 |
| | SA | 4780 $\pm$ 1456 | **4983** $\pm$ 1122 | **5035** $\pm$ 1245 | 3759 $\pm$ 1963 | 2727 $\pm$ 1707 | 1443 $\pm$ 1082 | 1551 $\pm$ 1139 | 1443 $\pm$ 1082 | -0.698 |
| | RADIAL | 4739 $\pm$ 80 | 4642 $\pm$ 75 | 4546 $\pm$ 329 | 2961 $\pm$ 1478 | 1327 $\pm$ 1112 | 1522 $\pm$ 1059 | 1968 $\pm$ 1284 | 1327 $\pm$ 1112 | -0.720 |
| | WocaR | 4723 $\pm$ 462 | 4798 $\pm$ 69 | 4846 $\pm$ 349 | 4543 $\pm$ 566 | 3302 $\pm$ 1718 | 2270 $\pm$ 1318 | 2498 $\pm$ 1224 | 2270 $\pm$ 1318 | -0.519 |
| | BAR | 4837 $\pm$ 99 | 4741 $\pm$ 501 | 4803 $\pm$ 54 | **4729** $\pm$ 105 | **4265** $\pm$ 1077 | **3894** $\pm$ 1322 | **3181** $\pm$ 1593 | **3181** $\pm$ 1593 | **-0.342** |
| **Ant** ($\epsilon = 0.15$) | PPO | **5381** $\pm$ 1308 | **5329** $\pm$ 976 | 4696 $\pm$ 1015 | 1768 $\pm$ 929 | 1097 $\pm$ 633 | -1398 $\pm$ 318 | -3107 $\pm$ 1071 | -3107 $\pm$ 1071 | -1.577 |
| | SA | 5367 $\pm$ 94 | 5217 $\pm$ 410 | **5012** $\pm$ 113 | **5114** $\pm$ 294 | **4396** $\pm$ 1225 | **4227** $\pm$ 177 | 2355 $\pm$ 1620 | 2355 $\pm$ 1620 | -0.539 |
| | RADIAL | 4358 $\pm$ 98 | 4309 $\pm$ 206 | 3628 $\pm$ 267 | 4205 $\pm$ 127 | 3742 $\pm$ 696 | 2364 $\pm$ 47 | 3261 $\pm$ 134 | 2364 $\pm$ 47 | -0.462 |
| | WocaR | 4069 $\pm$ 598 | 3911 $\pm$ 659 | 3978 $\pm$ 190 | 3689 $\pm$ 841 | 3176 $\pm$ 908 | 1868 $\pm$ 99 | 1830 $\pm$ 102 | 1830 $\pm$ 102 | -0.550 |
| | BAR | 5024 $\pm$ 117 | 4979 $\pm$ 114 | 4777 $\pm$ 122 | 4843 $\pm$ 120 | 4171 $\pm$ 826 | 3367 $\pm$ 902 | **2825** $\pm$ 757 | **2825** $\pm$ 757 | **-0.438** |

against the victim to perturb the state; (6) PA-AD (Sun et al., 2022): trains an adversarial agent to identify a vulnerable perturbation direction, followed by an FGSM attack along that direction.

**BAR-PPO.** We implement both ARPO and BARPO on top of the PPO algorithm (Schulman et al., 2017), a widely adopted policy-gradient method that serves as a standard baseline in adversarially robust DRL research. To improve convergence and efficiency, we integrate SPO dynamics into the bilevel framework using a regularization weight $\kappa$, resulting in a practical instantiation of BARPO, referred to as BAR-PPO. For solving the inner optimization problem, we employ SGLD (Gelfand & Mitter, 1991) to obtain the adversarial perturbation policy $\nu^\diamond$. In the outer optimization, we treat the inner solution as fixed and omit second-order terms to facilitate efficient updates of the primary policy. A complete description and further implementation details can be found in Appendix F. In the following, *BARPO* refers to the combination of SPO-based guidance with the bilevel framework, *BARPO without guidance* uses only the bilevel framework, *ARPO* uses only the maximin framework, and *ARPO with guidance* improves the maximin training through combining the SPO guidance.

## 5.2 COMPARISON RESULTS

**BARPO Consistently Improves over ARPO.** Figure 5 compares the natural and robust performance of four methods: SPO, ARPO, BARPO without SPO guidance, and BARPO with SPO guidance. Detailed evaluation results are provided in Appendix G.1. Notably, BARPO without SPO guidance consistently outperforms ARPO across all environments, improving both natural and robust returns. Even in environments like Walker2d and Halfcheetah, it achieves substantial gains in robustness. Specifically, BARPO without SPO guidance improves the natural return by approximately 75%, 104%, 156%, and 113% on Hopper, Walker2d, Halfcheetah, and Ant, respectively, while boosting robust return by about 4%, 173%, 209%, and 61% in the worst-case setting. Remarkably, in the Hopper and Walker2d tasks, BARPO without SPO guidance achieves natural returns on par with state-of-the-art methods. As further illustrated in Figure 6, BARPO improves the training dynamics by guiding policy learning toward regions in the landscape that yield higher returns.

**BAR-PPO Exhibits Superior Natural and Robust Performance.** As shown in Table 2, BAR-PPO demonstrates strong overall performance across all four MuJoCo environments, achieving both high natural returns and robust resilience to adversarial perturbations. In particular, BAR-PPO outperforms vanilla PPO in terms of natural returns on Hopper and Walker2d, and achieves comparable performance on Halfcheetah and Ant. In terms of worst-case robust returns, BAR-PPO significantly surpasses all baselines, with improvements of 4%, 154%, 40%, and 20% over the second best, respectively. Notably, BAR-PPO achieves the highest robustness across Walker2d, Halfcheetah, and Ant, and is on par with the strongest robustness on Hopper, with a negligible gap of only 0.3%.

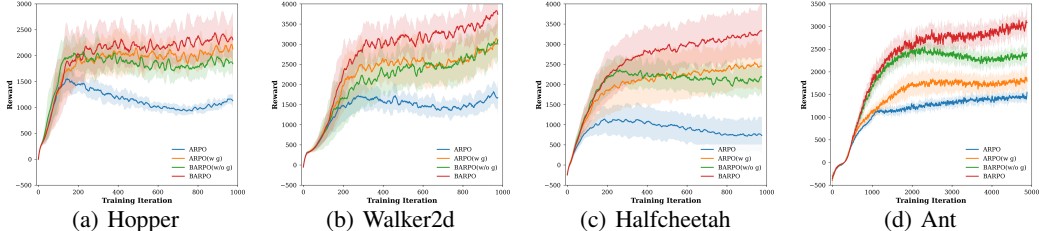

| (a) Hopper | (b) Walker2d | (c) Halfcheetah | (d) Ant |

Figure 6: Natural training curves of ARPO and BARPO with and without SPO-guidance.

Table 3: Comparisons between BARPO and ARPO with SPO-based guidance.

| Environment | Method | Natural Return | Random | Critic | MAD | RS | SA-RL | PA-AD | Worst | Worst-case Robustness |
|---|---|---|---|---|---|---|---|---|---|---|
| | | | | | Return under Attack | | | | | |
| **Hopper** | ARPO (w g) | **3699** $\pm$ 11 | 3652 $\pm$ 222 | **3693** $\pm$ 8 | 3132 $\pm$ 692 | 1130 $\pm$ 30 | 1278 $\pm$ 359 | 1756 $\pm$ 415 | 1130 $\pm$ 30 | -0.694 |
| ($\epsilon = 0.075$) | BARPO | 3684 $\pm$ 20 ↓0.4% | **3692** $\pm$ 24 | 3678 $\pm$ 19 | **3437** $\pm$ 555 | **1340** $\pm$ 42 | **1728** $\pm$ 62 | **1908** $\pm$ 410 | **1340** $\pm$ 42 ↑19% | **-0.636** ↑8.4% |
| **Walker2d** | ARPO (w g) | 4206 $\pm$ 30 | 4231 $\pm$ 40 | 4283 $\pm$ 38 | 4226 $\pm$ 68 | 3139 $\pm$ 1442 | 3565 $\pm$ 988 | 2496 $\pm$ 1871 | 2496 $\pm$ 1871 | **-0.406** |
| ($\epsilon = 0.05$) | BARPO | **4732** $\pm$ 86 ↑11% | **4718** $\pm$ 59 | **4727** $\pm$ 36 | **4750** $\pm$ 65 | **4646** $\pm$ 53 | **4285** $\pm$ 1282 | **2699** $\pm$ 1192 | **2699** $\pm$ 1192 ↑8.1% | -0.436 ↓7.4% |
| **Halfcheetah** | ARPO (w g) | **4997** $\pm$ 71 | **4934** $\pm$ 57 | **5040** $\pm$ 50 | 2570 $\pm$ 1588 | 3605 $\pm$ 48 | 1356 $\pm$ 1165 | 1086 $\pm$ 758 | 1086 $\pm$ 758 | -0.783 |
| ($\epsilon = 0.15$) | BARPO | 4837 $\pm$ 99 ↓3.2% | 4741 $\pm$ 501 | 4803 $\pm$ 54 | **4729** $\pm$ 105 | **4265** $\pm$ 1077 | **3894** $\pm$ 1322 | **3181** $\pm$ 1593 | **3181** $\pm$ 1593 ↑193% | **-0.342** ↑56% |
| **Ant** | ARPO (w g) | **5390** $\pm$ 120 | **5227** $\pm$ 186 | 4768 $\pm$ 92 | 4059 $\pm$ 921 | 2670 $\pm$ 1365 | 1185 $\pm$ 253 | 1757 $\pm$ 811 | 1185 $\pm$ 253 | -0.773 |
| ($\epsilon = 0.15$) | BARPO | 5024 $\pm$ 117 ↓6.8% | 4979 $\pm$ 114 | **4777** $\pm$ 122 | **4843** $\pm$ 120 | **4171** $\pm$ 826 | **3367** $\pm$ 902 | **2825** $\pm$ 757 | **2825** $\pm$ 757 ↑138% | **-0.438** ↑43% |

## 5.3 ABLATION STUDIES

**Effect of SPO-Guidance.** As shown in Figures 5 and 6, incorporating SPO-based guidance further enhances the natural performance of BARPO, particularly in more complex environments such as Halfcheetah and Ant. In certain cases, the enhancement in natural returns is accompanied by corresponding improvements in robust returns, as observed in the Hopper and Ant tasks. However, this incorporation often involves a trade-off, where improvements in natural performance may lead to a reduction in the worst-case robustness, as presented in the Walker2d and Halfcheetah environments.

**Comparisons between BARPO and ARPO with SPO-Guidance.** To further validate the effectiveness of BARPO, we compare it with ARPO enhanced by SPO-based guidance. As presented in Table 3, on the Hopper, Halfcheetah, and Ant tasks, BARPO exhibits a slight reduction in natural returns compared to ARPO with guidance, approximately 0.4%, 3.2%, and 6.8%, respectively. However, BARPO achieves significantly higher robust returns and markedly stronger robustness, with improvements of 19%, 193%, 138% in robust returns, and 8.4%, 56%, and 43% in robustness metrics. On the Walker2d task, BARPO attains gains in both natural and robust returns, with a minor compromise in robustness. These findings indicate that although integrating SPO guidance into ARPO can enhance performance to some extent, this direct combination does not effectively mitigate the reshaping effect induced by the strongest adversary. Consequently, numerous sticky FOSPs with low returns persist, posing challenges for optimization. In contrast, BARPO substantially reshapes the landscape, enabling the discovery of improved navigation that yields higher returns.

We also conduct additional ablations in Appendices G.6 and G.7 to identify the effects of bilevel structure, SPO guidance, and KL surrogate. These further validate the effectiveness of BARPO.

## 6 CONCLUSION

In this paper, we explore whether the theoretical alignment between optimality and robustness can be realized in practical policy optimization, and propose BARPO, a novel bilevel framework that unifies standard and adversarially robust policy optimization. Through extensive analysis, we reveal a fundamental tradeoff between optimality and robustness in existing policy gradient methods, exposing a critical gap between theory and practice. Notably, our work demystifies the underlying cause of this tension from the perspective of optimization landscapes and value space geometry. Specifically, the strongest adversaries in ARPO significantly reshape the learning landscapes by introducing numerous sticky and deceptive local optima. While this promotes robustness, it also hinders effective policy improvement by making the landscape more rugged and difficult to navigate. BARPO refines the landscapes by modulating the adversary strength, thereby creating pathways toward superior policies and reducing the prevalence of poor sticky local optima. Surprisingly, this bilevel framework empirically showcases the potential for approaching the globally optimal robust policy, suggesting a promising step toward bridging theoretical guarantees and practical performance.

ETHICS STATEMENT

All authors of this paper have carefully reviewed and adhered to the ICLR Code of Ethics.

REPRODUCIBILITY STATEMENT

All benchmarks, as well as the experimental setup, are based on or derived from open-sourced work. We have given relevant references in the paper, and the details of our algorithm and experiments are provided in Section 5 and Appendices F and G.

We have provided detailed analysis and complete proofs for each theorem and proposition in the Appendices C, D, E, and H.

ACKNOWLEDGMENTS

Congying Han, Haoran Li, Jiayu Lv, and Tiande Guo acknowledge funding support from the National Key R&D Program of China (2021YFA1000403) and the National Natural Science Foundation of China (Nos. 12431012, U23B2012).

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

# A   OUTLINE OF APPENDIX

- **Appendix B:** We review related work on adversarial attacks and defenses for reinforcement learning against perturbations on state observation.

- **Appendix C:** Detailed theoretical analysis of ARPO, including proofs for the adversary's policy gradient (Theorem 3.1) in Appendix C.1, convergence (Theorem 3.2) in Appendix C.2, and learning dynamics in Appendix C.3.

- **Appendix D:** An intuitive analysis of a toy ISA-MDP, corresponding to Propositions 3.1 and 3.2.

- **Appendix E:** The analysis and proof for the KL-based surrogate adversary (Theorem 4.1).

- **Appendix F:** Additional implementation details, including pseudocode for ARPO (Appendix F.1) and BARPO (Appendix F.2), hyperparameter settings (Appendix F.3), and compute resources (Appendix F.4).

- **Appendix G:** Additional experimental results, including detailed comparisons (Appendix G.1), statistical significance analysis (Appendix G.2), extended experiments on Humanoid (Appendix G.3), and ablation studies on the regularization coefficient $\kappa$ (Appendix G.4) and our proposed components (Appendices G.6 and G.7).

- **Appendix H:** Further discussions clarifying our contributions and convergence proof (Appendix H.1), the novelty of the bilevel framework (Appendix H.2), and a deeper theoretical examination of BARPO (Appendices H.3 and H.4).

- **Appendices I and J:** A discussion of limitations, future work, and a clarification on the use of large language models.

# B   RELATED WORK

## B.1   ATTACKS ON STATE OBSERVATIONS OF DRL AGENTS

**Adversarial Attacks on State Observations of DRL Agents.**   The susceptibility of DRL agents to adversarial attacks was initially exposed by Huang et al. (2017), who demonstrated that DRL policies could be significantly disrupted using the Fast Gradient Sign Method (FGSM) (Goodfellow et al., 2015) in Atari environments. This seminal discovery laid the foundation for a wave of subsequent studies on adversarial strategies and policy robustness in DRL. Building on this, Lin et al. (2017) and Kos & Song (2017) proposed limited-step perturbation techniques aimed at misleading DRL policies, revealing that even small, strategically applied modifications could impair agent behavior. In a further advancement, Pattanaik et al. (2018) exploited the critic function alongside gradient descent to craft adversarial inputs that degrade policy performance more systematically. Behzadan & Munir (2017a) extended the threat model by introducing black-box attacks on DQN, confirming that adversarial examples could successfully transfer across different network architectures, thereby emphasizing their real-world applicability. Meanwhile, Inkawhich et al. (2020) demonstrated that even adversaries with access restricted to action and reward signals, without internal model information, could still inflict considerable harm. In the domain of continuous control, Weng et al. (2019) devised a two-step adversarial framework leveraging learned dynamics models to generate more effective attacks. Furthermore, Zhang et al. (2021) and Sun et al. (2022) advanced the field by training adversarial agents using reinforcement learning, giving rise to adaptive attack strategies known as SA-RL and PA-AD. Korkmaz (2023) investigated adversarial directions in the Arcade Learning Environment and discovered that even state-of-the-art robust agents (Zhang et al., 2020c; Oikarinen et al., 2021) remain susceptible to perturbations aligned with policy-independent sensitivity axes. Most recently, Liang et al. (2024) proposed a temporally-coupled attack strategy that exploits temporal correlations to further compromise the performance of robust policies.

**Other Attacks on DRL Agents.**   Research by Kiourti et al. (2020); Wang et al. (2021); Bharti et al. (2022); Guo et al. (2023) delved deeper into backdoor attacks within reinforcement learning, revealing profound vulnerabilities that pose significant threats to policy integrity. In a novel contribution, Lu et al. (2023) proposed an adversarial cheap talk framework, where an attacker is trained via meta-learning to manipulate agent behavior through indirect communication. Franzmeyer et al. (2024) introduced a dual ascent-based approach to train an end-to-end illusory attack capable of deceiving DRL agents effectively. In the context of multi-agent environments, Gleave et al. (2020)

examined how adversarial policies can manipulate or destabilize cooperative or competitive dynamics among agents.

Collectively, these studies reveal the broad and persistent vulnerability of DRL agents to adversarial manipulation across diverse threat models and environments. From observation perturbations and black-box attacks to communication-based deception and multi-agent exploitation, adversaries have consistently demonstrated the ability to significantly degrade policy performance, highlighting the urgent need for continued advancements in defense mechanisms.

### B.2 DEVELOPING ROBUST DRL AGENTS AGAINST PERTURBATIONS ON STATE OBSERVATION

**Early Attempts to Develop Robust Agents.** Early efforts by Kos & Song (2017); Behzadan & Munir (2017b) explored incorporating adversarial states into the replay buffer during training on Atari environments, although these approaches yielded only limited robustness improvements. To address this, Fischer et al. (2019) proposed a decoupled architecture for DQN, separating it into a Q-network and a policy network, and robustly trained the policy component using generated adversarial states under provable robustness guarantees.

**Smoothing Techniques.** Shen et al. (2020) demonstrated that applying smoothness regularization can simultaneously improve both the natural performance and robustness of TRPO (Schulman et al., 2015) and DDPG (Silver et al., 2014) algorithms. Building on this idea, Wu et al. (2022a) and Kumar et al. (2022) employed Randomized Smoothing (RS) techniques to achieve certifiable robustness guarantees. More recently, Sun et al. (2024) proposed an innovative smoothing approach designed to mitigate the common issue of robustness overestimation, thereby enhancing both natural and robust returns in tandem. Importantly, these smoothing methods are complementary to our work and can be seamlessly integrated into our proposed algorithms.

**Robust Multi-Agent DRL.** In the context of multi-agent systems, He et al. (2023) investigated state adversaries within Markov Games and developed robust multi-agent Q-learning and actor-critic algorithms to achieve robust equilibrium solutions. Building on robustness regularization techniques (Shen et al., 2020; Zhang et al., 2020c), Bukharin et al. (2024) extended these methods to multi-agent environments by considering sub-optimal Lipschitz continuous policies in smooth settings. Liu et al. (2024a) proposed an adversarial training framework employing two timescales to ensure effective convergence towards robust policies. Another research direction involves alternating training between agents and learned adversaries (Zhang et al., 2021; Sun et al., 2022), which Liang et al. (2024) further formalized within a game-theoretic framework.

**Beyond the Worst-Case Robustness.** Furthermore, Liu et al. (2024b) introduced an adaptive defense mechanism during testing, leveraging a set of non-dominated policies. The latest work by Belaire et al. (2025) considers the partial observability setting and balances the value optimization with robustness based on beliefs.

**Adversarially Robust Training for DRL Agents.** In this paper, we mainly focus on the adversarially robust training paradigm. Zhang et al. (2020c) formally modeled state-adversarial reinforcement learning as a state-adversarial Markov Decision Process and highlighted the potential absence of an Optimal Robust Policy. To tackle this, they introduced a TV-distance-based regularization technique to balance robustness with nominal performance. Building on this, Oikarinen et al. (2021) employed certified robustness bounds to construct the adversarial loss, which they integrated with standard training objectives. Liang et al. (2022) enhanced training efficiency by estimating worst-case value functions and combining them with classical Temporal Difference (TD) target (Sutton, 1988) or Generalized Advantage Estimation (GAE) (Schulman et al., 2016). Nie et al. (2024) proposed a DRL framework for discrete action spaces based on SortNet (Zhang et al., 2022), which ensures global Lipschitz continuity and eliminates the need for training additional adversarial agents. More recently, Li et al. (2024; 2025) developed the universal ISA-MDP framework, within which they formally proved the existence of the ORP and demonstrated its consistency with Bellman optimality policies. Moreover, they established that an infinity-measurement error is a necessary condition for adversarial robustness. However, their theoretical analysis is limited to worst-case settings for RE-

INFORCE and PPO. In contrast, our analysis builds upon practical policy gradient methods, making it broadly applicable to real-world policy optimization.

These studies underscore the crucial importance of developing robust DRL policies and highlight the ongoing challenges in enhancing adversarial robustness in DRL agents.

### B.3 ROBUST RL AGENTS AGAINST PERTURBATIONS ON TRANSITION DYNAMICS

Research on robustness against transition uncertainty (Pinto et al., 2017; Gu et al., 2019; Kamalaruban et al., 2020; Dong et al., 2025) is primarily formulated as Robust Markov Decision Processes (RMDPs) (Nilim & Ghaoui, 2003; Iyengar, 2005; Wiesemann et al., 2013). This paradigm differs fundamentally from SA-MDP. In SA-MDP, the adversary corrupts the policy input, forcing the agent to act on distorted observations. Consequently, robustness depends on the policy's stability with respect to state variations. In contrast, RMDPs involve perturbations to the environment's dynamics. The agent observes the true state, but the transition leads to a worst-case outcome. Since the decision precedes the perturbation, state-space smoothness does not guarantee robustness against dynamic shifts.

## C ADVERSARIALLY ROBUST POLICY OPTIMIZATION

In this section, we present a detailed theoretical analysis of ARPO. In the subsection C.1, we show the analysis of policy gradient for the adversary. In the subsection C.2, we show the convergence analysis of ARPO. And in the subsection C.3, we explore the learning dynamics in the neighborhood of FOSPs in both ARPO and SPO.

### C.1 POLICY GRADIENT FOR ADVERSARY

**Theorem C.1** (Policy Gradient for Adversary). *Given a policy $\pi_\theta$, for all state $s \in \mathcal{S}$, we have*

$$
\begin{aligned}
\nabla_\vartheta V^{\pi_\theta \circ \nu_\vartheta}(s) &= \mathbb{E}_{\tau \sim \pi_\theta \circ \nu_\vartheta, \mathbb{P}}\left[ R(\tau) \sum_{t=0}^\infty \nabla_\vartheta \log \pi_\theta(a_t | \nu_\vartheta(s_t)) \right] \\
&= \mathbb{E}_{\tau \sim \pi_\theta \circ \nu_\vartheta, \mathbb{P}}\left[ \sum_{t=0}^\infty \gamma^t Q^{\pi_\theta \circ \nu_\vartheta}(s_t, a_t) \nabla_\vartheta \log \pi_\theta(a_t | \nu_\vartheta(s_t)) \right] \\
&= \frac{1}{1-\gamma} \mathbb{E}_{(s', a') \sim d^{\pi_\theta \circ \nu_\vartheta}} \left[ Q^{\pi_\theta \circ \nu_\vartheta}(s', a') \nabla_\vartheta \log \pi_\theta(a' | \nu_\vartheta(s')) \right].
\end{aligned}
$$

*Specifically, consider the direct parameterization representation for adversary $\nu_\vartheta : \mathcal{S} \to \mathcal{S}$, $s \mapsto s + \vartheta_s \in B(s)$. Then, for any state $s_i \in \mathcal{S}$, we have the state-wise policy gradient for the adversary*

$$
\begin{aligned}
&\nabla_{\vartheta_{s_i}} V^{\pi_\theta \circ \nu_\vartheta}(s) \\
&= \mathbb{E}_{\tau \sim \pi_\theta \circ \nu_\vartheta, \mathbb{P}}\left[ R(\tau) \sum_{t=0}^\infty \nabla_{\vartheta_{s_t}} \log \pi_\theta(a_t | s_t + \vartheta_{s_t}) \cdot \mathbb{I}(s_t = s_i) \right], \\
&= \mathbb{E}_{\tau \sim \pi_\theta \circ \nu_\vartheta, \mathbb{P}}\left[ \sum_{t=0}^\infty \gamma^t Q^{\pi_\theta \circ \nu_\vartheta}(s_t, a_t) \nabla_{\vartheta_{s_t}} \log \pi_\theta(a_t | s_t + \vartheta_{s_t}) \cdot \mathbb{I}(s_t = s_i) \right] \\
&= \frac{1}{1-\gamma} \mathbb{E}_{(s', a') \sim d^{\pi_\theta \circ \nu_\vartheta}} \left[ Q^{\pi_\theta \circ \nu_\vartheta}(s', a') \nabla_{\vartheta_{s'}} \log \pi_\theta(a' | s' + \vartheta_{s'}) \cdot \mathbb{I}(s' = s_i) \right],
\end{aligned} \tag{2}
$$

*where $d^{\pi \circ \nu}$ is the state-action visitation distribution under $\pi \circ \nu$ and $\mathbb{I}(\cdot)$ is the indicator function.*

**Remark C.1.** *Equation (2) indicates that for computing the strongest adversary, we can decompose it into calculating the strongest adversary in every state after sampling.*

*Proof.* The definition of $V^{\pi_\theta \circ \nu_\vartheta}(s)$

$$
V^{\pi_\theta \circ \nu_\vartheta}(s) = \mathbb{E}_{\tau \sim \pi_\theta \circ \nu_\vartheta, \mathbb{P}}\left[ \sum_{t=0}^\infty \gamma^t r(s_t, a_t) | s_0 = s \right] = \mathbb{E}_{\tau \sim \pi_\theta \circ \nu_\vartheta, \mathbb{P}}\left[ R(\tau) | s_0 = s \right].
$$

The probability of attaining trajectory $\tau = \{s_0, a_0, s_1, a_1, \dots\}$ under $\pi_\theta$ and $\nu_\vartheta$ with initial state distribution $\mu_0$ is

$$P_{\theta,\vartheta}(\tau) = \mu_0(s_0)\pi_\theta(a_0|\nu_\vartheta(s_0)) \prod_{i=1}^\infty \mathbb{P}(s_i|s_{i-1}, a_{i-1})\pi_\theta(a_i|\nu_\vartheta(s_i))$$

$$= \mu_0(s_0)(\pi_\theta \circ \nu_\vartheta)(a_0|s_0) \prod_{i=1}^\infty \mathbb{P}(s_i|s_{i-1}, a_{i-1})(\pi_\theta \circ \nu_\vartheta)(a_i|s_i).$$

For any state $s$, compute the gradient of the adversarial value function with respect to $\nu$.

$$\nabla_\vartheta V^{\pi_\theta \circ \nu_\vartheta}(s)$$

$$= \nabla_\vartheta \int_\tau R(\tau) P_{\theta,\vartheta}(\tau|s_0 = s)$$

$$= \int_\tau R(\tau) \nabla_\vartheta P_{\theta,\vartheta}(\tau|s_0 = s)$$

$$= \int_\tau R(\tau) P_{\theta,\vartheta}(\tau|s_0 = s) \nabla_\vartheta \log P_{\theta,\vartheta}(\tau|s_0 = s)$$

$$= \int_\tau R(\tau) P_{\theta,\vartheta}(\tau) \sum_{t=0}^\infty \nabla_\vartheta \log(\pi_\theta \circ \nu_\vartheta)(a_t|s_t)$$

$$= \mathbb{E}_{\tau \sim \pi_\theta \circ \nu_\vartheta, \mathbb{P}} \left[ R(\tau) \sum_{t=0}^\infty \nabla_\vartheta \log(\pi_\theta \circ \nu_\vartheta)(a_t|s_t) \right]$$

$$= \mathbb{E}_{\tau \sim \pi_\theta \circ \nu_\vartheta, \mathbb{P}} \left[ R(\tau) \sum_{t=0}^\infty \nabla_\vartheta \log \pi_\theta(a_t|\nu_\vartheta(s_t)) \right].$$

Let $\tau_{:t} = \{s_0, a_0, s_1, a_1, \dots, s_t, a_t\}$. Furthermore, we have that

$$\nabla_\vartheta V^{\pi_\theta \circ \nu_\vartheta}(s)$$

$$= \mathbb{E}_{\tau \sim \pi_\theta \circ \nu_\vartheta, \mathbb{P}} \left[ \sum_{i=0}^\infty \gamma^i r(s_i, a_i) \sum_{t=0}^\infty \nabla_\vartheta \log \pi_\theta(a_t|\nu_\vartheta(s_t)) \right]$$

$$= \mathbb{E}_{\tau \sim \pi_\theta \circ \nu_\vartheta, \mathbb{P}} \left[ \sum_{t=0}^\infty \sum_{i=0}^\infty \gamma^i r(s_i, a_i) \nabla_\vartheta \log \pi_\theta(a_t|\nu_\vartheta(s_t)) \right]$$

$$= \mathbb{E}_{\tau \sim \pi_\theta \circ \nu_\vartheta, \mathbb{P}} \left[ \sum_{t=0}^\infty \mathbb{E}_{\tau_{t:}} \left[ \sum_{i=0}^\infty \gamma^i r(s_i, a_i) \nabla_\vartheta \log \pi_\theta(a_t|\nu_\vartheta(s_t)) \mid \tau_{:t-1} \right] \right]$$

$$= \mathbb{E}_{\tau \sim \pi_\theta \circ \nu_\vartheta, \mathbb{P}} \left[ \sum_{t=0}^\infty \mathbb{E}_{\tau_{t:}} \left[ \left( \sum_{i<t} \gamma^i r(s_i, a_i) + \sum_{i \geq t} \gamma^i r(s_i, a_i) \right) \nabla_\vartheta \log \pi_\theta(a_t|\nu_\vartheta(s_t)) \mid \tau_{:t-1} \right] \right]$$

$$= \mathbb{E}_{\tau \sim \pi_\theta \circ \nu_\vartheta, \mathbb{P}} \left[ \sum_{t=0}^\infty \mathbb{E}_{\tau_{t:}} \left[ \sum_{i=t}^\infty \gamma^i r(s_i, a_i) \nabla_\vartheta \log \pi_\theta(a_t|\nu_\vartheta(s_t)) \mid \tau_{:t-1} \right] \right]$$

$$= \mathbb{E}_{\tau \sim \pi_\theta \circ \nu_\vartheta, \mathbb{P}} \left[ \sum_{t=0}^\infty \sum_{i=t}^\infty \gamma^i r(s_i, a_i) \nabla_\vartheta \log \pi_\theta(a_t|\nu_\vartheta(s_t)) \right]$$

$$= \mathbb{E}_{\tau \sim \pi_\theta \circ \nu_\vartheta, \mathbb{P}} \left[ \sum_{t=0}^\infty \mathbb{E}_{\tau_{t+1:}} \left[ \sum_{i=t}^\infty \gamma^i r(s_i, a_i) \nabla_\vartheta \log \pi_\theta(a_t|\nu_\vartheta(s_t)) \mid \tau_{:t} \right] \right]$$

$$= \mathbb{E}_{\tau \sim \pi_\theta \circ \nu_\vartheta, \mathbb{P}} \left[ \sum_{t=0}^\infty \mathbb{E}_{\tau_{t+1:}} \left[ \sum_{i=t}^\infty \gamma^i r(s_i, a_i) \mid s_t, a_t \right] \nabla_\vartheta \log \pi_\theta(a_t|\nu_\vartheta(s_t)) \right]$$

$$= \mathbb{E}_{\tau \sim \pi_\theta \circ \nu_\vartheta, \mathbb{P}} \left[ \sum_{t=0}^\infty \gamma^t Q^{\pi_\theta \circ \nu_\vartheta}(s_t, a_t) \nabla_\vartheta \log \pi_\theta(a_t|\nu_\vartheta(s_t)) \right].$$

---

**Algorithm 1** Adversarially Robust Policy Optimization with $\delta$-Approximation Adversary

---

**Input:** training steps $K$ in outer loop, outer loop step size $\eta_k$, inner loop PGD step $K_P$, inner loop PGD step size $\eta_P$, inner loop approximation accuracy $\delta$, batch size $B$

**Output:** adversarially robust policy $\pi_\theta$

1: Initialize $\theta$ and $\vartheta$
2: **for** $k = 1$ **to** $K$ **do**
3:      Initialize indicator vector $I = 1_{|\mathcal{S}|}$, PGD step number $j = 0$
4:      **while** $\|I\|_1 > 0$ & $j < K_P$ **do**
5:          Sample trajectories $\tau \sim \pi_\theta \circ \nu_\vartheta, \mathbb{P}$ until the batch is full.
6:          Compute the unbiased estimation of the action-value function $Q^{\pi_\theta \circ \nu_\vartheta}(s_t, a_t)$: $\widehat{Q^{\pi_\theta \circ \nu_\vartheta}}(s_t, a_t) :=$ $\sum_{t' \geq t} \gamma^{t'-t} r(s_{t'}, a_{t'})$, for all $(s_t, a_t)$ in the batch
7:          **for** $s$ in the batch **do**
8:              $\nu_\vartheta(s) \longleftarrow \nu_\vartheta(s) - \eta_P \cdot \text{sign}\left(\widehat{Q^{\pi_\theta \circ \nu_\vartheta}}(s, a) \cdot \nabla_\vartheta \log \pi_\theta(a|\nu_\vartheta(s))\right) \cdot I(s)$
9:              $\nu_\vartheta(s) \longleftarrow \text{clip}(\nu_\vartheta(s), s - \epsilon, s + \epsilon)$
10:              $I(s) = \mathbb{I}\left(\max_{s_\nu \in B(s)} \left\langle \nu_\vartheta(s) - s_\nu, \frac{1}{1-\gamma}\widehat{Q^{\pi_\theta \circ \nu_\vartheta}}(s, a) \cdot \nabla_\vartheta \log \pi_\theta(a|\nu_\vartheta(s))\right\rangle > \delta\right)$
11:              $j \longleftarrow j + 1$
12:          **end for**
13:      **end while**
14:      Compute the unbiased estimation of the gradient of the value function $\nabla_\theta V^{\pi_\theta \circ \nu_\vartheta}(\mu_0)$: $\nabla_\theta \widehat{V^{\pi_\theta \circ \nu_\vartheta}}(\mu_0) := \frac{1}{(1-\gamma)|B|} \sum_{(s,a) \in B} \widehat{Q^{\pi_\theta \circ \nu_\vartheta}}(s, a) \cdot \nabla_\theta \log \pi_\theta(a|\nu_\vartheta(s))$
15:      $\theta \longleftarrow \theta + \eta_k \nabla_\theta \widehat{V^{\pi_\theta \circ \nu_\vartheta}}(\mu_0)$
16: **end for**

---

Furthermore, through the definition of the state-action visitation distribution following the policy $\pi_\theta \circ \nu_\vartheta$ starting from $s$: $d_s^{\pi_\theta \circ \nu_\vartheta}(s', a') = \mathbb{E}_{s_t \sim \mathbb{P}(\cdot|s_{t-1}, a_{t-1}), a_t \sim \pi_\theta \circ \nu_\vartheta(\cdot|s_t)} \left[(1-\gamma) \sum_{t=0}^{\infty} \gamma^t \mathbb{I}(s_t = s', a_t = a')|s_0 = s\right]$, we have that

$$\nabla_\vartheta V^{\pi_\theta \circ \nu_\vartheta}(s) = \frac{1}{1-\gamma}\mathbb{E}_{(s',a') \sim d^{\pi_\theta \circ \nu_\vartheta}}\left[Q^{\pi_\theta \circ \nu_\vartheta}(s', a')\nabla_\vartheta \log \pi_\theta(a'|\nu_\vartheta(s'))\right].$$

Specifically, for the representation for adversary $\nu_\vartheta : \mathcal{S} \to \mathcal{S}, \ s \mapsto s + \vartheta_s \in B(s)$, we have

$$\nabla_{\vartheta_{s_i}} V^{\pi_\theta \circ \nu_\vartheta}(s)$$

$$= \mathbb{E}_{\tau \sim \pi_\theta \circ \nu_\vartheta, \mathbb{P}}\left[R(\tau)\sum_{t=0}^{\infty}\nabla_{\vartheta_{s_i}}\log \pi_\theta(a_t|\nu_\vartheta(s_t))\right]$$

$$= \mathbb{E}_{\tau \sim \pi_\theta \circ \nu_\vartheta, \mathbb{P}}\left[R(\tau)\sum_{t=0}^{\infty}\nabla_{\vartheta_{s_i}}\log \pi_\theta(a_t|s_t + \vartheta_{s_t})\right]$$

$$= \mathbb{E}_{\tau \sim \pi_\theta \circ \nu_\vartheta, \mathbb{P}}\left[R(\tau)\sum_{t=0}^{\infty}\nabla_{\vartheta_{s_t}}\log \pi_\theta(a_t|s_t + \vartheta_{s_t}) \cdot \mathbb{I}(s_t = s_i)\right].$$

Similarly, we have

$$\nabla_{\vartheta_{s_i}} V^{\pi_\theta \circ \nu_\vartheta}(s) = \mathbb{E}_{\tau \sim \pi_\theta \circ \nu_\vartheta, \mathbb{P}}\left[\sum_{t=0}^{\infty}\gamma^t Q^{\pi_\theta \circ \nu_\vartheta}(s_t, a_t)\nabla_{\vartheta_{s_t}}\log \pi_\theta(a_t|s_t + \vartheta_{s_t}) \cdot \mathbb{I}(s_t = s_i)\right]$$

$$= \frac{1}{1-\gamma}\mathbb{E}_{(s',a') \sim d^{\pi_\theta \circ \nu_\vartheta}}\left[Q^{\pi_\theta \circ \nu_\vartheta}(s', a')\nabla_{\vartheta_{s'}}\log \pi_\theta(a'|s' + \vartheta_{s'}) \cdot \mathbb{I}(s' = s_i)\right].$$

This completes the proof. □

## C.2 CONVERGENCE OF ADVERSARIALLY ROBUST POLICY OPTIMIZATION

We analyze the convergence properties of ARPO with $\delta$-Approximation Adversary (Algorithm 1) in this subsection.

### C.2.1 NOTATIONS

For a sampling trajectory $\tau = (s_0, a_0, r_0, \dots) \sim \pi_\theta \circ \nu_\vartheta, \mathbb{P}$, denote the unbiased estimation of the action-value function $Q^{\pi_\theta \circ \nu_\vartheta}$ as

$$\widehat{Q^{\pi_\theta \circ \nu_\vartheta}}(s_t, a_t) := \sum_{t' \geq t} \gamma^{t'-t} r(s_{t'}, a_{t'}), \forall (s_t, a_t) \in \tau.$$

For simplicity of writing and reading of the latter, denote the sampled estimate as:

$$\hat{v}(s, a; \theta, \vartheta) := \frac{1}{1 - \gamma} \operatorname{sg}\left(\widehat{Q^{\pi_\theta \circ \nu_\vartheta}}(s, a)\right) \cdot \log \pi_\theta\left(a | \nu_\vartheta(s)\right), \tag{3}$$

whereas $\operatorname{sg}(\cdot)$ is the stop-gradient operator, meaning that $\widehat{Q^{\pi_\theta \circ \nu_\vartheta}}(s, a)$ is seen as a function according to $(s, a)$ and is detached from the gradient operation for $(\theta, \vartheta)$ in the following context. This implies that

$$\nabla_\theta \hat{v}(s, a; \theta, \vartheta) = \frac{1}{1 - \gamma} \widehat{Q^{\pi_\theta \circ \nu_\vartheta}}(s, a) \cdot \nabla_\theta \log \pi_\theta\left(a | \nu_\vartheta(s)\right)$$

$$\nabla_\vartheta \hat{v}(s, a; \theta, \vartheta) = \frac{1}{1 - \gamma} \widehat{Q^{\pi_\theta \circ \nu_\vartheta}}(s, a) \cdot \nabla_\vartheta \log \pi_\theta\left(a | \nu_\vartheta(s)\right).$$

Furthermore, denote the unbiased estimation of the gradient of the value function $\nabla_\theta V^{\pi_\theta \circ \nu_\vartheta}(\mu_0)$ as

$$\nabla_\theta \widehat{V^{\pi_\theta \circ \nu_\vartheta}}(\mu_0) := \frac{1}{|\tau|} \sum_{(s,a) \in \tau} \nabla_\theta \hat{v}(s, a; \theta, \vartheta),$$

and denote the unbiased estimation of the gradient of the value function $\nabla_\vartheta V^{\pi_\theta \circ \nu_\vartheta}(\mu_0)$ as

$$\nabla_\vartheta \widehat{V^{\pi_\theta \circ \nu_\vartheta}}(\mu_0) := \frac{1}{|\tau|} \sum_{(s,a) \in \tau} \nabla_\vartheta \hat{v}(s, a; \theta, \vartheta).$$

### C.2.2 PREPARATIONS

We quantify the solution of inner optimization, that is, the adversary, by extending the First-Order Stationary Condition (FOSC) proposed by Wang et al. (2019).

**Definition C.1** ($\delta$-Approximation Adversary). *For a given policy $\pi_\theta$ and the sampled estimation $\widehat{Q^{\pi_\theta \circ \nu_\vartheta}}(s, a)$, if $\nu_\vartheta(s)$ satisfies the following condition:*

$$\max_{s_\nu \in B(s)} \left\langle \nu_\vartheta(s) - s_\nu, \frac{1}{1 - \gamma} \widehat{Q^{\pi_\theta \circ \nu_\vartheta}}(s, a) \cdot \nabla_\vartheta \log \pi_\theta\left(a | \nu_\vartheta(s)\right) \right\rangle \leq \delta,$$

*then, we called that the $\nu_\vartheta(s)$ is $\delta$-approximation adversary for the strongest adversary $\nu^*(s; \pi_\theta)$.*

Before our main convergence analysis, we provide a few assumptions utilized in our analysis.

**Assumption C.1** (Lipschitz of Sampled Policy Gradient). *The function $\hat{v}(s, a; \theta, \vartheta)$ satisfies the gradient Lipschitz conditions as follows:*

$$\sup_{s,a,\vartheta} \|\nabla_\theta \hat{v}(s, a; \theta, \vartheta) - \nabla_\theta \hat{v}(s, a; \theta', \vartheta)\|_2 \leq L_{\theta\theta} \|\theta - \theta'\|_2,$$

$$\sup_{s,a,\vartheta} \|\nabla_\vartheta \hat{v}(s, a; \theta, \vartheta) - \nabla_\vartheta \hat{v}(s, a; \theta', \vartheta)\|_2 \leq L_{\vartheta\theta} \|\theta - \theta'\|_2,$$

$$\sup_{a,\theta} \|\nabla_\theta \hat{v}(s, a; \theta, \vartheta) - \nabla_\theta \hat{v}(s, a; \theta, \vartheta')\|_2 \leq L_{\theta\vartheta} \|\vartheta(s) - \vartheta'(s)\|_2, \ \forall s,$$

*where $L_{\theta\theta}$, $L_{\theta\vartheta}$, $L_{\vartheta\theta}$ are positive constants.*

Assumption C.1 has been made in previous research on adversarial robustness (Sinha et al., 2018; Wang et al., 2019). A line of studies on deep neural networks helps to justify this assumption (Allen-Zhu et al., 2019; Du et al., 2019; Zou et al., 2020).

**Assumption C.2** (Bounded Sampled Policy Gradient). *The gradient of the function $\hat{v}(s, a; \theta, \vartheta)$ with respect to $\theta$ is uniformly bounded, i.e., $\exists M_{\hat{v}} > 0$, such that*

$$\sup_{s,a,\theta,\vartheta} \|\nabla_\theta \hat{v}(s, a; \theta, \vartheta)\| \le M_{\hat{v}}.$$

Assumption C.2 can be verified by bounded rewards and the lower-bounded probability of the sampled action, which are mild conditions in practical algorithms.

**Assumption C.3** (Locally Strongly Convex Adversary). *For any state $s$ and action $a$, the function $\hat{v}(s, a; \theta, \vartheta)$ is locally $\mu$-strongly convex in $B_\epsilon(s) = \{\nu_\vartheta(s) := s + \vartheta_s \mid \|\vartheta_s\|_\infty \le \epsilon\}$, i.e., for any $\vartheta_1, \vartheta_2$, we have that*

$$\hat{v}(s, a; \theta, \vartheta_1) \ge \hat{v}(s, a; \theta, \vartheta_2) + \langle \nabla_\vartheta \hat{v}(s, a; \theta, \vartheta_2), \vartheta_1(s) - \vartheta_2(s) \rangle + \frac{\mu}{2} \|\vartheta_1(s) - \vartheta_2(s)\|_2^2.$$

Assumption C.3 has been studied in Sinha et al. (2018); Lee & Raginsky (2018), which can be verified by the relation between robust optimization and distributional robust optimization.

**Assumption C.4** (Lipschitz Conditions of State-Action Visitation Distribution). *The state-action visitation distribution under policy $\pi_\theta \circ \nu_\vartheta$ $d_{\theta,\vartheta}(s, a) := d_{\mu_0}^{\pi_\theta \circ \nu_\vartheta}(s, a) = \mathbb{E}_{s_0 \sim \mu_0(\cdot), s_t \sim \mathbb{P}(\cdot|s_{t-1}, a_{t-1}), a_t \sim \pi_\theta \circ \nu_\vartheta(\cdot|s_t)} \left[ (1 - \gamma) \sum_{t=0}^{\infty} \gamma^t \mathbb{I}(s_t = s, a_t = a) \right]$ satisfies the Lipschitz conditions under the total variation distance as follows:*

$$\sup_{\vartheta} \|d_{\theta,\vartheta} - d_{\theta',\vartheta}\|_{\mathrm{TV}} \le L_{d\theta} \|\theta - \theta'\|_2,$$

$$\sup_{\theta} \|d_{\theta,\vartheta} - d_{\theta,\vartheta'}\|_{\mathrm{TV}} \le L_{d\vartheta} \|\vartheta - \vartheta'\|_2.$$

Assumption C.4 has been derived by Pirotta et al. (2015) based on Lipschitz environments and policy. Certain natural environments show smooth reward functions and transition dynamics, especially in continuous control tasks where the transition dynamics come from some physical laws (Bukharin et al., 2024; Li et al., 2025), such as MuJoCo environments, where we conduct numerical experiments. These environments are thus Lipschitz. The policy parameterized by neural networks can be seen as Lipschitz based on neural network analysis (Allen-Zhu et al., 2019; Du et al., 2019; Zou et al., 2020).

**Assumption C.5** (Bounded Variance). *The variance of the stochastic gradient is bounded as follows:*

$$\mathbb{E} \left[ \left\| \nabla_\theta \widehat{V^{\pi_\theta \circ \nu^*}}(\pi_\theta)(\mu_0) - \nabla_\theta V^{\pi_\theta \circ \nu^*}(\pi_\theta)(\mu_0) \right\| \right] \le \sigma^2.$$

Assumption C.5 is a common assumption in the analysis of stochastic optimization.

The proof framework of Theorem C.2 is drawn on Wang et al. (2019); Pirotta et al. (2015). We begin by proving the following four technical lemmas.

**Lemma C.1** (Lipschitz of Strongest Adversary). *Suppose assumptions C.1 and C.3 hold. Then we have that the strongest adversary $\vartheta^*(s; \theta_1)$ is $\frac{L_{\vartheta\theta}}{\mu}$-smooth, i.e., given a state $s$, for any $\theta_1$ and $\theta_2$, we have that*

$$\|\vartheta^*(s; \theta_1) - \vartheta^*(s; \theta_2)\|_2 \le \frac{L_{\vartheta\theta}}{\mu} \|\theta_1 - \theta_2\|_2.$$

*Proof.* For any state $s$ and action $a$, under assumption C.3, we have that

$$\begin{aligned}
&\hat{v}(s, a; \theta_2, \vartheta^*(\theta_1)) \\
&\ge \hat{v}(s, a; \theta_2, \vartheta^*(\theta_2)) + \langle \nabla_\vartheta \hat{v}(s, a; \theta_2, \vartheta^*(\theta_2)), \vartheta^*(s; \theta_1) - \vartheta^*(s; \theta_2) \rangle \\
&\quad + \frac{\mu}{2} \|\vartheta^*(s; \theta_1) - \vartheta^*(s; \theta_2)\|_2^2 \\
&\ge \hat{v}(s, a; \theta_2, \vartheta^*(\theta_2)) + \frac{\mu}{2} \|\vartheta^*(s; \theta_1) - \vartheta^*(s; \theta_2)\|_2^2,
\end{aligned}$$

where the last inequality comes from $\langle \nabla_\vartheta \hat{v}(s, a; \theta_2, \vartheta^*(\theta_2)), \vartheta^*(s; \theta_1) - \vartheta^*(s; \theta_2) \rangle \ge 0$. Similarly, we have

$$\begin{aligned}
&\hat{v}(s, a; \theta_2, \vartheta^*(\theta_2)) \\
&\ge \hat{v}(s, a; \theta_2, \vartheta^*(\theta_1)) + \langle \nabla_\vartheta \hat{v}(s, a; \theta_2, \vartheta^*(\theta_1)), \vartheta^*(s; \theta_2) - \vartheta^*(s; \theta_1) \rangle \\
&\quad + \frac{\mu}{2} \|\vartheta^*(s; \theta_1) - \vartheta^*(s; \theta_2)\|_2^2.
\end{aligned}$$

Combining the above two inequalities, we have

$$
\begin{aligned}
&\mu\|\vartheta^*(s;\theta_1) - \vartheta^*(s;\theta_2)\|_2^2 \\
&\leq \langle \nabla_\vartheta \hat{v}(s,a;\theta_2,\vartheta^*(\theta_1)), \vartheta^*(s;\theta_1) - \vartheta^*(s;\theta_2)\rangle \\
&\leq \langle \nabla_\vartheta \hat{v}(s,a;\theta_2,\vartheta^*(\theta_1)) - \nabla_\vartheta \hat{v}(s,a;\theta_1,\vartheta^*(\theta_1)), \vartheta^*(s;\theta_1) - \vartheta^*(s;\theta_2)\rangle \\
&\leq \|\nabla_\vartheta \hat{v}(s,a;\theta_2,\vartheta^*(\theta_1)) - \nabla_\vartheta \hat{v}(s,a;\theta_1,\vartheta^*(\theta_1))\|_2 \cdot \|\vartheta^*(s;\theta_1) - \vartheta^*(s;\theta_2)\|_2 \\
&\leq L_{\vartheta\theta}\|\theta_2 - \theta_1\|_2 \cdot \|\vartheta^*(s;\theta_1) - \vartheta^*(s;\theta_2)\|_2.
\end{aligned}
$$

The second inequality comes from $\langle -\nabla_\vartheta \hat{v}(s,a;\theta_1,\vartheta^*(\theta_1)), \vartheta^*(s;\theta_1) - \vartheta^*(s;\theta_2)\rangle \geq 0$. The third inequality comes from the Cauchy-Schwarz inequality, and the last inequality comes from Assumption C.1. Therefore, we have that

$$
\|\vartheta^*(s;\theta_1) - \vartheta^*(s;\theta_2)\|_2 \leq \frac{L_{\vartheta\theta}}{\mu}\|\theta_2 - \theta_1\|_2.
$$

This proof is concluded. $\qquad\square$

**Lemma C.2** (Lipschitz of Sampled Policy Gradient against Strongest Adversary). *Suppose assumptions C.1 and C.3 hold. Then we have that the sampling estimation $\hat{v}(s,a;\theta,\vartheta^*(\theta))$ is $L_{\hat{v}}$-smooth, i.e., for any $\theta_1$ and $\theta_2$, we have that*

$$
\|\nabla_\theta \hat{v}(s,a;\theta_1,\vartheta^*(\theta_1)) - \nabla_\theta \hat{v}(s,a;\theta_2,\vartheta^*(\theta_2))\|_2 \leq L_{\hat{v}}\|\theta_1 - \theta_2\|_2,
$$

*where $L_{\hat{v}} = \frac{L_{\theta\vartheta}L_{\vartheta\theta}}{\mu} + L_{\theta\theta}$.*

*Proof.* For any $\theta_1$ and $\theta_2$, we have that

$$
\begin{aligned}
&\|\nabla_\theta \hat{v}(s,a;\theta_1,\vartheta^*(\theta_1)) - \nabla_\theta \hat{v}(s,a;\theta_2,\vartheta^*(\theta_2))\|_2 \\
&\leq \|\nabla_\theta \hat{v}(s,a;\theta_1,\vartheta^*(\theta_1)) - \nabla_\theta \hat{v}(s,a;\theta_1,\vartheta^*(\theta_2))\|_2 \\
&\quad + \|\nabla_\theta \hat{v}(s,a;\theta_1,\vartheta^*(\theta_2)) - \nabla_\theta \hat{v}(s,a;\theta_2,\vartheta^*(\theta_2))\|_2 \\
&\leq L_{\theta\vartheta}\|\vartheta^*(s;\theta_1) - \vartheta^*(s;\theta_2)\|_2 + L_{\theta\theta}\|\theta_1 - \theta_2\|_2 \\
&\leq \left(\frac{L_{\theta\vartheta}L_{\vartheta\theta}}{\mu} + L_{\theta\theta}\right)\|\theta_1 - \theta_2\|_2.
\end{aligned}
$$

The first inequality comes from the triangle inequality, the second inequality comes from Assumption C.1, and the last inequality comes from Lemma C.1. Therefore, the proof is concluded. $\quad\square$

**Lemma C.3** (Smoothness of Adversarial Value Function against Strongest Adversary). *Suppose assumptions C.1, C.2, C.3, and C.4 hold. Then we have that $V^{\pi_\theta \circ \nu^*(\pi_\theta)}(\mu_0)$ is $L$-smooth, i.e., for any $\theta_1$ and $\theta_2$, we have that*

$$
\left| V^{\pi_{\theta_1} \circ \nu^*(\pi_{\theta_1})}(\mu_0) - V^{\pi_{\theta_2} \circ \nu^*(\pi_{\theta_2})}(\mu_0) - \langle \nabla_\theta V^{\pi_{\theta_2} \circ \nu^*(\pi_{\theta_2})}(\mu_0), \theta_1 - \theta_2\rangle \right| \leq \frac{L}{2}\|\theta_1 - \theta_2\|_2^2,
$$

$$
\|\nabla_\theta V^{\pi_{\theta_1} \circ \nu^*(\pi_{\theta_1})}(\mu_0) - \nabla_\theta V^{\pi_{\theta_2} \circ \nu^*(\pi_{\theta_2})}(\mu_0)\|_2 \leq L\|\theta_1 - \theta_2\|_2,
$$

*where $L = \frac{L_{\theta\vartheta}L_{\vartheta\theta}}{\mu} + L_{\theta\theta} + M_{\hat{v}}\left(\frac{L_{d\vartheta}L_{\vartheta\theta}}{\mu} + L_{d\theta}\right)$.*

*Proof.* For any $\theta_1$ and $\theta_2$, we have

$$
\begin{aligned}
&\|\nabla_\theta V^{\pi_{\theta_1} \circ \nu^*(\pi_{\theta_1})}(\mu_0) - \nabla_\theta V^{\pi_{\theta_2} \circ \nu^*(\pi_{\theta_2})}(\mu_0)\|_2 \\
&= \left\| \mathbb{E}_{(s,a) \sim d_{\theta_1,\vartheta^*(\theta_1)}} [\nabla_\theta \hat{v}(s,a;\theta_1,\vartheta^*(\theta_1))] - \mathbb{E}_{(s,a) \sim d_{\theta_2,\vartheta^*(\theta_2)}} [\nabla_\theta \hat{v}(s,a;\theta_2,\vartheta^*(\theta_2))] \right\|_2 \\
&\leq \left\| \mathbb{E}_{(s,a) \sim d_{\theta_1,\vartheta^*(\theta_1)}} [\nabla_\theta \hat{v}(s,a;\theta_1,\vartheta^*(\theta_1)) - \nabla_\theta \hat{v}(s,a;\theta_2,\vartheta^*(\theta_2))] \right\|_2 \\
&\quad + \left\| \mathbb{E}_{(s,a) \sim (d_{\theta_1,\vartheta^*(\theta_1)} - d_{\theta_2,\vartheta^*(\theta_2)})} [\nabla_\theta \hat{v}(s,a;\theta_2,\vartheta^*(\theta_2))] \right\|_2.
\end{aligned}
$$

This inequality comes from the triangle inequality. For the first item, we have

$$\left\|\mathbb{E}_{(s,a)\sim d_{\theta_1,\vartheta^*(\theta_1)}}\left[\nabla_\theta\hat{v}(s,a;\theta_1,\vartheta^*(\theta_1)) - \nabla_\theta\hat{v}(s,a;\theta_2,\vartheta^*(\theta_2))\right]\right\|_2$$

$$\leq \mathbb{E}_{(s,a)\sim d_{\theta_1,\vartheta^*(\theta_1)}}\left[\left\|\nabla_\theta\hat{v}(s,a;\theta_1,\vartheta^*(\theta_1)) - \nabla_\theta\hat{v}(s,a;\theta_2,\vartheta^*(\theta_2))\right\|_2\right]$$

$$\leq \left(\frac{L_{\theta\vartheta}L_{\vartheta\theta}}{\mu} + L_{\theta\theta}\right)\|\theta_1 - \theta_2\|_2,$$

where the last inequality comes from Lemma C.2. For the second item, we have that

$$\left\|\mathbb{E}_{(s,a)\sim\left(d_{\theta_1,\vartheta^*(\theta_1)} - d_{\theta_2,\vartheta^*(\theta_2)}\right)}\left[\nabla_\theta\hat{v}(s,a;\theta_2,\vartheta^*(\theta_2))\right]\right\|_2$$

$$\leq \left\|\mathbb{E}_{(s,a)\sim\left(d_{\theta_1,\vartheta^*(\theta_1)} - d_{\theta_1,\vartheta^*(\theta_2)}\right)}\left[\nabla_\theta\hat{v}(s,a;\theta_2,\vartheta^*(\theta_2))\right]\right\|_2$$

$$+ \left\|\mathbb{E}_{(s,a)\sim\left(d_{\theta_1,\vartheta^*(\theta_2)} - d_{\theta_2,\vartheta^*(\theta_2)}\right)}\left[\nabla_\theta\hat{v}(s,a;\theta_2,\vartheta^*(\theta_2))\right]\right\|_2$$

$$\leq \|d_{\theta_1,\vartheta^*(\theta_1)} - d_{\theta_1,\vartheta^*(\theta_2)}\|_{\mathrm{TV}} \cdot \sup_{s,a}\|\nabla_\theta\hat{v}(s,a;\theta_2,\vartheta^*(\theta_2))\|$$

$$+ \|d_{\theta_1,\vartheta^*(\theta_2)} - d_{\theta_2,\vartheta^*(\theta_2)}\|_{\mathrm{TV}} \cdot \sup_{s,a}\|\nabla_\theta\hat{v}(s,a;\theta_2,\vartheta^*(\theta_2))\|$$

$$\leq L_{d\vartheta}M_{\hat{v}}\|\vartheta^*(\theta_1) - \vartheta^*(\theta_2)\|_2 + L_{d\theta}M_{\hat{v}}\|\theta_1 - \theta_2\|_2$$

$$\leq L_{d\vartheta}M_{\hat{v}}\frac{L_{\vartheta\theta}}{\mu}\|\theta_1 - \theta_2\|_2 + L_{d\theta}M_{\hat{v}}\|\theta_1 - \theta_2\|_2$$

$$= M_{\hat{v}}\left(\frac{L_{d\vartheta}L_{\vartheta\theta}}{\mu} + L_{d\theta}\right)\|\theta_1 - \theta_2\|_2.$$

The penultimate equation comes from assumptions C.2 and C.4. The last inequality comes from Lemma C.1. Thus, we have that

$$\|\nabla_\theta V^{\pi_{\theta_1}\circ\nu^*(\pi_{\theta_1})}(\mu_0) - \nabla_\theta V^{\pi_{\theta_2}\circ\nu^*(\pi_{\theta_2})}(\mu_0)\|_2$$

$$\leq \left(\frac{L_{\theta\vartheta}L_{\vartheta\theta}}{\mu} + L_{\theta\theta} + M_{\hat{v}}\left(\frac{L_{d\vartheta}L_{\vartheta\theta}}{\mu} + L_{d\theta}\right)\right)\|\theta_1 - \theta_2\|_2.$$

Therefore, the proof of this lemma is concluded. $\square$

**Lemma C.4** (Bounded Error of Approximate Stochastic Gradient). *Suppose assumptions C.1 and C.3 hold. Then we can control the error of the approximate stochastic gradient in the ARPO with $\delta$-approximation adversary in definition C.1 (Algorithm 1). Specifically, we have that*

$$\left\|\nabla_\theta\widehat{V^{\pi_\theta\circ\nu_\vartheta}}(\mu_0) - \nabla_\theta\widehat{V^{\pi_\theta\circ\nu^*(\pi_\theta)}}(\mu_0)\right\|_2 \leq L_{\theta\vartheta}\sqrt{\frac{\delta}{\mu}}.$$

*Proof.* We have that

$$\left\|\nabla_\theta\widehat{V^{\pi_\theta\circ\nu_\vartheta}}(\mu_0) - \nabla_\theta\widehat{V^{\pi_\theta\circ\nu^*(\pi_\theta)}}(\mu_0)\right\|_2$$

$$= \left\|\frac{1}{|\tau|}\sum_{(s,a)\in\tau}\left(\nabla_\theta\hat{v}(s,a;\theta,\vartheta) - \nabla_\theta\hat{v}(s,a;\theta,\vartheta^*(\theta))\right)\right\|_2$$

$$\leq \frac{1}{|\tau|}\sum_{(s,a)\in\tau}\|\nabla_\theta\hat{v}(s,a;\theta,\vartheta) - \nabla_\theta\hat{v}(s,a;\theta,\vartheta^*(\theta))\|_2$$

$$\leq \frac{1}{|\tau|}\sum_{(s,a)\in\tau}L_{\theta\vartheta}\|\vartheta(s) - \vartheta^*(s;\theta)\|_2.$$

The first inequality comes from the triangle inequality, and the last inequality comes from Assumption C.1. Under Assumption C.3, we have that

$$\hat{v}(s,a;\theta,\vartheta) \geq \hat{v}(s,a;\theta,\vartheta^*(\theta)) + \langle\nabla_\vartheta\hat{v}(s,a;\theta,\vartheta^*(\theta)), \vartheta(s) - \vartheta^*(s;\theta)\rangle + \frac{\mu}{2}\|\vartheta(s) - \vartheta^*(s;\theta)\|_2^2,$$

$$\hat{v}(s,a;\theta,\vartheta^*(\theta)) \geq \hat{v}(s,a;\theta,\vartheta) + \langle\nabla_\vartheta\hat{v}(s,a;\theta,\vartheta), \vartheta^*(s;\theta) - \vartheta(s)\rangle + \frac{\mu}{2}\|\vartheta(s) - \vartheta^*(s;\theta)\|_2^2.$$

Combining the above two inequalities, we have that

$$
\begin{aligned}
&\mu\|\vartheta(s) - \vartheta^*(s;\theta)\|_2^2 \\
&\leq \langle \nabla_\vartheta \hat{v}(s,a;\theta,\vartheta) - \nabla_\vartheta \hat{v}(s,a;\theta,\vartheta^*(\theta)), \vartheta(s) - \vartheta^*(s;\theta) \rangle \\
&\leq \delta - \langle \nabla_\vartheta \hat{v}(s,a;\theta,\vartheta^*(\theta)), \vartheta(s) - \vartheta^*(s;\theta) \rangle \\
&\leq \delta.
\end{aligned}
$$

The second inequality comes from that $\nu_\vartheta$ is a $\delta$-approximate maximizer of $\hat{v}(s,a;\theta,\vartheta)$, i.e.,

$$
\langle \nabla_\vartheta \hat{v}(s,a;\theta,\vartheta), \vartheta(s) - \vartheta^*(s;\theta) \rangle \leq \delta.
$$

The last inequality comes from the optimality of $\vartheta^*$, i.e.,

$$
\langle \nabla_\vartheta \hat{v}(s,a;\theta,\vartheta^*(\theta)), \vartheta(s) - \vartheta^*(s;\theta) \rangle \geq 0.
$$

Therefore, we have that

$$
\left\| \nabla_\theta \widehat{V^{\pi_\theta \circ \nu_\vartheta}}(\mu_0) - \nabla_\theta \widehat{V^{\pi_\theta \circ \nu^*}(\pi_\theta)}(\mu_0) \right\|_2 \leq L_{\theta\vartheta} \sqrt{\frac{\delta}{\mu}}.
$$

This completes the proof of this lemma. $\qquad\square$

### C.2.3 MAIN CONVERGENCE RESULTS

Based on these assumptions, we derive the main convergence property of ARPO.

**Theorem C.2** (Convergence of ARPO). *Denote* $\Delta := \max_\theta \min_\vartheta V^{\pi_\theta \circ \nu_\vartheta}(\mu_0) - \min_\vartheta V^{\pi_{\theta_0} \circ \nu_\vartheta}(\mu_0)$. *Under assumptions C.1, C.2, C.3, C.4, and C.5, set the step size* $\eta_k = \sqrt{\frac{\Delta}{\sigma^2 L K}}$. *Then, for the ARPO with $\delta$-approximation adversary in Definition C.1 (Algorithm 1) and $K \geq \frac{\Delta L}{\sigma^2}$, we have*

$$
\frac{1}{K} \sum_{k=0}^{K-1} \mathbb{E}\left[ \left\| \nabla_\theta V^{\pi_{\theta_k} \circ \nu^*(\pi_{\theta_k})}(\mu_0) \right\|_2^2 \right] \leq 4\sigma \sqrt{\frac{\Delta L}{K}} + \frac{2 L_{\theta\vartheta}^2 \delta}{\mu},
$$

*where* $L = \frac{L_{\theta\vartheta} L_{\vartheta\theta}}{\mu} + L_{\theta\theta} + M_{\hat{v}}\left( \frac{L_{d\vartheta} L_{\vartheta\theta}}{\mu} + L_{d\theta} \right)$.

*Proof of Theorem C.2.* For convenience, we denote the objective function $V(\theta) := V^{\pi_\theta \circ \nu^*(\pi_\theta)}(\mu_0)$, the accuracy stochastic gradient $G(\theta) := \nabla_\theta \widehat{V^{\pi_\theta \circ \nu^*}(\pi_\theta)}(\mu_0)$, and the approximation stochastic gradient $\hat{G}(\theta) := \nabla_\theta \widehat{V^{\pi_\theta \circ \nu_\vartheta}}(\mu_0)$. According to the gradient Lipschitz of $V^{\pi_\theta \circ \nu^*}(\pi_\theta)$ shown in Lemma C.3, we have that

$$
\begin{aligned}
&V(\theta_{k+1}) \\
&\geq V(\theta_k) + \langle \nabla_\theta V(\theta_k), \theta_{k+1} - \theta_k \rangle - \frac{L}{2}\|\theta_{k+1} - \theta_k\|_2^2 \\
&= V(\theta_k) + \eta_k \langle \nabla_\theta V(\theta_k), \hat{G}(\theta_k) \rangle - \frac{L\eta_k^2}{2}\|\hat{G}(\theta_k)\|_2^2 \\
&= V(\theta_k) + \eta_k \|\nabla_\theta V(\theta_k)\|_2^2 - \frac{L\eta_k^2}{2}\|\hat{G}(\theta_k)\|_2^2 + \eta_k \langle \nabla_\theta V(\theta_k), \hat{G}(\theta_k) - \nabla_\theta V(\theta_k) \rangle \\
&= V(\theta_k) + \eta_k \left(1 - \frac{L\eta_k}{2}\right) \|\nabla_\theta V(\theta_k)\|_2^2 \\
&\quad + \eta_k (1 - L\eta_k) \langle \nabla_\theta V(\theta_k), \hat{G}(\theta_k) - \nabla_\theta V(\theta_k) \rangle - \frac{L\eta_k^2}{2}\|\hat{G}(\theta_k) - \nabla_\theta V(\theta_k)\|_2^2 \\
&= V(\theta_k) + \eta_k \left(1 - \frac{L\eta_k}{2}\right) \|\nabla_\theta V(\theta_k)\|_2^2 + \eta_k (1 - L\eta_k) \langle \nabla_\theta V(\theta_k), \hat{G}(\theta_k) - G(\theta_k) \rangle \\
&\quad + \eta_k (1 - L\eta_k) \langle \nabla_\theta V(\theta_k), G(\theta_k) - \nabla_\theta V(\theta_k) \rangle \\
&\quad - \frac{L\eta_k^2}{2}\|\hat{G}(\theta_k) - G(\theta_k) + G(\theta_k) - \nabla_\theta V(\theta_k)\|_2^2.
\end{aligned}
$$

By Young's inequality, we have

$$\left| \langle \nabla_\theta V(\theta_k), \hat{G}(\theta_k) - G(\theta_k) \rangle \right| \leq \frac{1}{2} \left( \|\nabla_\theta V(\theta_k)\|_2^2 + \|\hat{G}(\theta_k) - G(\theta_k)\|_2^2 \right),$$

$$\|\hat{G}(\theta_k) - G(\theta_k) + G(\theta_k) - \nabla_\theta V(\theta_k)\|_2^2 \leq 2 \left( \|\hat{G}(\theta_k) - G(\theta_k)\|_2^2 + \|G(\theta_k) - \nabla_\theta V(\theta_k)\|_2^2 \right).$$

Therefore, we have that

$$
\begin{aligned}
&V(\theta_{k+1}) \\
&\geq V(\theta_k) + \eta_k \left( 1 - \frac{L\eta_k}{2} \right) \|\nabla_\theta V(\theta_k)\|_2^2 \\
&\quad - \frac{\eta_k}{2} (1 - L\eta_k) \left( \|\nabla_\theta V(\theta_k)\|_2^2 + \|\hat{G}(\theta_k) - G(\theta_k)\|_2^2 \right) \\
&\quad + \eta_k (1 - L\eta_k) \langle \nabla_\theta V(\theta_k), G(\theta_k) - \nabla_\theta V(\theta_k) \rangle \\
&\quad - L\eta_k^2 \left( \|\hat{G}(\theta_k) - G(\theta_k)\|_2^2 + \|G(\theta_k) - \nabla_\theta V(\theta_k)\|_2^2 \right). \\
&= V(\theta_k) + \frac{\eta_k}{2} \|\nabla_\theta V(\theta_k)\|_2^2 - \frac{\eta_k}{2} (1 + L\eta_k) \|\hat{G}(\theta_k) - G(\theta_k)\|_2^2 \\
&\quad + \eta_k (1 - L\eta_k) \langle \nabla_\theta V(\theta_k), G(\theta_k) - \nabla_\theta V(\theta_k) \rangle - L\eta_k^2 \|G(\theta_k) - \nabla_\theta V(\theta_k)\|_2^2.
\end{aligned}
$$

Taking the conditional expectation on both sides of the above inequality, we have that

$$
\begin{aligned}
&\mathbb{E}\left[ V(\theta_k) - V(\theta_{k+1}) \mid \theta_k \right] \\
&\leq -\frac{\eta_k}{2} \mathbb{E}\left[ \|\nabla_\theta V(\theta_k)\|_2^2 \mid \theta_k \right] + \frac{\eta_k}{2} (1 + L\eta_k) \mathbb{E}\left[ \|\hat{G}(\theta_k) - G(\theta_k)\|_2^2 \mid \theta_k \right] \\
&\quad + \eta_k (1 - L\eta_k) \mathbb{E}\left[ \langle \nabla_\theta V(\theta_k), \nabla_\theta V(\theta_k) - G(\theta_k) \rangle \mid \theta_k \right] \\
&\quad + L\eta_k^2 \mathbb{E}\left[ \|G(\theta_k) - \nabla_\theta V(\theta_k)\|_2^2 \mid \theta_k \right] \\
&= -\frac{\eta_k}{2} \mathbb{E}\left[ \|\nabla_\theta V(\theta_k)\|_2^2 \mid \theta_k \right] + \frac{\eta_k}{2} (1 + L\eta_k) \mathbb{E}\left[ \|\hat{G}(\theta_k) - G(\theta_k)\|_2^2 \mid \theta_k \right] \\
&\quad + L\eta_k^2 \mathbb{E}\left[ \|G(\theta_k) - \nabla_\theta V(\theta_k)\|_2^2 \mid \theta_k \right] \\
&\leq -\frac{\eta_k}{2} \mathbb{E}\left[ \|\nabla_\theta V(\theta_k)\|_2^2 \mid \theta_k \right] + \frac{\eta_k}{2} (1 + L\eta_k) \frac{L_{\theta\vartheta}^2 \delta}{\mu} \\
&\quad + L\eta_k^2 \mathbb{E}\left[ \|G(\theta_k) - \nabla_\theta V(\theta_k)\|_2^2 \mid \theta_k \right] \\
&\leq -\frac{\eta_k}{2} \mathbb{E}\left[ \|\nabla_\theta V(\theta_k)\|_2^2 \mid \theta_k \right] + \frac{\eta_k}{2} (1 + L\eta_k) \frac{L_{\theta\vartheta}^2 \delta}{\mu} + L\eta_k^2 \sigma^2.
\end{aligned}
$$

The first equation comes from the unbiased estimation $G(\theta)$, i.e., $\mathbb{E}[G(\theta)] = \nabla_\theta V(\theta)$. The second inequality comes from Lemma C.4, and the last inequality comes from Assumption C.5. Taking the telescope sum of the above inequality, we have that

$$\sum_{k=0}^{K-1} \frac{\eta_k}{2} \mathbb{E}\left[ \|\nabla_\theta V(\theta_k)\|_2^2 \right] \leq \mathbb{E}\left[ V(\theta_K) - V(\theta_0) \right] + \sum_{k=0}^{K-1} \frac{\eta_k}{2} (1 + L\eta_k) \frac{L_{\theta\vartheta}^2 \delta}{\mu} + \sum_{k=0}^{K-1} L\eta_k^2 \sigma^2.$$

Because $\eta_k = \sqrt{\frac{\Delta}{\sigma^2 L K}}$ and $K \geq \frac{\Delta L}{\sigma^2}$, we have that $L\eta_k \leq 1$. Furthermore, we have that

$$\frac{1}{K} \sum_{k=0}^{K-1} \mathbb{E}\left[ \|\nabla_\theta V(\theta_k)\|_2^2 \right] \leq 4\sigma \sqrt{\frac{\Delta L}{K}} + \frac{2L_{\theta\vartheta}^2 \delta}{\mu}.$$

Therefore, the proof of this theorem is concluded. □

## C.3 Learning Dynamics Near First-Order Stationary Policies

We explore the properties of FOSPs in ARPO and SPO by analyzing the learning dynamics in their neighborhoods. Our framework of analysis is drawn on the linear stability analysis of stochastic gradient descent (Wu et al., 2022b) and sharpness-aware minimization (Zhou et al., 2025).

### C.3.1 Learning Dynamics Near FOSPs in ARPO

We first consider the learning dynamics near FOSPs in ARPO. Let $(\theta^*, \vartheta^*)$ be a first-order stationary policy-adversary of $V^{\pi_\theta \circ \nu_\vartheta}(\mu_0)$. To simplify analysis, we consider $(\theta^*, \vartheta^*)$ satisfies the following second-order optimality conditions:

$$\nabla_\theta V^{\pi_{\theta^*} \circ \nu_{\vartheta^*}}(\mu_0) = 0, \ \nabla_\vartheta V^{\pi_{\theta^*} \circ \nu_{\vartheta^*}}(\mu_0) = 0, \ \nabla^2_{\theta\theta} V^{\pi_{\theta^*} \circ \nu_{\vartheta^*}}(\mu_0) \prec 0, \ \nabla^2_{\vartheta\vartheta} V^{\pi_{\theta^*} \circ \nu_{\vartheta^*}}(\mu_0) \succ 0. \tag{4}$$

Note that it makes no sense on the gradient when the objective adds a constant. Therefore, to explore the local properties around $(\theta^*, \vartheta^*)$, denote $\mathcal{L}(\theta, \vartheta) := V^{\pi_{\theta^*} \circ \nu_{\vartheta^*}}(\mu_0) - V^{\pi_\theta \circ \nu_\vartheta}(\mu_0)$. In the neighborhoods of $(\theta^*, \vartheta^*)$, we considering the local quadratic approximation of $\mathcal{L}(\theta, \vartheta)$ as:

$$\mathcal{L}(\theta, \vartheta) = \underbrace{\frac{1}{2}(\theta - \theta^*)^\top H_\theta(\theta^*, \vartheta^*)(\theta - \theta^*)}_{\mathcal{L}_\theta(\theta, \vartheta)} + \underbrace{\frac{1}{2}(\vartheta - \vartheta^*)^\top H_\vartheta(\theta^*, \vartheta^*)(\vartheta - \vartheta^*)}_{\mathcal{L}_\vartheta(\theta, \vartheta)}, \tag{5}$$

where $H_\theta(\theta^*, \vartheta^*) := \nabla^2_{\theta\theta} \mathcal{L}(\theta^*, \vartheta^*)$ and $H_\vartheta(\theta^*, \vartheta^*) := \nabla^2_{\vartheta\vartheta} \mathcal{L}(\theta^*, \vartheta^*)$ are the Hessians with respect to $\theta$ and $\vartheta$, respectively. Then, we have the following optimality conditions from (4).

$$\nabla_\theta \mathcal{L}(\theta^*, \vartheta^*) = 0, \ \nabla_\vartheta \mathcal{L}(\theta^*, \vartheta^*) = 0, \ \nabla^2_{\theta\theta} \mathcal{L}(\theta^*, \vartheta^*) \succ 0, \ \nabla^2_{\vartheta\vartheta} \mathcal{L}(\theta^*, \vartheta^*) \prec 0. \tag{6}$$

Let $\xi$ denote the stochastic variable of sampling and let the sampled estimation $f(s; \theta, \vartheta) := -\hat{v}(s, a; \theta, \vartheta)$ as defined in (3). Denote the sampled policy gradient estimation as

$$\nabla_\theta \hat{\mathcal{L}}_\xi(\theta, \vartheta) = \frac{1}{B} \sum_{i \in \xi} \nabla_\theta f(s_i; \theta, \vartheta), \ \nabla_\vartheta \hat{\mathcal{L}}_\xi(\theta, \vartheta) = \frac{1}{B} \sum_{i \in \xi} \nabla_\vartheta f(s_i; \theta, \vartheta).$$

Consider the following single-loop adversarially robust policy optimization iteration:

$$\begin{aligned} \vartheta_{k+1} &= \vartheta_k + \epsilon \nabla_\vartheta \hat{\mathcal{L}}_\xi(\theta_k, \vartheta_k), \\ \theta_{k+1} &= \theta_k - \eta \nabla_\theta \hat{\mathcal{L}}_\xi(\theta_k, \vartheta_{k+1}). \end{aligned} \qquad \text{(Single-Loop ARPO Iterator)}$$

**Policy Gradient Noise.** Denote the 1-batch gradient noise as $\xi_{\theta,i}(\theta, \vartheta) = \nabla_\theta f(s_i; \theta, \vartheta) - \nabla_\theta \mathcal{L}(\theta, \vartheta)$ and $\xi_{\vartheta,i}(\theta, \vartheta) = \nabla_\vartheta f(s_i; \theta, \vartheta) - \nabla_\vartheta \mathcal{L}(\theta, \vartheta)$. Let $\Sigma_\theta(\theta, \vartheta) := \mathbb{E}\left[\xi_{\theta,i}(\theta, \vartheta)\xi_{\theta,i}(\theta, \vartheta)^\top\right]$ and $\Sigma_\vartheta(\theta, \vartheta) := \mathbb{E}\left[\xi_{\vartheta,i}(\theta, \vartheta)\xi_{\vartheta,i}(\theta, \vartheta)^\top\right]$ as the 1-batch gradient noise covariance. Then, we have that

$$\Sigma_\theta(\theta, \vartheta) = \mathbb{E}\left[\nabla_\theta f(s_i; \theta, \vartheta)\nabla_\theta f(s_i; \theta, \vartheta)^\top\right] - \nabla_\theta \mathcal{L}(\theta, \vartheta)\nabla_\theta \mathcal{L}(\theta, \vartheta)^\top, \tag{7}$$

$$\Sigma_\vartheta(\theta, \vartheta) = \mathbb{E}\left[\nabla_\vartheta f(s_i; \theta, \vartheta)\nabla_\theta f(s_i; \theta, \vartheta)^\top\right] - \nabla_\vartheta \mathcal{L}(\theta, \vartheta)\nabla_\vartheta \mathcal{L}(\theta, \vartheta)^\top. \tag{8}$$

Furthermore, denote the sampled full-batch gradient noise as $\xi^B_\theta(\theta, \vartheta) := \nabla_\theta \hat{\mathcal{L}}_\xi(\theta, \vartheta) - \nabla_\theta \mathcal{L}(\theta, \vartheta) = \frac{1}{B} \sum_{i \in \xi} \xi_{\theta,i}(\theta, \vartheta)$ and $\xi^B_\vartheta(\theta, \vartheta) := \nabla_\vartheta \hat{\mathcal{L}}_\xi(\theta, \vartheta) - \nabla_\vartheta \mathcal{L}(\theta, \vartheta) = \frac{1}{B} \sum_{i \in \xi} \xi_{\vartheta,i}(\theta, \vartheta)$. Then, for the full-batch gradient noise, we have that

$$\mathbb{E}\left[\xi^B_\theta(\theta, \vartheta)\xi^B_\theta(\theta, \vartheta)^\top\right] = \frac{1}{B}\mathbb{E}\left[\xi_{\theta,i}(\theta, \vartheta)\xi_{\theta,i}(\theta, \vartheta)^\top\right] = \frac{1}{B}\Sigma_\theta(\theta, \vartheta), \tag{9}$$

$$\mathbb{E}\left[\xi^B_\vartheta(\theta, \vartheta)\xi^B_\vartheta(\theta, \vartheta)^\top\right] = \frac{1}{B}\mathbb{E}\left[\xi_{\vartheta,i}(\theta, \vartheta)\xi_{\vartheta,i}(\theta, \vartheta)^\top\right] = \frac{1}{B}\Sigma_\vartheta(\theta, \vartheta). \tag{10}$$

Based on these notations, we make a coercive assumption about gradient noise based on Hessians. Similar insights have been theoretically and empirically validated in a variety of studies about stochastic gradient descent in supervised learning settings (Zhu et al., 2019; Feng & Tu, 2021; Wu et al., 2020; 2022b; Mori et al., 2022; Wang & Wu, 2023; Wojtowytsch, 2024; Zhou et al., 2025).

**Assumption C.6** (Coercive Policy Gradient Noise). *There exists $\kappa > 0$, such that for all $\theta$ and $\vartheta$, the following coercive conditions hold:*

$$\text{Tr}\left(\frac{\Sigma_\theta(\theta, \vartheta)}{|\mathcal{L}(\theta, \vartheta)|} H_\theta(\theta, \vartheta)\right) \geq 2\kappa_\theta \|H_\theta(\theta, \vartheta)\|^2_F, \ \text{Tr}\left(\frac{\Sigma_\vartheta(\theta, \vartheta)}{|\mathcal{L}(\theta, \vartheta)|} H_\vartheta(\theta, \vartheta)\right) \geq -2\kappa_\vartheta \|H_\vartheta(\theta, \vartheta)\|^2_F.$$

**Local Stability of ARPO.** Similar to the linear stability of stochastic gradient descent (Wu et al., 2022b) and sharpness-aware minimization (Zhou et al., 2025), we define the local stability of ARPO. Note that in the practical optimization process, only local stable optima can be selected.

**Definition C.2** (Local Stability). *The first-order stationary policy-adversary* $(\theta^*, \vartheta^*)$ *is said to be locally stable if there exists a constant* $C > 0$ *such that for the Single-Loop ARPO Iterator, the local quadratic model* (5) $\mathcal{L}(\theta, \vartheta)$ *satisfies* $\mathbb{E}[\mathcal{L}(\theta_k, \vartheta_k)] \leq C\mathbb{E}[\mathcal{L}(\theta_0, \vartheta_0)], \ \forall k \geq 0.$

Based on these, we characterize the Hessian properties of FOSPs in ARPO.

**Theorem C.3** (Flatness Bound of FOSPs in ARPO). *Let* $(\theta^*, \vartheta^*)$ *be a first-order stationary policy-adversary in ARPO and satisfy the second-order optimality condition* (6). *Denote* $\lambda_{\min}(\cdot)$ *and* $\lambda_{\max}(\cdot)$ *as the minimum and maximum eigenvalues, and let* $d$ *be the state space dimension. If* $(\theta^*, \vartheta^*)$ *is locally stable,* $\mathcal{L}(\theta_0, \vartheta_0) > 0$, $\eta \geq 2/\lambda_{\min}(H_\theta(\theta^*, \vartheta^*))$, $\epsilon \leq 2/\lambda_{\max}(-H_\vartheta(\theta^*, \vartheta^*))$, $B \geq d\epsilon^2 \kappa_\vartheta \lambda_{\max}(-H_\vartheta(\theta^*, \vartheta^*))^2$, *and Assumption C.6 holds, then for single-loop ARPO, we have that*

$$\left( \|H_\theta(\theta^*, \vartheta^*)\|_F^2 + \frac{B}{\kappa_\theta \eta^2} \right) \left( 1 - \frac{\epsilon^2 \kappa_\vartheta \|H_\vartheta(\theta^*, \vartheta^*)\|_F^2}{B} \right) \leq \frac{B}{\kappa_\theta \eta^2}, \tag{11}$$

*where* $B$ *is the batch size,* $\kappa$ *is the coercive coefficient,* $\epsilon$ *is the attack budget, and* $\eta$ *is the step size.*

**Remark C.2.** $\mathcal{L}(\theta_0, \vartheta_0) > 0$ *means that* $V^{\pi_{\theta^*} \circ \nu_{\vartheta^*}}(\mu_0) > V^{\pi_{\theta_0} \circ \nu_{\vartheta_0}}(\mu_0)$, *which holds in practical training from the scratch.*

*Proof.* Because $\mathcal{L}(\theta_0, \vartheta_0) > 0$, by recursion, we can prove that $\mathcal{L}(\theta_{k-1}, \vartheta_k) > 0$ and $\mathcal{L}(\theta_k, \vartheta_k) > 0, \forall k \geq 1$. Then, we have that

$$\mathcal{L}(\theta_{k+1}, \vartheta_{k+1})$$
$$= \frac{1}{2} \underbrace{(\theta_{k+1} - \theta^*)^\top H_\theta(\theta^*, \vartheta^*)(\theta_{k+1} - \theta^*)}_{I} + \frac{1}{2} \underbrace{(\vartheta_{k+1} - \vartheta^*)^\top H_\vartheta(\theta^*, \vartheta^*)(\vartheta_{k+1} - \vartheta^*)}_{II}$$

Without loss of generality, we can let $\theta^* = \vartheta^* = 0$. Denote $H_1 := H_\theta(\theta^*, \vartheta^*)$, $H_2 := H_\vartheta(\theta^*, \vartheta^*)$. According to the definition of the sampled full-batch gradient noise, we have that $\xi_\theta^B(\theta, \vartheta) = \nabla_\theta \hat{\mathcal{L}}_\xi(\theta, \vartheta) - \nabla_\theta \mathcal{L}(\theta, \vartheta)$, and $\xi_\vartheta^B(\theta, \vartheta) = \nabla_\vartheta \hat{\mathcal{L}}_\xi(\theta, \vartheta) - \nabla_\vartheta \mathcal{L}(\theta, \vartheta)$.

Then, on one hand, we have that

$$\mathbb{E}[I]$$
$$= \mathbb{E}\left[ (\theta_k - \eta \nabla_\theta \hat{\mathcal{L}}_\xi(\theta_k, \vartheta_{k+1}))^\top H_1 (\theta_k - \eta \nabla_\theta \hat{\mathcal{L}}_\xi(\theta_k, \vartheta_{k+1})) \right]$$
$$= \mathbb{E}\left[ (\theta_k - \eta \nabla_\theta \mathcal{L}(\theta_k, \vartheta_{k+1}) - \eta \xi_\theta^B(\theta_k, \vartheta_{k+1}))^\top H_1 (\theta_k - \eta \nabla_\theta \mathcal{L}(\theta_k, \vartheta_{k+1}) - \eta \xi_\theta^B(\theta_k, \vartheta_{k+1})) \right]$$
$$= \mathbb{E}\left[ (\theta_k - \eta \nabla_\theta \mathcal{L}(\theta_k, \vartheta_{k+1}))^\top H_1 (\theta_k - \eta \nabla_\theta \mathcal{L}(\theta_k, \vartheta_{k+1})) \right]$$
$$\quad + \eta^2 \mathbb{E}\left[ \xi_\theta^B(\theta_k, \vartheta_{k+1})^\top H_1 \xi_\theta^B(\theta_k, \vartheta_{k+1}) \right]$$
$$= \mathbb{E}\left[ (\theta_k - \eta \nabla_\theta \mathcal{L}(\theta_k, \vartheta_{k+1}))^\top H_1 (\theta_k - \eta \nabla_\theta \mathcal{L}(\theta_k, \vartheta_{k+1})) \right] + \frac{\eta^2}{B} \mathbb{E}\left[ \text{Tr}\left( \Sigma_\theta(\theta_k, \vartheta_{k+1}) H_1 \right) \right]$$
$$\geq \mathbb{E}\left[ (\theta_k - \eta \nabla_\theta \mathcal{L}(\theta_k, \vartheta_{k+1}))^\top H_1 (\theta_k - \eta \nabla_\theta \mathcal{L}(\theta_k, \vartheta_{k+1})) \right] + \frac{2\eta^2 \kappa_\theta \|H_1\|_F^2}{B} \mathbb{E}\left[ \mathcal{L}(\theta_k, \vartheta_{k+1}) \right]$$
$$= \mathbb{E}\left[ \theta_k^\top (I - \eta H_1) H_1 (I - \eta H_1) \theta_k \right] + \frac{2\eta^2 \kappa_\theta \|H_1\|_F^2}{B} \mathbb{E}\left[ \mathcal{L}(\theta_k, \vartheta_{k+1}) \right]$$
$$\geq \mathbb{E}\left[ \theta_k^\top H_1 \theta_k \right] + \frac{2\eta^2 \kappa_\theta \|H_1\|_F^2}{B} \mathbb{E}\left[ \mathcal{L}(\theta_k, \vartheta_{k+1}) \right]$$
$$= 2\mathbb{E}\left[ \mathcal{L}_\theta(\theta_k, \vartheta_k) \right] + \frac{2\eta^2 \kappa_\theta \|H_1\|_F^2}{B} \mathbb{E}\left[ \mathcal{L}(\theta_k, \vartheta_{k+1}) \right].$$

The third equation comes from $\mathbb{E}\left[ \xi_\theta^B(\theta, \vartheta) \right] = 0$. The fourth equation comes from (9). The first inequality comes from Assumption C.6. Because of $\eta \geq \frac{2}{\lambda_{\min}(H_1)}$, we have that $H_1 \succeq \frac{2}{\eta} I$. This

comes that $H_1 - 2\eta H_1^2 + \eta^2 H_1^3 \succeq H_1$, which leads to the last inequality. Now, we need to estimate the lower bound of $\mathbb{E}\left[\mathcal{L}_\theta(\theta_k, \vartheta_{k+1})\right]$.

$$
\mathbb{E}\left[\mathcal{L}(\theta_k, \vartheta_{k+1})\right]
$$
$$
= \frac{1}{2}\mathbb{E}\left[\theta_k^\top H_1 \theta_k + \vartheta_{k+1}^\top H_2 \vartheta_{k+1}\right]
$$
$$
= \mathbb{E}\left[\mathcal{L}_\theta(\theta_k, \vartheta_k)\right] + \frac{1}{2}II.
$$

On the other hand, we have that

$$
\mathbb{E}\left[II\right]
$$
$$
= \mathbb{E}\left[(\vartheta_k + \epsilon\nabla_\vartheta\hat{\mathcal{L}}_\xi(\theta_k, \vartheta_k))^\top H_2(\vartheta_k + \epsilon\nabla_\vartheta\hat{\mathcal{L}}_\xi(\theta_k, \vartheta_k))\right]
$$
$$
= \mathbb{E}\left[(\vartheta_k + \epsilon\nabla_\vartheta\mathcal{L}(\theta_k, \vartheta_k) + \epsilon\xi_\vartheta^B(\theta_k, \vartheta_k))^\top H_2(\vartheta_k + \epsilon\nabla_\vartheta\mathcal{L}(\theta_k, \vartheta_k) + \epsilon\xi_\vartheta^B(\theta_k, \vartheta_k))\right]
$$
$$
= \mathbb{E}\left[(\vartheta_k + \epsilon\nabla_\vartheta\mathcal{L}(\theta_k, \vartheta_k))^\top H_2(\vartheta_k + \epsilon\nabla_\vartheta\mathcal{L}(\theta_k, \vartheta_k))\right] + \epsilon^2\mathbb{E}\left[\xi_\vartheta^B(\theta_k, \vartheta_k)^\top H_2 \xi_\vartheta^B(\theta_k, \vartheta_k)\right]
$$
$$
= \mathbb{E}\left[(\vartheta_k + \epsilon\nabla_\vartheta\mathcal{L}(\theta_k, \vartheta_k))^\top H_2(\vartheta_k + \epsilon\nabla_\vartheta\mathcal{L}(\theta_k, \vartheta_k))\right] + \frac{\epsilon^2}{B}\mathbb{E}\left[\mathrm{Tr}\left(\Sigma_\vartheta(\theta_k, \vartheta_k)H_2\right)\right]
$$
$$
\geq \mathbb{E}\left[(\vartheta_k + \epsilon\nabla_\vartheta\mathcal{L}(\theta_k, \vartheta_k))^\top H_2(\vartheta_k + \epsilon\nabla_\vartheta\mathcal{L}(\theta_k, \vartheta_k))\right] - \frac{2\epsilon^2\kappa_\vartheta\|H_2\|_F^2}{B}\mathbb{E}\left[\mathcal{L}(\theta_k, \vartheta_k)\right]
$$
$$
= \mathbb{E}\left[\vartheta_k^\top(I + \epsilon H_2)H_2(I + \epsilon H_2)\vartheta_k\right] - \frac{2\epsilon^2\kappa_\vartheta\|H_2\|_F^2}{B}\mathbb{E}\left[\mathcal{L}(\theta_k, \vartheta_k)\right]
$$
$$
\geq \mathbb{E}\left[\vartheta_k^\top H_2 \vartheta_k\right] - \frac{2\epsilon^2\kappa_\vartheta\|H_2\|_F^2}{B}\mathbb{E}\left[\mathcal{L}(\theta_k, \vartheta_k)\right]
$$
$$
= 2\mathbb{E}\left[\mathcal{L}_\vartheta(\theta_k, \vartheta_k)\right] - \frac{2\epsilon^2\kappa_\vartheta\|H_2\|_F^2}{B}\mathbb{E}\left[\mathcal{L}(\theta_k, \vartheta_k)\right].
$$

The third equation comes from $\mathbb{E}\left[\xi_\vartheta^B(\theta, \vartheta)\right] = 0$. The fourth equation comes from (10). The first inequality comes from Assumption C.6. Because of $\epsilon \leq \frac{2}{\lambda_{\max}(-H_2)}$, we have that $H_2 \succeq -\frac{2}{\epsilon}I$. This comes that $(I + \epsilon H_2)H_2(I + \epsilon H_2) = H_2 + 2\epsilon H_2^2 + \epsilon^2 H_2^3 \succeq H_2$, which leads to the last inequality. Therefore, we have that

$$
\mathbb{E}\left[\mathcal{L}(\theta_k, \vartheta_{k+1})\right] \geq \left(1 - \frac{\epsilon^2\kappa_\vartheta\|H_2\|_F^2}{B}\right)\mathbb{E}\left[\mathcal{L}(\theta_k, \vartheta_k)\right]. \tag{12}
$$

Because of $B \geq d\epsilon^2\kappa_\vartheta\lambda_{\max}(-H_2)^2$, we have that

$$
\|H_2\|_F^2 \leq d\lambda_{\max}(-H_2)^2 \leq \frac{B}{\epsilon^2\kappa_\vartheta},
$$

which indicates that $1 - \frac{\epsilon^2\kappa_\vartheta\|H_2\|_F^2}{B} > 0$.

Furthermore, we have that

$$
\mathbb{E}\left[\mathcal{L}(\theta_{k+1}, \vartheta_{k+1})\right] \geq \left(1 + \frac{\eta^2\kappa_\theta\|H_1\|_F^2}{B}\right)\left(1 - \frac{\epsilon^2\kappa_\vartheta\|H_2\|_F^2}{B}\right)\mathbb{E}\left[\mathcal{L}(\theta_k, \vartheta_k)\right]. \tag{13}
$$

Hence, $(\theta^*, \vartheta^*)$ is locally stable for stochastic ARPO, so the following inequality must hold:

$$
\left(1 + \frac{\eta^2\kappa_\theta\|H_1\|_F^2}{B}\right)\left(1 - \frac{\epsilon^2\kappa_\vartheta\|H_2\|_F^2}{B}\right) \leq 1. \tag{14}
$$

In conclusion, we have that for a local optimizer $(\theta^*, \vartheta^*)$ of $V^{\pi_\theta \circ \nu_\vartheta}(\mu_0)$, it holds that:

$$
\left(\|H_1\|_F^2 + \frac{B}{\kappa_\theta\eta^2}\right)\left(1 - \frac{\epsilon^2\kappa_\vartheta\|H_2\|_F^2}{B}\right) \leq \frac{B}{\kappa_\theta\eta^2}. \tag{15}
$$

This completes the proof. $\qquad\square$

From Theorem C.3, we observe that for FOSP in ARPO, if the adversary-side curvature is bounded as $\|\nabla^2_{\vartheta\vartheta} V^{\pi_{\theta^*} \circ \nu_{\vartheta^*}}(\mu_0)\|^2_F \leq B/(2\kappa_\vartheta \epsilon^2)$, then the policy-side curvature in ARPO matches that of SPO (as later shown in Theorem C.4), i.e., $\|\nabla^2_{\theta\theta} V^{\pi_{\theta^*} \circ \nu_{\vartheta^*}}(\mu_0)\|^2_F \leq B/(\kappa_\theta \eta^2)$. This implies that if ARPO achieves sufficiently strong robustness, i.e., flat adversarial curvature, it can preserve generalization comparable to SPO. Moreover, greater robustness may further enhance generalization. However, if robustness is insufficient, generalization may be significantly degraded.

### C.3.2 LEARNING DYNAMICS NEAR FOSPs IN SPO

Similarly, for an FOSP $\theta^*$ in SPO, we establish the learning dynamics analysis. Let $\theta^*$ be a first-order stationary policy-adversary of $V^{\pi_\theta}(\mu_0)$. To simplify analysis, we consider $\theta^*$ satisfies the following second-order optimality conditions:

$$\nabla_\theta V^{\pi_{\theta^*}}(\mu_0) = 0, \ \nabla^2_{\theta\theta} V^{\pi_{\theta^*}}(\mu_0) \prec 0. \tag{16}$$

Note that it makes no sense on the gradient when the objective adds a constant. Therefore, to explore the local properties around $\theta^*$, denote $\mathcal{L}(\theta) := V^{\pi_{\theta^*}}(\mu_0) - V^{\pi_\theta}(\mu_0)$. In the neighborhoods of $\theta^*$, we considering the local quadratic approximation of $\mathcal{L}(\theta)$ as:

$$\mathcal{L}(\theta) = \frac{1}{2}(\theta - \theta^*)^\top H_\theta(\theta^*)(\theta - \theta^*), \tag{17}$$

where $H_\theta(\theta^*) := \nabla^2_{\theta\theta} \mathcal{L}(\theta^*)$ is the Hessians with respect to $\theta$ respectively. Then, we have the following optimality conditions from (16).

$$\nabla_\theta \mathcal{L}(\theta^*) = 0, \ \nabla^2_{\theta\theta} \mathcal{L}(\theta^*) \succ 0. \tag{18}$$

For a sampling trajectory $\tau = (s_0, a_0, r_0, \dots) \sim \pi_\theta, \mathbb{P}$, denote the unbiased estimation of the action-value function $Q^{\pi_\theta}$ as

$$\widehat{Q^{\pi_\theta}}(s_t, a_t) := \sum_{t' \geq t} \gamma^{t'-t} r(s_{t'}, a_{t'}), \forall (s_t, a_t) \in \tau.$$

For simplicity of writing and reading of the latter, denote the sampled estimate as:

$$\hat{v}(s, a; \theta) := \frac{1}{1-\gamma} \text{sg}\left(\widehat{Q^{\pi_\theta}}(s, a)\right) \cdot \log \pi_\theta(a|s), \tag{19}$$

whereas $\text{sg}(\cdot)$ is the stop-gradient operator, meaning that $\widehat{Q^{\pi_\theta}}(s, a)$ is seen as a function according to $(s, a)$ and is detached from the gradient operation for $\theta$ in the following context.

Let $\xi$ denote the stochastic variable of sampling and let the sampled estimation $f(s; \theta) := -\hat{v}(s, a; \theta)$ as defined in (19). Denote the sampled policy gradient estimation as

$$\nabla_\theta \hat{\mathcal{L}}_\xi(\theta) = \frac{1}{B} \sum_{i \in \xi} \nabla_\theta f(s_i; \theta).$$

Consider the following SPO iteration:

$$\theta_{k+1} = \theta_k - \eta \nabla_\theta \hat{\mathcal{L}}_\xi(\theta_k). \tag{SPO Iterator}$$

**Policy Gradient Noise in SPO.** Denote the 1-batch gradient noise as $\xi_{\theta,i}(\theta) = \nabla_\theta f(s_i; \theta) - \nabla_\theta \mathcal{L}(\theta)$. Let $\Sigma_\theta(\theta) := \mathbb{E}\left[\xi_{\theta,i}(\theta)\xi_{\theta,i}(\theta)^\top\right]$ as the 1-batch gradient noise covariance. Then, we have that

$$\Sigma_\theta(\theta) = \mathbb{E}\left[\nabla_\theta f(s_i; \theta)\nabla_\theta f(s_i; \theta)^\top\right] - \nabla_\theta \mathcal{L}(\theta)\nabla_\theta \mathcal{L}(\theta)^\top, \tag{20}$$

Furthermore, denote the sampled full-batch gradient noise as $\xi_\theta^B(\theta) := \nabla_\theta \hat{\mathcal{L}}_\xi(\theta) - \nabla_\theta \mathcal{L}(\theta) = \frac{1}{B} \sum_{i \in \xi} \xi_{\theta,i}(\theta)$. Then, for the full-batch gradient noise, we have that

$$\mathbb{E}\left[\xi_\theta^B(\theta)\xi_\theta^B(\theta)^\top\right] = \frac{1}{B}\mathbb{E}\left[\xi_{\theta,i}(\theta)\xi_{\theta,i}(\theta)^\top\right] = \frac{1}{B}\Sigma_\theta(\theta). \tag{21}$$

Based on these notations, we make a coercive assumption about gradient noise based on Hessians. Similar insights have been theoretically and empirically validated in a variety of studies about stochastic gradient descent in supervised learning settings (Zhu et al., 2019; Feng & Tu, 2021; Wu et al., 2020; 2022b; Mori et al., 2022; Wang & Wu, 2023; Wojtowytsch, 2024; Zhou et al., 2025).

**Assumption C.7** (Coercive Policy Gradient Noise in SPO). *There exists $\kappa > 0$, such that for all $\theta$, the following coercive conditions hold:*

$$\mathrm{Tr}\left(\frac{\Sigma_\theta(\theta)}{|\mathcal{L}(\theta)|} H_\theta(\theta)\right) \geq 2\kappa_\theta \|H_\theta(\theta)\|_F^2.$$

**Local Stability of SPO.** Similar to the linear stability of stochastic gradient descent (Wu et al., 2022b) and sharpness-aware minimization (Zhou et al., 2025), we define the local stability of SPO. Note that in the practical optimization process, only local stable optima can be selected.

**Definition C.3** (Local Stability of SPO). *The first-order stationary policy $\theta^*$ is said to be locally stable if there exists a constant $C > 0$ such that for the SPO Iterator, the local quadratic model (17) $\mathcal{L}(\theta)$ satisfies $\mathbb{E}[\mathcal{L}(\theta_k)] \leq C\mathbb{E}[\mathcal{L}(\theta_0)], \ \forall k \geq 0.$*

Based on these, we characterize the Hessian properties of FOSPs in SPO.

**Theorem C.4** (Flatness Bound of FOSPs in SPO). *Let $\theta^*$ be a first-order stationary policy in SPO that satisfies the second-order optimality condition. If $\theta^*$ is locally stable, $\mathcal{L}(\theta_0) > 0$, and Assumption C.7 holds, then for SPO, we have that*

$$\|H_\theta(\theta^*)\|_F^2 \leq \frac{B}{\kappa\eta^2}. \tag{22}$$

*Proof.* Because $\mathcal{L}(\theta_0) > 0$, by recursion, we can prove that $\mathcal{L}(\theta_k) > 0, \forall k \geq 1$. Then, we have that

$$\mathcal{L}(\theta_{k+1}) = \frac{1}{2}(\theta_{k+1} - \theta^*)^\top H_\theta(\theta^*)(\theta_{k+1} - \theta^*).$$

Without loss of generality, we can let $\theta^* = 0$. Denote $H_1 := H_\theta(\theta^*)$. According to the definition of the sampled full-batch gradient noise, we have that $\xi_\theta^B(\theta) = \nabla_\theta \hat{\mathcal{L}}_\xi(\theta) - \nabla_\theta \mathcal{L}(\theta)$.

Then, we have that

$$\mathbb{E}\left[\theta_{k+1}^\top H_\theta(\theta^*)\theta_{k+1}\right]$$
$$= \mathbb{E}\left[(\theta_k - \eta\nabla_\theta\hat{\mathcal{L}}_\xi(\theta_k))^\top H_1(\theta_k - \eta\nabla_\theta\hat{\mathcal{L}}_\xi(\theta_k))\right]$$
$$= \mathbb{E}\left[(\theta_k - \eta\nabla_\theta\mathcal{L}(\theta_k) - \eta\xi_\theta^B(\theta_k))^\top H_1(\theta_k - \eta\nabla_\theta\mathcal{L}(\theta_k) - \eta\xi_\theta^B(\theta_k))\right]$$
$$= \mathbb{E}\left[(\theta_k - \eta\nabla_\theta\mathcal{L}(\theta_k))^\top H_1(\theta_k - \eta\nabla_\theta\mathcal{L}(\theta_k))\right] + \eta^2\mathbb{E}\left[\xi_\theta^B(\theta_k)^\top H_1\xi_\theta^B(\theta_k)\right]$$
$$= \mathbb{E}\left[(\theta_k - \eta\nabla_\theta\mathcal{L}(\theta_k))^\top H_1(\theta_k - \eta\nabla_\theta\mathcal{L}(\theta_k))\right] + \frac{\eta^2}{B}\mathbb{E}\left[\mathrm{Tr}\left(\Sigma_\theta(\theta_k)H_1\right)\right]$$
$$\geq \mathbb{E}\left[(\theta_k - \eta\nabla_\theta\mathcal{L}(\theta_k))^\top H_1(\theta_k - \eta\nabla_\theta\mathcal{L}(\theta_k))\right] + \frac{2\eta^2\kappa_\theta\|H_1\|_F^2}{B}\mathbb{E}\left[\mathcal{L}(\theta_k)\right]$$
$$= \mathbb{E}\left[\theta_k^\top(I - \eta H_1)H_1(I - \eta H_1)\theta_k\right] + \frac{2\eta^2\kappa_\theta\|H_1\|_F^2}{B}\mathbb{E}\left[\mathcal{L}(\theta_k)\right]$$
$$\geq \frac{2\eta^2\kappa_\theta\|H_1\|_F^2}{B}\mathbb{E}\left[\mathcal{L}(\theta_k)\right].$$

The third equation comes from $\mathbb{E}\left[\xi_\theta^B(\theta, \vartheta)\right] = 0$. The fourth equation comes from (21). The first inequality comes from Assumption C.7. Because of $H_1 \succeq 0$, we have that $H_1 - 2\eta H_1^2 + \eta^2 H_1^3 \succeq 0$, which leads to the last inequality.

Furthermore, we have that

$$\mathbb{E}\left[\mathcal{L}(\theta_{k+1})\right] \geq \frac{\eta^2\kappa_\theta\|H_1\|_F^2}{B}\mathbb{E}\left[\mathcal{L}(\theta_k)\right]. \tag{23}$$

Hence, $\theta^*$ is locally stable for stochastic SPO, so the following inequality must hold:

$$\frac{\eta^2\kappa_\theta\|H_1\|_F^2}{B} \leq 1. \tag{24}$$

In conclusion, we have that for a local optimizer $\theta^*$ of $V^{\pi_\theta}(\mu_0)$, it holds that:

$$\|H_1\|_F^2 \leq \frac{B}{\kappa\eta^2}. \tag{25}$$

This completes the proof. □

# D    ANALYSIS ON INTUITIVE ISA-MDPS

In this section, we illustrate and analyze detailed two-state and two-action ISA-MDPs to provide intuitive explanations from the numerical properties and their value space geometry. These examples are drawn on Dadashi et al. (2019).

**Lemma D.1** (Line Theorem (Dadashi et al., 2019)). *Let $s$ be a state and $\pi$ a policy. Then there exist two $s$-deterministic policies in $Y^\pi_{S\setminus\{s\}}$, denoted $\pi_\ell, \pi_u$, such that for all $\pi' \in Y^\pi_{S\setminus\{s\}}$,*

$$f_v(\pi_\ell) \preceq f_v(\pi') \preceq f_v(\pi_u),$$

*where $f_v(\pi) = V^\pi$, and $Y^\pi_{S\setminus\{s\}}$ describe the set of policies that agree with $\pi$ on all states except $s$.*

*Furthermore, the image of $f_v$ restricted to $Y^\pi_{S\setminus\{s\}}$ is a line segment, and the following three sets are equivalent:*

$$(i) f_v\left(Y^\pi_{S\setminus\{s\}}\right),$$

$$(ii)\{f_v\left(\alpha\pi_\ell + (1-\alpha)\pi_u\right) \mid \alpha \in [0,1]\},$$

$$(iii)\{\alpha f_v(\pi_\ell) + (1-\alpha)f_v(\pi_u) \mid \alpha \in [0,1]\}.$$

Based on the above Lemma, we derive the following Proposition.

**Proposition D.1.** *There exists an ISA-MDP that satisfies the following properties:*

*(1) Its robust policy space has a cut point, i.e., there exists a policy $\pi$ such that $\Pi_{Robust}/\{\pi\}$ is disconnected, where $\Pi_{Robust} = \{\pi \in \Pi | V^\pi(s) = V^{\pi \circ \nu^*(\pi)}(s), \forall s\}$.*

*(2) The FOSPs in ARPO are more than those in SPO. Moreover, let $\pi_S$ be a FOSP under SPO, and $\pi_A$ be a FOSP under ARPO. Then, $\pi_A$ is a robust policy with $V^{\pi_A}(\mu_0) - V^-(\mu_0) < \frac{1}{2}(V^{\pi_S}(\mu_0) - V^-(\mu_0))$, where $V^-(\mu_0) = \min_\pi V^\pi(\mu_0)$ is the worst value in ISA-MDP.*

*Proof.* Consider a two-state, two-action ISA-MDP $\mathcal{M} = (\mathcal{S}, \mathcal{A}, r, P, \gamma, B)$, with $\mathcal{S} = \{s_1, s_2\}$, $\mathcal{A} = \{a_1, a_2\}$, and reward and transition structure specified as

$$r(s_i, a_j) = \hat{r}[(i-1)|\mathcal{A}| + j - 1], \quad P(s_k|s_i, a_j) = \hat{P}[(i-1)|\mathcal{A}| + j - 1][k],$$

where

$$\hat{r} = [-0.45, -0.1, 0.5, 0.5], \quad \hat{P} = [[0.7, 0.3], [0.99, 0.01], [0.2, 0.8], [0.99, 0.01]].$$

Set the discount factor $\gamma = 0.9$, and let $B(s_1) = B(s_2) = \{s_1, s_2\}$. Parametrize policies as $\Theta = \{(\alpha, \beta) : 0 \le \alpha \le 1, 0 \le \beta \le 1\}$, with $\pi_{\alpha,\beta}(a_1|s_1) = \alpha$, $\pi_{\alpha,\beta}(a_1|s_2) = \beta$.

The robust policy parameter space is defined as

$$\Theta_{Robust} = \{(\alpha, \beta) \in \Theta : V^{\pi_{\alpha,\beta} \circ \nu^*(\pi_{\alpha,\beta})} = V^{\pi_{\alpha,\beta}}\}.$$

We derive that:

$$V^{\pi_{\alpha,\beta}}(s_1) = \frac{A_{\alpha,\beta}}{P_{\alpha,\beta}}, \tag{26}$$

$$V^{\pi_{\alpha,\beta}}(s_2) = \frac{B_{\alpha,\beta}}{P_{\alpha,\beta}}, \tag{27}$$

$$\frac{\partial V^{\pi_{\alpha,\beta}}(s_1)}{\partial \alpha} = \frac{0.24885\beta - 0.21635}{P_{\alpha,\beta}} - \frac{0.0261 A_{\alpha,\beta}}{P_{\alpha,\beta}^2}, \tag{28}$$

$$\frac{\partial V^{\pi_{\alpha,\beta}}(s_1)}{\partial \beta} = \frac{0.24885\alpha + 0.0711}{P_{\alpha,\beta}} + \frac{0.0711 A_{\alpha,\beta}}{P_{\alpha,\beta}^2}, \tag{29}$$

$$\frac{\partial V^{\pi_{\alpha,\beta}}(s_2)}{\partial \alpha} = \frac{0.24885\beta - 0.18135}{P_{\alpha,\beta}} - \frac{0.0261 B_{\alpha,\beta}}{P_{\alpha,\beta}^2}, \tag{30}$$

$$\frac{\partial V^{\pi_{\alpha,\beta}}(s_2)}{\partial \beta} = \frac{0.24885\alpha + 0.0711}{P_{\alpha,\beta}} + \frac{0.0711 B_{\alpha,\beta}}{P_{\alpha,\beta}^2}, \tag{31}$$

where

$$P_{\alpha,\beta} = |I - \gamma P^{\pi_{\alpha,\beta}}|$$
$$= (0.991 - 0.711\beta)(0.261\alpha + 0.109) + (0.261\alpha + 0.009)(0.711\beta - 0.891),$$
$$A_{\alpha,\beta} = 0.1305\alpha + (0.35\alpha + 0.1)(0.711\beta - 0.991) + 0.0045,$$
$$B_{\alpha,\beta} = 0.1305\alpha + (0.35\alpha + 0.1)(0.711\beta - 0.891) + 0.0545.$$

Direct calculation from (26) and (27) shows that the policy $\pi_{1,1}$ achieves the highest value, i.e., $\pi_{1,1}$ is globally optimal. Moreover, the following ordering holds:

$$V^{\pi_{1,1}} \approx (0.16, 1.89)^\top \succeq V^{\pi_{0,1}} \approx (-0.81, 1.26)^\top$$
$$\succeq V^{\pi_{0,0}} \approx (-0.95, -0.35)^\top \succeq V^{\pi_{1,0}} \approx (-2.47, -1.71)^\top. \tag{32}$$

To further analyze value relations, define

$$f(\beta, s) := V^{\pi_{1,\beta}}(s) - V^{\pi_{0,\beta}}(s), \qquad s \in \{s_1, s_2\}.$$

Explicit calculation shows there exists a threshold $\tilde{\beta} \approx 0.777$ such that

$$\begin{cases} f(\beta, s) > 0, & \text{if } \beta > \tilde{\beta}, \\ f(\beta, s) = 0, & \text{if } \beta = \tilde{\beta}, \\ f(\beta, s) < 0, & \text{if } \beta < \tilde{\beta}, \end{cases} \tag{33}$$

and this sign structure holds identically for both $s_1$ and $s_2$.

Applying the Line Theorem D.1, for any $0 \le \alpha_1 < \alpha_2 \le 1$ and for each fixed $s$,

$$\begin{cases} V^{\pi_{\alpha_1,\beta}}(s) < V^{\pi_{\alpha_2,\beta}}(s), & \text{if } \beta > \tilde{\beta}, \\ V^{\pi_{\alpha_1,\beta}}(s) = V^{\pi_{\alpha_2,\beta}}(s), & \text{if } \beta = \tilde{\beta}, \\ V^{\pi_{\alpha_1,\beta}}(s) > V^{\pi_{\alpha_2,\beta}}(s), & \text{if } \beta < \tilde{\beta}, \end{cases} \tag{34}$$

again for both $s_1$ and $s_2$.

Now, for value improvement in the $\beta$ direction, define

$$g(\alpha, s) := V^{\pi_{\alpha,1}}(s) - V^{\pi_{\alpha,0}}(s).$$

Explicit computation confirms that $g(\alpha, s) \ge 0$ for any $0 \le \alpha \le 1$ and both $s_1, s_2$. Thus, for any $0 \le \beta_1 < \beta_2 \le 1$ and fixed $\alpha$,

$$V^{\pi_{\alpha,\beta_1}}(s) < V^{\pi_{\alpha,\beta_2}}(s). \tag{35}$$

Let us now determine the structure of the robust policy region. By definition, for any $(\alpha, \beta) \in \Theta$, $\pi_{\alpha,\beta} \circ \nu^*(\pi_{\alpha,\beta}) = \arg\min_{\pi \in \Pi_{\alpha,\beta}} V^\pi$, where $\Pi_{\alpha,\beta} = \{\pi_{\alpha,\alpha}, \pi_{\alpha,\beta}, \pi_{\beta,\alpha}, \pi_{\beta,\beta}\}$.

If $\alpha = \beta$, then $\Pi_{\alpha,\alpha} = \{\pi_{\alpha,\alpha}\}$, so $(\alpha, \beta)$ is always robust. If $\beta \le \min\{\alpha, \tilde{\beta}\}$: Using (34) for $\beta \le \alpha$ gives $V^{\pi_{\alpha,\beta}}(s) \le V^{\pi_{\alpha,\alpha}}(s)$, and $V^{\pi_{\beta,\alpha}}(s) \ge V^{\pi_{\beta,\beta}}(s)$. Together with (35) and $\beta \le \tilde{\beta}$, $V^{\pi_{\alpha,\beta}}(s) \le V^{\pi_{\beta,\beta}}(s)$. Therefore, $V^{\pi_{\alpha,\beta}}(s)$ is the minimal value in $\Pi_{\alpha,\beta}$ for both states, so robustness holds.

For other regions $(\alpha, \beta) \notin \{\alpha = \beta\} \cup \{\beta \le \min\{\alpha, \tilde{\beta}\}\}$, by similarly using (34) and (35), one can verify that the minimal value in $\Pi_{\alpha,\beta}$ is always attained at a different policy (specifically, at $\pi_{\beta,\alpha}$, $\pi_{\alpha,\alpha}$, or $\pi_{\beta,\beta}$ in the respective regions). Thus, the robust region is exactly the union

$$\Theta_{Robust} = \{(\alpha, \beta) : \alpha = \beta\} \cup \{(\alpha, \beta) : \beta \le \min\{\alpha, \tilde{\beta}\}\}.$$

Notably, the point $(\tilde{\beta}, \tilde{\beta})$ serves as a cut point: removing it disconnects $\Theta_{Robust}$. This completes the proof of property (1).

For property (2), we show that, unlike SPO, the ARPO objective can admit more FOSP. Indeed, we focus on

$$\max_{\alpha,\beta} V^{\pi_{\alpha,\beta}}(s) \qquad \text{subject to: } \begin{cases} \alpha - \beta \ge 0, \\ \beta \ge 0, \\ \tilde{\beta} - \beta \ge 0, \\ 1 - \alpha \ge 0. \end{cases}$$

We analyze the Karush-Kuhn-Tucker (KKT) conditions at $(\alpha, \beta) = (0, 0)$ to verify FOSP. The Lagrangian is defined as:

$$L(s, \alpha, \beta, \lambda) = V^{\pi_{\alpha,\beta}}(s) + \lambda_1(\alpha - \beta) + \lambda_2\beta + \lambda_3(\tilde{\beta} - \beta) + \lambda_4(1 - \alpha).$$

The KKT conditions at $(\alpha, \beta) = (0, 0)$ are:

Stationarity:

$$\frac{\partial V^{\pi_{\alpha,\beta}}(s)}{\partial \alpha}\big|_{(0,0)} + \lambda_1 - \lambda_4 = 0,$$

$$\frac{\partial V^{\pi_{\alpha,\beta}}(s)}{\partial \beta}\big|_{(0,0)} - \lambda_1 + \lambda_2 - \lambda_3 = 0.$$

Primal feasibility: $(0, 0)$ satisfies all inequality constraints.

Dual feasibility: $\lambda_j \geq 0, \quad j = 1, 2, 3, 4,$

Complementary slackness:

$$\lambda_1(\alpha - \beta) = 0, \ \lambda_2\beta = 0, \ \lambda_3(\tilde{\beta} - \beta) = \lambda_3\tilde{\beta} = 0, \ \lambda_4(1 - \alpha) = 0.$$

Since at $(\alpha, \beta) = (0, 0)$, the gradients $\frac{\partial V^{\pi_{\alpha,\beta}}(s)}{\partial \alpha}\big|_{(0,0)}$ and $\frac{\partial V^{\pi_{\alpha,\beta}}(s)}{\partial \beta}\big|_{(0,0)}$ both exist, and by the structure of the problem the corresponding dual variables can be chosen to satisfy the equations above, all KKT conditions are satisfied. Therefore, $(0, 0)$ is a FOSP for ARPO, even though it is not a FOSP for SPO.

Let $\pi_S = \pi_{1,1}, \pi_A = \pi_{0,0}, \mu_0$ is the uniform distribution on $\mathcal{S}$. In this example, $\pi^- = \pi_{1,0}$. By (32), we know that

$$V^{\pi_S}(\mu_0) \approx 1.03, \ V^{\pi_A}(\mu_0) = -0.65, \ V^-(\mu_0) = -2.09.$$

Then, we have that

$$V^{\pi_S}(\mu_0) - V^-(\mu_0) = 3.12, \ V^{\pi_A}(\mu_0) - V^-(\mu_0) = 1.44 < \frac{1}{2}(V^{\pi_S}(\mu_0) - V^-(\mu_0)).$$

This completes the proof. $\qquad\square$

# E ANALYSIS ON SURROGATE ADVERSARY

In this section, we provide the analysis and proof about the surrogate adversary.

**Theorem E.1** (Surrogate Adversary). *For any policy $\pi_\theta$, state $s \in \mathcal{S}$, and $K > 0$, let the adversary $\vartheta$ be computed via $K$-step gradient descent on the adversarial value function, i.e., $\vartheta_{s,k} = \vartheta_{s,k-1} - \eta_{s,k-1}\nabla_{\vartheta_s}V^{\pi_\theta \circ \nu_\vartheta}(s)\big|_{\vartheta_s=\vartheta_{s,k-1}}, 1 \leq k \leq K$. Assume step sizes $\eta_{s,k} \leq 1/(\lambda_s + \delta)$ for some $\delta > 0$, and if $G(s; \pi, \nu) := \mathcal{D}_{\mathrm{KL}}(\pi(\cdot|s)\|(\pi \circ \nu)(\cdot|s)) \ll 1$. Then, we have*

$$V^{\pi_\theta}(s) - V^{\pi_\theta \circ \nu_\vartheta}(s) \geq \frac{2\delta}{\lambda_{\max}(F_{s,\theta})K}G(s; \pi_\theta, \nu_\vartheta) + O\left(G(s; \pi_\theta, \nu_\vartheta)^{\frac{3}{2}}\right),$$

*where $F_{s,\theta}$ is the Fisher information matrix of $\log(\pi_\theta \circ \nu_\vartheta)(a|s)$ with respect to $\vartheta_s = 0$, defined as $F_{s,\theta} := F(\theta, s) = \mathbb{E}_{a \sim \pi_\theta(\cdot|s)}\left[(\nabla_{\vartheta_s}\log\pi_\theta(a|s + \vartheta_s)\big|_{\vartheta_s=0})(\nabla_{\vartheta_s}\log\pi_\theta(a|s + \vartheta_s)\big|_{\vartheta_s=0})^T\right], \lambda_s = \max_{s+\vartheta_s \in B(s)}\lambda_{\max}(\nabla^2_{\vartheta\vartheta}V^{\pi_\theta \circ \nu_\vartheta}(s)\big|_{\vartheta=\vartheta_s})$ and $\lambda_{\max}(\cdot)$ is the largest eigenvalue of matrix.*

*Proof.* We begin by examining the local behavior of the KL-divergence between the policy at the original and perturbed state:

$$G(s; \pi_\theta, \nu_\vartheta) := D_{\mathrm{KL}}\left(\pi_\theta(\cdot|s)\|\pi_\theta(\cdot|s + \vartheta_s)\right).$$

Let us expand $G(s; \pi_\theta, \nu_\vartheta)$ for small $\vartheta_s$:

$$G(s; \pi_\theta, \nu_\vartheta) = \sum_a \pi_\theta(a|s)\log\frac{\pi_\theta(a|s)}{\pi_\theta(a|s + \vartheta_s)}$$

$$= \mathbb{E}_{a \sim \pi_\theta(\cdot|s)}\left[\log\pi_\theta(a|s) - \log\pi_\theta(a|s + \vartheta_s)\right].$$

Now, expand $\log \pi_\theta(a|s + \vartheta_s)$ at $\vartheta_s = 0$ using a Taylor expansion:

$$\log \pi_\theta(a|s+\vartheta_s) = \log \pi_\theta(a|s) + \nabla_{\vartheta_s} \log \pi_\theta(a|s)|_{\vartheta_s=0}^\top \vartheta_s + \frac{1}{2} \vartheta_s^\top \nabla_{\vartheta_s}^2 \log \pi_\theta(a|s)|_{\vartheta_s=0} \vartheta_s + O(\|\vartheta_s\|_2^3).$$

Plugging this into the difference, $\log \pi_\theta(a|s) - \log \pi_\theta(a|s + \vartheta_s)$, and taking the expectation over $a \sim \pi_\theta(\cdot|s)$:

$$\mathbb{E}_{a\sim\pi_\theta(\cdot|s)} \left[ -\nabla_{\vartheta_s} \log \pi_\theta(a|s)|_{\vartheta_s=0}^\top \vartheta_s - \frac{1}{2} \vartheta_s^\top \nabla_{\vartheta_s}^2 \log \pi_\theta(a|s)|_{\vartheta_s=0} \vartheta_s \right] + O(\|\vartheta_s\|_2^3).$$

The first term equals zero because

$$\mathbb{E}_{a\sim\pi_\theta(\cdot|s)}[\nabla_{\vartheta_s} \log \pi_\theta(a|s)|_{\vartheta_s=0}] = \nabla_{\vartheta_s} \sum_a \pi_\theta(a|s+\vartheta_s)|_{\vartheta_s=0} = 0,$$

since the sum of probabilities is always 1.

The second term can also be separated using Fisher information matrix properties:

$$F_{s,\theta} := \mathbb{E}_{a\sim\pi_\theta(\cdot|s)}\left[\nabla_{\vartheta_s} \log \pi_\theta(a|s)\nabla_{\vartheta_s} \log \pi_\theta(a|s)^\top\right],$$

and

$$\mathbb{E}_{a\sim\pi_\theta(\cdot|s)}[\nabla_{\vartheta_s}^2 \log \pi_\theta(a|s)] = -F_{s,\theta}.$$

Therefore, combining the above,

$$G(s; \pi_\theta, \nu_\vartheta) = \frac{1}{2}\vartheta_s^\top F_{s,\theta}\vartheta_s + O(\|\vartheta_s\|_2^3).$$

From the expansion above, since $F_{s,\theta}$ is positive definite (or at least positive semi-definite), we have:

$$\frac{1}{2}\lambda_{\min}(F_{s,\theta})\|\vartheta_s\|_2^2 \le \frac{1}{2}\vartheta_s^\top F_{s,\theta}\vartheta_s \le \frac{1}{2}\lambda_{\max}(F_{s,\theta})\|\vartheta_s\|_2^2.$$

Neglecting the lower bound (since we only need an upper bound for $\|\vartheta_s\|_2^2$), isolate $\|\vartheta_s\|_2^2$:

$$\|\vartheta_s\|_2^2 \le \frac{2}{\lambda_{\max}(F_{s,\theta})} G(s; \pi_\theta, \nu_\vartheta) + O(G(s; \pi_\theta, \nu_\vartheta)^{3/2}).$$

Here, the $O(G^{3/2})$ term comes from inverting the Taylor expansion when $\vartheta_s$ is small. We now analyze the decrease in the value function resulting from $K$ steps of gradient descent:

For each step,
$$\vartheta_{s,k} = \vartheta_{s,k-1} - \eta_{s,k-1}\nabla_{\vartheta_s} V^{\pi_\theta \circ \nu_\vartheta}(s)|_{\vartheta_s=\vartheta_{s,k-1}}.$$
Let $V_k = V^{\pi_\theta \circ \nu_{\vartheta_{s,k}}}(s)$. Expanding $V_k$ around $\vartheta_{s,k-1}$ using Taylor's theorem:

$$V_k = V_{k-1} + \nabla_{\vartheta_s} V^{\pi_\theta \circ \nu_\vartheta}(s)|_{\vartheta_s=\vartheta_{s,k-1}}^\top (\vartheta_{s,k} - \vartheta_{s,k-1})$$
$$+ \frac{1}{2}(\vartheta_{s,k} - \vartheta_{s,k-1})^\top \nabla_{\vartheta_s}^2 V^{\pi_\theta \circ \nu_\vartheta}(s)|_{\vartheta_s=\vartheta_{s,k-1}}(\vartheta_{s,k} - \vartheta_{s,k-1}) + O(\|\vartheta_{s,k} - \vartheta_{s,k-1}\|_2^3).$$

Summing over $k = 1$ to $K$ and noting telescoping sum of $V_{k-1} - V_k$:

$$V^{\pi_\theta}(s) - V^{\pi_\theta \circ \nu_{\vartheta_s,K}}(s) = \sum_{k=1}^K (V^{\pi_\theta \circ \nu_{\vartheta_{s,k-1}}}(s) - V^{\pi_\theta \circ \nu_{\vartheta_{s,k}}}(s))$$

$$+ O\left(\sum_{k=1}^K \|\vartheta_{s,k} - \vartheta_{s,k-1}\|_2^3\right)$$

$$= -\sum_{k=1}^K \nabla_{\vartheta_s} V^{\pi_\theta \circ \nu_\vartheta}(s)|_{\vartheta_s=\vartheta_{s,k-1}}^\top (\vartheta_{s,k} - \vartheta_{s,k-1})$$

$$- \frac{1}{2}\sum_{k=1}^K (\vartheta_{s,k} - \vartheta_{s,k-1})^\top \nabla_{\vartheta_s}^2 V^{\pi_\theta \circ \nu_\vartheta}(s)|_{\vartheta_s=\vartheta_{s,k-1}}(\vartheta_{s,k} - \vartheta_{s,k-1})$$

$$+ O\left(\sum_{k=1}^K \|\vartheta_{s,k} - \vartheta_{s,k-1}\|_2^3\right).$$

The gradient update yields:

$$\vartheta_{s,k} - \vartheta_{s,k-1} = -\eta_{s,k-1}\nabla_{\vartheta_s}V^{\pi_\theta\circ\nu_\vartheta}(s)|_{\vartheta_s=\vartheta_{s,k-1}}.$$

For small enough steps, and since the step size $\eta_{s,k-1} \leq 1/(\lambda_s + \delta)$ ensures sufficient decrease (similar to the classic descent lemma in optimization), we can bound the value difference as:

$$V^{\pi_\theta}(s) - V^{\pi_\theta\circ\nu_{\vartheta_{s,K}}}(s) \geq \delta\sum_{k=1}^{K}\|\vartheta_{s,k} - \vartheta_{s,k-1}\|_2^2 + O\left(\sum_{k=1}^{K}\|\vartheta_{s,k} - \vartheta_{s,k-1}\|_2^3\right).$$

By Jensen's inequality,

$$\sum_{k=1}^{K}\|\vartheta_{s,k} - \vartheta_{s,k-1}\|_2^2 \geq \frac{1}{K}\left\|\sum_{k=1}^{K}(\vartheta_{s,k} - \vartheta_{s,k-1})\right\|_2^2 = \frac{1}{K}\|\vartheta_{s,K}\|_2^2.$$

Now substitute the earlier quadratic relationship between $\|\vartheta_{s,K}\|^2$ and $G(s;\pi_\theta,\nu_{\vartheta_{s,K}})$:

$$V^{\pi_\theta}(s) - V^{\pi_\theta\circ\nu_{\vartheta_{s,K}}}(s) \geq \frac{\delta}{K}\|\vartheta_{s,K}\|_2^2 + O(\|\vartheta_{s,K}\|_2^3)$$

$$\geq \frac{2\delta}{\lambda_{\max}(F_{s,\theta})K}G(s;\pi_\theta,\nu_{\vartheta_{s,K}}) + O(G(s;\pi_\theta,\nu_{\vartheta_{s,K}})^{3/2}).$$

This completes the proof. $\qquad\square$

## F  ADDITIONAL ALGORITHM DETAILS

### F.1  ADVERSARIALLY ROBUST PROXIMAL POLICY OPTIMIZATION (AR-PPO)

We apply the ARPO paradigm on top of the PPO algorithm and further incorporate the SPO-based guidance, yielding Adversarily Robust Proximal Policy Optimization (AR-PPO). The overall algorithm is presented in Algorithm 2.

### F.2  BILEVEL ADVERSARIALLY ROBUST PROXIMAL POLICY OPTIMIZATION (BAR-PPO)

We apply the BARPO paradigm on top of the PPO algorithm and further incorporate the SPO-based guidance, yielding the Bilevel Adversarially Robust Proximal Policy Optimization (BAR-PPO). The overall algorithm is presented in Algorithm 3.

### F.3  ADDITIONAL IMPLEMENTATION DETAILS

We run 2048 simulation steps per iteration and use a simple MLP network for all agents. SA-PPO, WocaR-PPO, and BAR-PPO are trained for 2 million steps (976 iterations) on Hopper, Walker2d, and Halfcheetah, and 10 million steps (4882 iterations) on Ant for convergence. RADIAL-PPO is trained for 4 million steps (2000 iterations) on Hopper, Walker2d, and Halfcheetah following the official implementation and 10 million steps (4882 iterations) on Ant. The attack budget $\epsilon$ is linearly increased from 0 to the target value during the first 732 iterations on Hopper, Walker2d, and Halfcheetah, and 3662 iterations on Ant, before continuing with the target value for the remaining iterations. This scheduler is aligned with Zhang et al. (2020c); Liang et al. (2022) without extra tuning. The regularization weight $\kappa$ is chosen from $\{0.1, 0.3, 1.0\}$ for BAR-PPO. We run 10 iterations with step size $\epsilon/10$ for both methods and set the temperature parameter $\beta = 1 \times 10^{-5}$ for SGLD.

We implement all PPO agents with the same fully connected (MLP) structure as Zhang et al. (2020c); Oikarinen et al. (2021). In the Ant environment, we choose the best regularization parameter $\kappa$ in $\{0.01, 0.03, 0.1, 0.3, 1.0\}$ for SA-PPO, RADIAL-PPO, and WocaR-PPO to achieve better robustness. For fair and comparable agent selection, we conduct multiple experiments for each setup, repeating each 17 times to account for the inherent performance variability in RL. After training, we attack all agents using random, critic, MAD, and RS attacks. Then, we select the median agent by considering the natural and robust returns as our final agent. This chosen agent is then attacked using the SA-RL and PA-AD attacks to further robustness evaluation, because these attacks involve quite high computational costs.

## F.4 EXPERIMENTS COMPUTE RESOURCES

All MuJoCo experiments were conducted on a machine with an AMD EPYC 7742 CPU and no GPU acceleration.

BARPO's training takes approximately 2 hours for Hopper, Walker2d, and HalfCheetah, and 9 hours for the more complex Ant environment. This represents roughly a 2x increase in wall-clock time compared to standard, non-robust PPO training. Crucially, this overhead is on par with other leading robust baselines (Zhang et al., 2020c; Oikarinen et al., 2021; Liang et al., 2022), which require similar inner-loop computations to find the adversary. The choice of a CPU-only setup follows baselines, as the wall-clock time in these MuJoCo tasks is overwhelmingly dominated by the environment simulation overhead, ensuring a fair comparison of the algorithmic sample complexity.

Furthermore, we confirm that BARPO introduces no significant additional memory overhead compared to standard PPO training.

---

**Algorithm 2** Adversarially Robust Proximal Policy Optimization (AR-PPO).

**Input:** Number of iterations $T$, a schedule $\epsilon_t$ for the perturbation radius $\epsilon$, robustness weighting $\kappa$.

1: Initialize policy network $\pi_\theta(a|s)$ and value network $V_{\theta_V}(s)$.
2: **for** $t = 1$ **to** $T$ **do**
3:     Run $\pi_{\theta_t}$ in the environment to collect a set of trajectories $\mathcal{D} = \{\tau_k\}$ containing $|\mathcal{D}|$ episodes, each $\tau_k$ is a trajectory containing $|\tau_k|$ samples, $\tau_k := \{(s_{k,i}, a_{k,i}, r_{k,i}, s_{k,i+1})\}, i \in [|\tau_k|]$.
4:     Compute reward-to-go $\hat{R}_{k,i}$ for each step $i$ in every episode $k$ using the trajectories and discount factor $\gamma$.
5:     Update value network $V_{\theta_V}(s)$ by regression on the mean-square error:

$$\theta_V \leftarrow \arg\min_{\theta_V} \frac{1}{\sum_k |\tau_k|} \sum_{\tau_k \in D} \sum_{i=0}^{|\tau_k|} \left( V(s_{k,i}) - \hat{R}_{k,i} \right)^2 .$$

6:     Estimate advantage $\hat{A}_{k,i}$ for each step $i$ in every episode $k$ using generalized advantage estimation (GAE) and the current value function $V_{\theta_V}(s)$.
7:     Solve the inner optimization using gradient descent (PGD):
8:         For all $k, i$, run PGD to solve (the objective can be solved in a batch):

$$s_{i,\nu} = \arg\min_{s_\nu \in B_{\epsilon_t}(s_{k,i})} g\left( \frac{\pi_\theta(a_{k,i}|s_\nu)}{\pi_{\theta_t}(a_{k,i}|s_{k,i})}, \hat{A}_{k,i} \right),$$

    where $g(x,y) = \min\{x \cdot y, \text{clip}(x, 1 - \eta, 1 + \eta) \cdot y\}$.
9:         Compute the approximation of the outer optimization:

$$\mathcal{L}_{ar}(\theta) = \frac{1}{\sum_k |\tau_k|} \sum_{\tau_k \in D} \sum_{i=0}^{|\tau_k|} \left( -\frac{1}{\beta} \mathcal{H}(\pi_\theta(\cdot|s_{k,i})) - g\left( \frac{\pi_\theta(a_{k,i}|s_{i,\nu})}{\pi_{\theta_t}(a_{k,i}|s_{k,i})}, \hat{A}_{k,i} \right) \right).$$

10:     Update the policy network by minimizing the vanilla PPO objective and the AR objective (the minimization is solved using ADAM):

$$\theta_{t+1} \leftarrow \arg\min_{\theta} \frac{1}{\sum_k |\tau_k|} \sum_{\tau_k \in D} \sum_{i=0}^{|\tau_k|} -g\left( \frac{\pi_\theta(a_{k,i}|s_{k,i})}{\pi_{\theta_t}(a_{k,i}|s_{k,i})}, \hat{A}_{k,i} \right) + \kappa \cdot \mathcal{L}_{ar}(\theta).$$

11: **end for**

---

---

**Algorithm 3** Bilevel Adversarially Robust Proximal Policy Optimization (BAR-PPO).

---

**Input:** Number of iterations $T$, a schedule $\epsilon_t$ for the perturbation radius $\epsilon$, robustness weighting $\kappa$.

---

1: Initialize policy network $\pi_\theta(a|s)$ and value network $V_{\theta_V}(s)$.
2: **for** $t = 1$ **to** $T$ **do**
3:     Run $\pi_{\theta_t}$ in the environment to collect a set of trajectories $\mathcal{D} = \{\tau_k\}$ containing $|\mathcal{D}|$ episodes, each $\tau_k$ is a trajectory containing $|\tau_k|$ samples, $\tau_k := \{(s_{k,i}, a_{k,i}, r_{k,i}, s_{k,i+1})\}, i \in [|\tau_k|]$.
4:     Compute reward-to-go $\hat{R}_{k,i}$ for each step $i$ in every episode $k$ using the trajectories and discount factor $\gamma$.
5:     Update value network $V_{\theta_V}(s)$ by regression on the mean-square error:

$$\theta_V \leftarrow \arg\min_{\theta_V} \frac{1}{\sum_k |\tau_k|} \sum_{\tau_k \in D} \sum_{i=0}^{|\tau_k|} \left( V(s_{k,i}) - \hat{R}_{k,i} \right)^2.$$

6:     Estimate advantage $\hat{A}_{k,i}$ for each step $i$ in every episode $k$ using generalized advantage estimation (GAE) and the current value function $V_{\theta_V}(s)$.
7:     Solve the inner optimization using stochastic gradient Langevin dynamics (SGLD):
8:         For all $k, i$, run SGLD to solve (the objective can be solved in a batch):

$$s_{i,\nu} = \arg\max_{s_\nu \in B_{\epsilon_t}(s_{k,i})} \mathrm{KL}\left( \pi_\theta(\cdot|s_{k,i}) \| \pi_\theta(\cdot|s_\nu) \right).$$

9:         Compute the approximation of the outer optimization:

$$\mathcal{L}_{bar}(\theta) = \frac{1}{\sum_k |\tau_k|} \sum_{\tau_k \in D} \sum_{i=0}^{|\tau_k|} \left( -\frac{1}{\beta} \mathcal{H}\left( \pi_\theta(\cdot|s_{k,i}) \right) - g\left( \frac{\pi_\theta(a_{k,i}|s_{i,\nu})}{\pi_{\theta_t}(a_{k,i}|s_{k,i})}, \hat{A}_{k,i} \right) \right),$$

        where $g(x,y) = \min\left\{ x \cdot y, \mathrm{clip}\left( x, 1 - \eta, 1 + \eta \right) \cdot y \right\}$.
10:    Update the policy network by minimizing the vanilla PPO objective and the BAR objective (the minimization is solved using ADAM):

$$\theta_{t+1} \leftarrow \arg\min_\theta \frac{1}{\sum_k |\tau_k|} \sum_{\tau_k \in D} \sum_{i=0}^{|\tau_k|} -g\left( \frac{\pi_\theta(a_{k,i}|s_{k,i})}{\pi_{\theta_t}(a_{k,i}|s_{k,i})}, \hat{A}_{k,i} \right) + \kappa \cdot \mathcal{L}_{bar}(\theta).$$

11: **end for**

---

# G    ADDITIONAL EXPERIMENT RESULTS

## G.1    COMPARISON OF SPO, ARPO, BARPO WITHOUT GUIDANCE, AND BARPO

In Table 4, we compare the natural and robust performance of SPO, ARPO, BARPO without guidance, and BARPO for continuous control tasks in MuJoCo. We also plot Figure 7, the version of Figure 5 without the standard deviation. We can see that BARPO, without guidance, consistently and significantly outperforms ARPO in natural and robust returns with respect to the mean performance. Moreover, BARPO further improves the overall performance by utilizing the SPO guidance.

## G.2    STATISTICAL SIGNIFICANCE OF COMPARISON RESULTS

We discuss the statistical significance of comparison results between BARPO and baselines from two key considerations: (1) the comparison of mean performance, and (2) a comprehensive analysis incorporating both the mean and the standard deviation.

**Superiority in Mean Performance.** From the perspective of mean returns, BARPO demonstrates a clear and significant advantage. To quantify this, we have calculated the average improvement of BARPO's mean returns over each baseline across all four environments, both with and with-

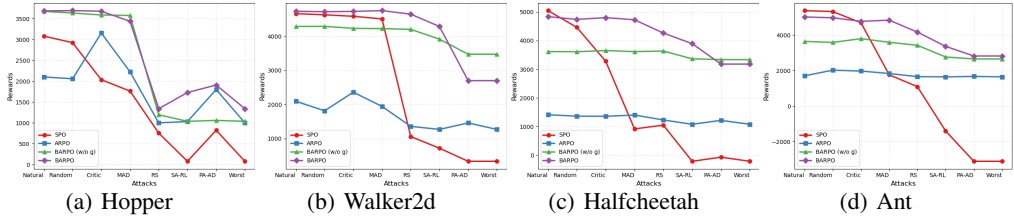

| (a) Hopper | (b) Walker2d | (c) Halfcheetah | (d) Ant |

Figure 7: Natural and robust performance of SPO, ARPO, BARPO without guidance, and BARPO for four continuous control tasks. BARPO (w/o g) consistently outperforms ARPO.

Table 4: Natural and robust performance of SPO, ARPO, BARPO without SPO-guidance, and BARPO for continuous control tasks in MuJoCo. Bold and underlined values indicate the top and second-best performances, respectively.

| Environment | Paradigms | Natural Return | Return under Attack | | | | | | | Worst-case Robustness |
|---|---|---|---|---|---|---|---|---|---|---|
| | | | Random | Critic | MAD | RS | SA-RL | PA-AD | Worst | |
| **Hopper** ($\epsilon = 0.075$) | SPO | 3081 ± 638 | 2923 ± 767 | 2035 ± 1035 | 1763 ± 619 | 756 ± 36 | 79 ± 2 | 823 ± 182 | 79 ± 2 | -0.974 |
| | ARPO | 2101 ± 588 | 2058 ± 559 | 3155 ± 528 | 2225 ± 641 | 1001 ± 30 | 1032 ± 47 | 1799 ± 547 | 1001 ± 30 | **-0.524** |
| | BARPO (w/o g) | 3675 ± 15 | 3634 ± 170 | 3588 ± 263 | 3575 ± 295 | 1195 ± 79 | 1037 ± 24 | 1060 ± 125 | 1037 ± 24 | -0.718 |
| | BARPO | 3684 ± 20 | 3692 ± 24 | 3678 ± 19 | 3437 ± 555 | 1340 ± 42 | 1728 ± 62 | 1908 ± 410 | 1340 ± 42 | -0.636 |
| **Walker2d** ($\epsilon = 0.05$) | SPO | 4662 ± 22 | 4628 ± 21 | 4584 ± 15 | 4507 ± 675 | 1062 ± 150 | 719 ± 1079 | 336 ± 252 | 336 ± 252 | -0.928 |
| | ARPO | 2095 ± 983 | 1815 ± 930 | 2360 ± 1021 | 1944 ± 976 | 1360 ± 835 | 1270 ± 502 | 1460 ± 739 | 1270 ± 502 | -0.394 |
| | BARPO (w/o g) | 4287 ± 15 | 4289 ± 17 | 4231 ± 10 | 4223 ± 406 | 4201 ± 44 | 3913 ± 694 | 3473 ± 495 | 3473 ± 495 | **-0.190** |
| | BARPO | 4732 ± 86 | 4718 ± 59 | 4727 ± 36 | 4750 ± 65 | 4646 ± 53 | 4285 ± 1282 | 2699 ± 1192 | 2699 ± 1192 | -0.436 |
| **Halfcheetah** ($\epsilon = 0.15$) | SPO | 5048 ± 526 | 4463 ± 650 | 3281 ± 1101 | 918 ± 541 | 1049 ± 50 | -213 ± 103 | -69 ± 22 | -213 ± 103 | -1.042 |
| | ARPO | 1412 ± 99 | 1363 ± 285 | 1359 ± 59 | 1402 ± 64 | 1230 ± 75 | 1079 ± 48 | 1216 ± 458 | 1079 ± 48 | -0.236 |
| | BARPO (w/o g) | 3619 ± 44 | 3612 ± 28 | 3654 ± 28 | 3616 ± 37 | 3636 ± 523 | 3365 ± 40 | 3337 ± 45 | 3337 ± 45 | **-0.078** |
| | BARPO | 4837 ± 99 | 4741 ± 501 | 4803 ± 54 | 4729 ± 105 | 4265 ± 1077 | 3894 ± 1322 | 3181 ± 1593 | 3181 ± 1593 | -0.342 |
| **Ant** ($\epsilon = 0.15$) | SPO | 5381 ± 1308 | 5329 ± 976 | 4696 ± 1015 | 1768 ± 929 | 1097 ± 633 | -1398 ± 318 | -3107 ± 1071 | -3107 ± 1071 | -1.577 |
| | ARPO | 1709 ± 564 | 2026 ± 38 | 1976 ± 131 | 1839 ± 350 | 1661 ± 593 | 1648 ± 666 | 1675 ± 573 | 1648 ± 666 | **-0.034** |
| | BARPO (w/o g) | 3647 ± 128 | 3590 ± 431 | 3802 ± 84 | 3597 ± 357 | 3429 ± 679 | 2769 ± 608 | 2659 ± 541 | 2659 ± 541 | -0.270 |
| | BARPO | 5024 ± 117 | 4979 ± 114 | 4777 ± 122 | 4843 ± 120 | 4171 ± 826 | 3367 ± 902 | 2825 ± 757 | 2825 ± 757 | -0.438 |

out attacks. The results, summarized in Table 5, show that BARPO consistently and substantially outperforms the baselines on average.

Table 5: The average improvement of BARPO's mean of returns over baselines.

| Compared Method | Natural Return | Random | Critic | MAD | RS | SA-RL | PA-AD | Worst |
|---|---|---|---|---|---|---|---|---|
| SA | -0.85% | +4.17% | -2.19% | +8.20% | +28.29% | +48.31% | +85.93% | +74.96% |
| RADIAL | +29.40% | +37.89% | +34.20% | +48.35% | +77.60% | +294.27% | +24.88% | +156.68% |
| WocaR | +10.59% | +8.94% | +7.21% | +13.28% | +28.34% | +92.61% | +31.30% | +65.65% |
| SPO | +2.56% | +6.94% | +82.99% | +197.59% | +350.34% | +1238.33% | +1708.38% | +1000.94% |
| ARPO | +159.37% | +158.24% | +128.23% | +149.80% | +168.34% | +167.73% | +80.29% | +103.16% |

**Advantage Considering Standard Deviation.** Beyond the mean, a robust comparison must account for performance variance. Our results show that BARPO's advantage holds even when considering the standard deviation. (1) In the majority of cases, the upper performance bound of the baselines (i.e., mean + standard deviation) is still lower than the mean performance of BARPO alone. (2) More strikingly, in several scenarios, the upper bound of a baseline's performance (mean + standard deviation) is lower than the lower performance bound of BARPO (mean - standard deviation).

These points strongly indicate that the performance distributions are well-separated, and BARPO's superiority is not an artifact of sampling variance. This demonstrates a statistically significant edge over existing methods.

## G.3 EXTENDED EXPERIMENTS ON HUMANOID TASK

Humanoid task in the MuJoCo environment is a 3D bipedal robot designed to simulate a human and is widely considered the most difficult MuJoCo locomotion task, featuring:

• A 348-dimensional observation space ( 31.6x Hopper, 20.5x Walker2d/HalfCheetah, 3.3x Ant).

- A 17-dimensional action space ( 5.7x Hopper, 2.8x Walker2d/HalfCheetah, 2.1x Ant).

For this challenging environment, we implemented a comprehensive comparison of methods: SPO, ARPO, BARPO (w/o g), ARPO (w/ g), BARPO, and the robust baseline SA-PPO. Each method's training was repeated 17 times. We evaluated both the natural performance and the robust performance under four different attack types, and we report the results of the median-performing agent. For the training of ARPO (w/ g), BARPO, and SA-PPO, the hyperparameter $\kappa$ was selected via a grid search over the set $\{0.1, 0.3, 1.0\}$.

Our results, shown in Tables 6 and 7, demonstrate that BARPO continues to achieve a strong natural and robust performance even in this significantly more complex environment. This further validates the effectiveness of our approach.

Table 6: Performance of SPO, ARPO, and BARPO without guidance in the Humanoid task.

| Paradigm | Natural Return | Random | Critic | MAD | RS |
|---|---|---|---|---|---|
| SPO | $5227 \pm 963$ | $5320 \pm 730$ | $4232 \pm 1545$ | $1059 \pm 563$ | $910 \pm 595$ |
| ARPO | $551 \pm 44$ | $544 \pm 47$ | $553 \pm 51$ | $554 \pm 52$ | $522 \pm 60$ |
| BARPO (w/o g) | $\mathbf{6177} \pm 21$ | $\mathbf{6177} \pm 18$ | $\mathbf{6204} \pm 13$ | $\mathbf{6180} \pm 18$ | $\mathbf{5871} \pm 1066$ |

Table 7: Performance of SA-PPO, ARPO with guidance, and BARPO in the Humanoid task.

| Paradigm | Natural Return | Random | Critic | MAD | RS |
|---|---|---|---|---|---|
| SA-PPO | $6315 \pm 12$ | $6313 \pm 16$ | $6385 \pm 10$ | $6321 \pm 15$ | $5889 \pm 92$ |
| ARPO (w/ g) | $6035 \pm 38$ | $6041 \pm 41$ | $5978 \pm 559$ | $5940 \pm 93$ | $5637 \pm 622$ |
| BARPO | $\mathbf{6615} \pm 39$ | $\mathbf{6612} \pm 32$ | $\mathbf{6698} \pm 33$ | $\mathbf{6623} \pm 40$ | $\mathbf{6031} \pm 827$ |

### G.4    ABLATIONS ON REGULARIZATION COEFFICIENT

The regularization coefficient $\kappa$ plays a key role in robust training. Following baselines (Zhang et al., 2020c; Oikarinen et al., 2021; Liang et al., 2022), we performed a limited grid search for BARPO in $\{0.1, 0.3, 1.0\}$. We emphasize that adversarial RL typically requires moderate tuning, and our method introduces no additional burdens than standard methods in the field. We also add ablations to reflect the performance of BARPO across different $\kappa$ choices in Table 8.

Table 8: Natural and worst-case robust performance of BARPO across various $\kappa$.

| Env | $\kappa$ | Natural Return | Worst Robust Return |
|---|---|---|---|
| | 1.0 | 3636 | 1083 |
| Hopper | 0.3 | **3684** | **1340** |
| | 0.1 | 2151 | 959 |
| | 1.0 | 4401 | **3254** |
| Walker2d | 0.3 | **4732** | 2669 |
| | 0.1 | 4672 | 1615 |
| | 1.0 | 3843 | **3436** |
| Halfcheetah | 0.3 | 4837 | 3181 |
| | 0.1 | **5133** | 749 |
| | 1.0 | 5024 | **2825** |
| Ant | 0.3 | 5557 | 1413 |
| | 0.1 | **5596** | 439 |

### G.5    ABLATIONS ON INNER STEPS AND STEP SIZES

We further compare the BARPO's performance with different inner steps (10, 15, 20) and varying step sizes (scaling from $1\times$ to $3\times$ the base ratio of $\epsilon$/steps) in Tables 9 and 10. Across these variations, BARPO consistently maintained its performance, indicating it is not sensitive to specific parameter tuning.

Table 9: Natural and worst-case robust performance of BARPO across various inner steps.

| Env | Return | 10-steps | 15-steps | 20-steps |
|---|---|---|---|---|
| Hopper | Natural | 3677 | **3742** | 3652 |
| | Worst | **1240** | 1049 | 1194 |
| Walker2d | Natural | 4401 | **4733** | 4519 |
| | Worst | **4140** | 3536 | 3913 |
| Halfcheetah | Natural | **4623** | 4408 | 4455 |
| | Worst | 3453 | 3845 | **3972** |
| Ant | Natural | **5024** | 4917 | 4852 |
| | Worst | 4171 | **4333** | 4303 |

Table 10: Natural and worst-case robust performance of BARPO across various inner step sizes.

| Env | Return | $\epsilon$/steps | $2\epsilon$/steps | $3\epsilon$/steps |
|---|---|---|---|---|
| Hopper | Natural | **3677** | 3573 | 3673 |
| | Worst | 1240 | 1242 | **1250** |
| Walker2d | Natural | 4401 | **4424** | 4219 |
| | Worst | 4140 | **4284** | 3159 |
| Halfcheetah | Natural | **4623** | 4307 | 4549 |
| | Worst | 3453 | 3754 | **3988** |
| Ant | Natural | **5024** | 4772 | 4853 |
| | Worst | 4171 | 4124 | **4226** |

## G.6 Further Ablations on Effects of the Bilevel Structure and SPO

As shown in "Effect of SPO Guidance" of Section 5.3 (Figures 5 and 6), we perform ablations comparing BARPO with and without SPO guidance, showing that SPO improves natural return but reduces robustness. Additionally, as detailed in the second part of this section, Table 3 shows that introducing bilevel optimization on top of SPO substantially improves robust return with only minor degradation in natural return.

To further disentangle the effects of the bilevel structure and SPO, we now include additional comparisons (ARPO, ARPO w/ g, BARPO w/o g) and report the relative improvements of ARPO w/ g and BARPO w/o g compared to ARPO in Table 11.

Table 11: Relative improvements of ARPO w/ g and BARPO w/o g compared to ARPO

| Env | Paradigm | Natural Return | Random | Critic | MAD | RS | SA-RL | PA-AD | Worst |
|---|---|---|---|---|---|---|---|---|---|
| Hopper | ARPO (w g) | **+76.0%** | **+77.5%** | **+17.1%** | +40.8% | +12.9% | **+23.8%** | -2.4% | **+12.9%** |
| | BARPO (w/o g) | +74.9% | +76.6% | +13.7% | **+60.7%** | **+19.4%** | +0.5% | -41.1% | +3.6% |
| Walker2d | ARPO (w g) | +100.8% | +133.1% | **+81.5%** | **+117.4%** | +130.8% | +180.7% | +70.9% | +96.5% |
| | BARPO (w/o g) | **+104.6%** | **+136.3%** | +79.3% | +117.2% | **+208.9%** | **+208.1%** | **+137.9%** | **+173.5%** |
| Halfcheetah | ARPO (w g) | **+253.9%** | **+261.9%** | **+270.9%** | +83.3% | +193.1% | +25.7% | -10.7% | +0.6% |
| | BARPO (w/o g) | +156.3% | +165.0% | +168.8% | **+157.9%** | **+195.6%** | **+211.9%** | **+174.4%** | **+209.3%** |
| Ant | ARPO (w g) | **+215.3%** | **+158.0%** | **+141.3%** | **+120.7%** | +60.7% | -28.1% | +4.9% | -28.1% |
| | BARPO (w/o g) | +113.4% | +77.2% | +92.3% | +95.6% | **+106.4%** | **+68.0%** | **+58.7%** | **+61.3%** |

## G.7 Combination Effect of Bilevel Framework and KL Surrogate

We have found that BARPO improves the optimization landscape. We further analyze whether the improvement primarily arises from the bilevel optimization structure itself or from the KL surrogate objective. We clarify that the improved optimization landscape in BARPO arises from the combination effect of both the bilevel optimization structure and the KL surrogate objective. This means that neither component alone is sufficient to achieve the observed improvements.

First, a bilevel structure by itself does not guarantee a smooth or well-behaved landscape. As noted in Section 4.1, both SPO and ARPO can be interpreted as special cases of bilevel optimization. However, they suffer from undesirable first-order stationary policies. The empirical improvements of BARPO over SPO and ARPO thus highlight the importance of a well-chosen surrogate objective. Our Theorem 4.1 provides theoretical support for using the KL divergence as a meaningful and tractable surrogate. In contrast, poorly chosen surrogates without such properties may result in

worse behavior. In particular, the reformulation (36) offers a promising direction for identifying optimal surrogates theoretically.

Second, the KL surrogate alone is insufficient without the bilevel structure. To disentangle the contributions of each component, we conducted an ablation study comparing BARPO to an augmented version of SPO that uses the KL surrogate directly within a minimax formulation. The results (see Table 12) show that BARPO consistently achieves a better trade-off between natural and robust returns. Specifically, in Hopper and HalfCheetah environments, BARPO outperforms the Minimax baseline on both metrics. In Walker2d and Ant, BARPO achieves slightly lower natural returns but significantly higher robust performance, suggesting a better robustness-optimality balance overall.

Table 12: Comparison of KL surrogate with minimax and bilevel structure.

| Env | Structure | Natural Return | Worst Robust Return |
|---|---|---|---|
| Hopper | Minimax | 3518 | 1286 |
| | Bilevel | **3684** | **1340** |
| Walker2d | Minimax | **4875** | 997 |
| | Bilevel | 4732 | **2669** |
| Halfcheetah | Minimax | 4780 | 1443 |
| | Bilevel | **4837** | **3181** |
| Ant | Minimax | **5367** | 2355 |
| | Bilevel | 5024 | **2825** |

# H  ADDITIONAL DISCUSSIONS AND CLARIFICATIONS

## H.1  DISCUSSION ON ASSUMPTIONS IN THEORETICAL ANALYSIS OF CONVERGENCE

We offer the following clarifications and discussions on the assumptions in the theoretical analysis of convergence for ARPO.

**Clarification for core contributions.**  Our main contribution is not the convergence proof itself, but the identification of a fundamental tension between optimality and robustness in policy gradient methods, and further developing a principled solution to mitigate this tradeoff. Our theoretical analysis reveals the root cause of this tradeoff, the landscape distortion induced by the strongest adversaries, and motivates the design of a bilevel framework to alleviate it. The convergence proof merely supports the conclusion that ARPO converges to FOSPs.

**Justification of assumptions.**  We acknowledge that assumptions like Lipschitz smoothness and strong convexity may not strictly hold for deep neural networks. However, they are widely adopted in prior works on adversarial robustness (Sinha et al., 2018; Wang et al., 2019). Lipschitz continuity of sampled policy gradients has been explored and supported in various works (Allen-Zhu et al., 2019; Du et al., 2019; Zou et al., 2020) that study smoothness properties of deep neural networks. The assumption of locally strongly convex adversaries is also common in robust optimization literature (Sinha et al., 2018; Lee & Raginsky, 2018) and can be justified through its connection with distributionally robust optimization.

**Potential relaxations.**  These conditions can be further relaxed. Specifically, the Lipschitz gradient assumption can be relaxed to $(L_0, L_1)$-smoothness (Zhang et al., 2020d;a), generalized smoothness (Li et al., 2023), or smooth adaptivity (Bauschke et al., 2017; Bolte et al., 2018; Ding et al., 2025). Strong convexity can be weakened to weak strong convexity (Necoara et al., 2019), restricted secant inequality (Agarwal et al., 2012), PL condition (Karimi et al., 2016), KL condition (Bolte et al., 2014), error bounds, and quadratic growth conditions (Drusvyatskiy & Lewis, 2018).

## H.2  CLARIFICATION ON NOVELTY OF THE BILEVEL FRAMEWORK

We clarify that our core contributions extend significantly beyond the proposal of using a bilevel framework. We view the bilevel framework not as the end goal, but as a significant tool to address fundamental challenges in adversarial robust policy optimization that we, to our knowledge, are the first to identify and analyze. Our key contributions are cohesively threefold:

**New Insight into Inherent Vulnerability.** We reveal that even under theoretically aligned models, standard policy optimization is inherently vulnerable. We pinpoint the underlying cause: its tendency to converge to fragile first-order stationary policies. This vulnerability is not due to model mismatch but a fundamental issue in the optimization process, which is a perspective overlooked in the prior literature.

**Revealing and Analyzing a New Optimality-Robustness Tension.** We uncover and analyze a fundamental tension between optimality and adversarial robustness in practical policy gradient methods. We further attribute this trade-off to an intrinsic *reshaping effect* induced by the strongest adversary on the optimization landscape, an insight not explicitly identified in prior works. This analysis explains why naively optimizing for robustness can degrade performance.

**A Principled Solution to Mitigate the Trade-off.** Motivated by the above findings, we propose the bilevel optimization framework. More importantly, we propose a practical, theoretically grounded instantiation of this framework designed specifically to mitigate this trade-off and find policies that are both high-performing and robust.

While bilevel optimization as a concept is not new, **its application to analyze and solve this specific, newly identified problem in policy-based RL is, to the best of our knowledge, entirely novel**. We believe the main novelty lies in the complete narrative: identifying the problem, analyzing its root cause, and providing a theoretically-principled solution.

To the best of our knowledge, we are the first to use a bilevel framework to analyze the adversarial robustness of policy-based reinforcement learning.

## H.3 QUANTIFY THE GAP BETWEEN BARPO AND ARPO

We quantify the connection between our KL-based bilevel formulation (BARPO) and the original minimax formulation (ARPO) from two perspectives.

**Theoretical alignment.** We emphasize that under ISA-MDP, BARPO and ARPO share the same global optima in theory. Theorem 4.1 further shows that BARPO's inner optimization is approximately equivalent to that of ARPO.

**Optimization proximity.** Following Kwon et al. (2023); Lu & Mei (2024), for sufficiently large $\alpha$, BARPO can be reformulated:

$$
\max_{\theta, \nu^\diamond} \min_{\vartheta} L^\alpha(\theta, \nu^\diamond, \vartheta) = \mathbb{E}_{s \sim \mu_0} \left[ V^{\pi_\theta \circ \nu^\diamond}(s) \right]
$$
$$
+ \alpha \mathbb{E}_{s \sim \mu_0} \left[ \mathrm{KL}\left( \pi_\theta(s) \| \pi_\theta \circ \nu^\diamond(s) \right) - \mathrm{KL}\left( \pi_\theta(s) \| \pi_\theta \circ \nu_\vartheta(s) \right) \right].
$$

We quantify the deviation from the original objective using the bound:

$$
|L^\alpha(s; \theta, \nu^\diamond, \vartheta) - V^{\pi_\theta \circ \nu_\vartheta}(s)| \leq |V^{\pi_\theta \circ \nu^\diamond}(s) - V^{\pi_\theta \circ \nu_\vartheta}(s)|
$$
$$
+ \alpha |\mathrm{KL}\left( \pi_\theta(s) \| \pi_\theta \circ \nu^\diamond(s) \right) - \mathrm{KL}\left( \pi_\theta(s) \| \pi_\theta \circ \nu_\vartheta(s) \right)|.
$$

The first term is bounded by $C\sqrt{\mathrm{KL}(\pi_\theta \circ \nu^\diamond \| \pi_\theta \circ \nu_\vartheta)}$ through Theorem 5 in Zhang et al. (2020c). The second term can be bounded by $C \left\| \frac{\pi_\theta}{\pi_\theta \circ \nu^\diamond} \right\|_\infty \left\| \frac{\pi_\theta \circ \nu^\diamond}{\pi_\theta \circ \nu_\vartheta} \right\|_\infty \sqrt{\mathrm{KL}(\pi_\theta \circ \nu^\diamond \| \pi_\theta \circ \nu_\vartheta)}$ by reverse Pinsker's and Pinsker's inequalities. Specifically, when $\pi_\theta \circ \nu^\diamond$ and $\pi_\theta \circ \nu_\vartheta$ are not too far apart, the second term is also $O\left( \sqrt{\mathrm{KL}(\pi_\theta \circ \nu^\diamond \| \pi_\theta \circ \nu_\vartheta)} \right)$ by Taylor expansion. Thus, the deviation between BARPO and ARPO can be controlled by the square root of the maximum KL divergence between $\pi_\theta \circ \nu^\diamond$ and $\pi_\theta \circ \nu_\vartheta$.

These connections indicate that BARPO, while structurally different, retains a strong theoretical connection to the ARPO.

## H.4 FURTHER THEORETICAL UNDERSTANDING OF BARPO

We further delve into a deeper theoretical understanding of how BARPO reshapes the optimization landscape from three complementary perspectives:

**Adversarial Value Improvement through Landscape Lifting.** Let $\nu^\diamond$ denote the solution to BARPO's inner problem and $\nu^*$ denote the strongest adversary defined in ARPO. By definition, $V^{\pi \circ \nu^\diamond}(\pi) \geq V^{\pi \circ \nu^*}(\pi)$, indicating that BARPO tends to elevate low-value regions in ARPO's landscape. Furthermore, Theorem 4.1 shows that maximizing the KL surrogate aligns with minimizing adversarial value: when the KL surrogate is high, the adversarial value is correspondingly low. This provides a theoretical link between BARPO's surrogate objective and ARPO's robustness.

**Smoothing via Regularized Maximin Reformulation.** Following prior work (Kwon et al., 2023; Lu & Mei, 2024), when $\alpha$ is sufficiently large, BARPO can be reformulated as an equivalent maximin problem:

$$\max_{\theta, \nu^\diamond} \min_{\vartheta} L^\alpha(\theta, \nu^\diamond, \vartheta) = \mathbb{E}_{s \sim \mu_0}[V^{\pi_\theta \circ \nu^\diamond}(s) + \alpha(\mathrm{KL}\left(\pi_\theta(s) \| \pi_\theta \circ \nu^\diamond(s)\right) - \mathrm{KL}\left(\pi_\theta(s) \| \pi_\theta \circ \nu_\vartheta(s)\right))]. \tag{36}$$

Compared to ARPO's objective, this formulation introduces a regularization term that has a smoothing effect on the outer landscape. Structurally, this resembles the use of Bregman divergences in defining the Moreau envelope (Bauschke et al., 2018), which is known to smooth the nonsmooth optimization problems.

**Robustness under Date-driven Value Approximation.** In practice, ARPO optimizes a lower bound $L(\pi, \nu) \leq V^{\pi \circ \nu}$ due to practical data-driven estimation for $V^{\pi \circ \nu}$, i.e., the practical objective becomes $\max_\pi \min_\nu L(\pi, \nu)$, which is no longer equivalent to the ideal adversarial objective. In contrast, BARPO's inner optimization minimizes an upper bound on the adversarial value, while the outer loop maximizes a lower bound. This formulation is more faithful to the original robust motivation and offers better resilience under value function approximation, helping to improve performance in practice.

We believe these perspectives offer promising foundations toward a more complete theoretical understanding of BARPO and plan to explore them in greater depth in future work.

## I    LIMITATIONS AND FUTURE WORK

Despite the advancements introduced in this work, several challenges remain open for future research. First, we solve BARPO using a straightforward approach that ignores the second-order term. This solution process can be further improved via an advanced optimization method on a bilevel problem, such as transforming the bilevel optimization into a penalty-based minimax formulation and then solving it via fully first-order gradient methods. Additionally, we derive a surrogate for the strongest adversaries under specific conditions, which approximates but does not coincide with the strongest adversary. Nonetheless, BARPO, based on this surrogate, achieves strong robustness, even significantly outperforming ARPO, which relies on the strongest adversary, in tasks like Walker2d and HalfCheetah. While these empirical results are somewhat aligned with theoretical consistency between optimality and robustness, there remains a larger theoretical space to further explore. Finally, our analysis and results can be naturally extended to more general settings where strict theoretical consistency may not hold, offering valuable insights for related fields.

## J    USE OF LARGE LANGUAGE MODELS

The core method development in this research does not involve large language models (LLMs) as any important, original, or non-standard components. We utilized the LLM to assist only in the writing process of this manuscript. All content, ideas, and arguments were conceived and written by the human authors. The LLM was then used to provide suggestions for improving grammar, phrasing, and readability. The authors reviewed these suggestions and retained full editorial control, taking sole responsibility for the final version of the text.

