# OpenReview forum: "On the Tension Between Optimality and Adversarial Robustness in Policy Optimization"
_ICLR.cc/2026/Conference — ICLR 2026 Poster_

### Official Review · Reviewer_RuhM · 2025-10-27

**Soundness:** 3
**Presentation:** 3
**Contribution:** 3
**Rating:** 6
**Confidence:** 3

**Summary:**

This paper investigates the longstanding tradeoff between optimality and adversarial robustness in deep reinforcement learning (DRL). The authors find that, in practice, standard policy optimization (SPO) and adversarially robust policy optimization (ARPO) still exhibit a tension: SPO yields high returns but is fragile to perturbations, while ARPO improves robustness but suffers large performance drops. To address this, the authors propose BARPO (Bilevel Adversarially Robust Policy Optimization), a bilevel framework that modulates adversary strength to preserve robustness while improving landscape smoothness and navigability. BARPO generalizes SPO and ARPO as limiting cases. Theoretical analyses show convergence to FOSPs and introduce a KL-based surrogate for the adversary. Experiments on MuJoCo continuous-control tasks demonstrate that BARPO outperforms ARPO and strong baselines in both natural and robust returns.

**Strengths:**

- The paper tackles a central and underexplored issue with the trade-off between robustness and optimality, offering a novel and intuitive geometric explanation for the empirical gap between theory and practice.
- Strong theoretical support and comprehensive empirical results.
- Figures illustrating optimization landscapes (e.g., SPO vs. ARPO vs. BARPO) are intuitive and effectively convey the main ideas.

**Weaknesses:**

- The bilevel formulation and KL-based inner optimization introduce extra hyper-parameters  and computational cost. An explicit runtime and sensitivity analysis would strengthen the empirical case.
- Robust RL includes several major paradigms (e.g., distributionally robust MDPs, risk-sensitive RL, adversarial policy training). Could the authors explicitly clarify how BARPO relates to these categories? In particular, how does it extend or differ conceptually and algorithmically from prior adversarial training methods such as [1] and its latest extensions like [2]? A concise comparative discussion (e.g., in the related work) would make the contribution and novelty more evident.

#### [1] Pineto et al., "Robust Adversarial Reinforcement Learning." ICML 2017
#### [2] Dong et al., "Variational Adversarial Training Towards Policies with Improved Robustness." AISTATS 2025

**Questions:**

- Could BARPO and all the analysis between optimality and robustness extend to other adversarial settings instead of limiting to state perturbation?
- Although BARPO seems promising from the empirical results compared with all the baselines, those baselines may not be the state-of-the-art, where the latest one was published in 2022. Compare with more recent work may make this work more convincing. For example, [1] is also mentioned in the related work while no direct comparison in the experiments.

#### [1]  Liang et al., "Game-theoretic robust reinforcement learning handles temporally-coupled perturbations." ICLR 2024

---

> ### Author Response · Authors · 2025-11-21
>
> ---
> **W1: Analysis of computational overhead and hyperparameter sensitivity**
>
> **A:** We would like to clarify that our approach does not introduce more hyperparameters and computational costs than robust baselines.
>
> - Runtime analysis (provided in Appendix F.4): All MuJoCo experiments were conducted on a machine with an AMD EPYC 7742 CPU and no GPU acceleration.
> BARPO's training takes approximately 2 hours for Hopper, Walker2d, and HalfCheetah, and 9 hours for the more complex Ant environment. This represents roughly a 2x increase in wall-clock time compared to standard, non-robust PPO training. Crucially, this overhead is on par with other leading robust baselines, which require similar inner-loop computations to find the adversary. The choice of a CPU-only setup follows baselines, as the wall-clock time in these MuJoCo tasks is overwhelmingly dominated by the environment simulation overhead, ensuring a fair comparison of the algorithmic sample complexity.
> Furthermore, we confirm that BARPO introduces no significant additional memory overhead compared to standard PPO training.
>
> - Sensitivity analysis (provided in Appendix G.4): Our formulation does not impose a greater hyperparameter tuning burden than existing methods. The primary hyperparameter requiring adjustment is the regularization coefficient $\kappa$, which plays a key role in robust training. Following baselines, we performed a limited grid search for BARPO in $\{0.1, 0.3, 1.0\}$. We emphasize that adversarial RL typically requires moderate tuning, and our method introduces no additional burdens than standard methods in the field. We also add ablations to reflect the performance of BARPO across different $\kappa$ choices in the following.
>
> | Env |$\kappa$ | Natural Return | Worst Robust Return |
> |-|-|-|-|
> | Hopper | 1.0 | 3636 | 1083 |
> | | 0.3 | **3684** | **1340** |
> | | 0.1 | 2151 | 959 |
> | Walker2d | 1.0 | 4401 | **3254** |
> | | 0.3 | **4732** | 2669 |
> | | 0.1 | 4672 | 1615 |
> | Halfcheetah | 1.0 | 3843 | **3436** |
> | | 0.3 | 4837 | 3181 |
> | | 0.1 | **5133** | 749 |
> | Ant | 1.0 | 5024 | **2825** |
> | | 0.3 | 5557 | 1413 |
> | | 0.1 | **5596** | 439 |
>
> ---
> **W2: Clarification for the training paradigm**
>
> **A:**
>
> - Our approach follows the adversarial training paradigm from [1], where the adversary is explicitly computed by solving an inner optimization problem for a given agent policy, and then the agent is trained based on this adversary.
>
> - Comparison with [2,3]: While these works are seminal, our approach differs in three fundamental ways.
>
>     - Treat model: [2,3] focus on perturbations to the environment, i.e., the transition dynamics and reward functions. These perturbations are decoupled from the agent's policy. In contrast, our work focuses on state perturbations, which directly alter the information the agent uses for decision-making. This is a fundamentally different type of threat model.
>
>     - Training paradigm: [2,3] utilize a multi-agent paradigm, where an agent and an adversary are trained alternately. This is different from our paradigm, where the adversary is directly computed via an inner optimization problem.
>
>     - Algorithmically, [3] uses the adversary distribution instead of a deterministic adversary and maximizes the lower quantile of the value functions. In addition, BARPO can also be seen as a solution to the over-pessimism adversary proposed by [3].
>
> These paradigms and approaches are complementary to our work and can be integrated.
>
> [1] Madry et al., "Towards deep learning models resistant to adversarial attacks." ICLR 2018
>
> [2] Pinto et al., "Robust adversarial reinforcement learning." ICML 2017
>
> [3] Dong et al., "Variational Adversarial Training Towards Policies with Improved Robustness." AISTATS 2025

---

> ### Author Response · Authors · 2025-11-21
>
> ---
> **Q1: Generalization to other adversarial settings**
>
> **A:** We believe our framework and analysis can be effectively extended to broader adversarial settings beyond state perturbations.
>
> - Action perturbations: Extending our method to action perturbations is straightforward. Since these perturbations are directly coupled with the agent's policy, they share the same structural characteristics as state perturbations. Therefore, the current algorithm applies with minimal modification.
>
> - Transition perturbations: We also see a clear path to extending our work to robustness against perturbations in environment dynamics. Our analysis spans four main aspects: (1) the convergence of ARPO, (2) the learning dynamics of SPO and ARPO, (3) geometric analysis of value functions in the ISA-MDP, and (4) KL-based surrogates for the strongest adversary.
>
>     1. Theoretical convergence: For the first two aspects (1) and (2), the core ideas can be naturally extended following the framework of Distributionally Robust Optimization [4].
>
>     2. Geometric properties: For the third aspect (3), [5] has already considered robustness in transitions, and we believe our results on suboptimality and connectivity can be adapted to that setting by leveraging similar techniques.
>
>     3. Algorithmic implementation: Finally, our KL-based surrogate framework for approximating the inner maximization (i.e., the worst-case adversary) can also be generalized to uncertainty over transition kernels. In particular, it is possible to obtain analogous lower bound guarantees of the form $O(\operatorname{KL}(P^\pi | P^\pi_\nu))$ where $P^\pi$ and $P^\pi_\nu$ denote the nominal and perturbed transition kernels under policy $\pi$, respectively. Based on this, BARPO can extend to this setting.
>
> [4] Bertsimas et al., "Data-driven robust optimization." Mathematical Programming, 2018
>
> [5] Wang et al., "The geometry of robust value functions." ICML 2022
>
> ---
> **Q2: Clarification for baselines**
>
> **A:** We focused our comparison on baselines that share our specific training paradigm to ensure a fair and rigorous evaluation.
>
> - Relevance of selected baselines: BARPO operates within the "explicit adversary computation" paradigm. In this framework, the adversary is determined by solving an inner optimization problem before the agent updates. For this specific algorithmic class, the baselines we selected remain the current state-of-the-art.
>
> - Comparison with [6]: [6] employs a multi-agent alternating training paradigm. This differs fundamentally from our approach. Their method trains an agent and an adversary in turns, whereas we solve for the strongest adversary directly at each step. These approaches are complementary, and their framework could potentially integrate our proposed algorithms. It also seems that the official implementation and models are not currently open-source. Therefore, we were unable to perform a direct empirical comparison.
>
> [6] Liang et al., "Game-theoretic robust reinforcement learning handles temporally-coupled perturbations." ICLR 2024

---

> > ### Comment · Reviewer_RuhM · 2025-11-26
> >
> > Thanks authors' for all their responses. I do not have further questions and would like to keep my positive rating.

---

> > > ### Author Response · Authors · 2025-11-26
> > >
> > > Thanks for your feedback and thoughtful consideration! We sincerely appreciate your valuable comments, prompt response, and recognition of our work.

---

### Official Review · Reviewer_LLxi · 2025-10-30

**Soundness:** 3
**Presentation:** 3
**Contribution:** 3
**Rating:** 6
**Confidence:** 2

**Summary:**

The paper is mathematically sound and shows that ARPO can converge to first-order stationary policies (FOSPs) rather than global optima. It also empirically explains FOSP for both SPO and ARPO. To address these issues, the authors introduce BARPO, a bi-level variant that reshapes the optimization landscape and yields improved robustness in practice.

**Strengths:**

+ The theoretical development is clear, and the paper is well written.

+ The paper explains the FOSPs in SPO and ARPO and motivates a unified and clear way.

+ On MuJoCo benchmarks, BAR-PPO shows strong natural and robust performance and adding SPO guidance improves clean returns.

**Weaknesses:**

--

**Questions:**

- Is the BARPO also robust to various attacks? Are the $\varepsilon$ budgets predefined per task, and would changing $\varepsilon$ alter the reported metrics and the ranking between methods?

- How sensitive are results to the number of inner steps, the step size, and the inner optimizer used to solve the adversary?

- Table 3 shows BARPO with SPO guidance outperforming ARPO with guidance. Quantitatively, how does BARPO with guidance compare against the strongest baselines in Table 2 under the same attack settings?

- Why use SPO guidance in BAR-PPO? What would happen if you instead added an ARPO-style guidance term?

**Details Of Ethics Concerns:**

--

---

> ### Author Response · Authors · 2025-11-21
>
> ---
> **Q1: Evaluation of robustness**
>
> **A:** We evaluate BARPO with six different attacks (shown in Table 3), demonstrating its robustness against various types of attacks.
>
> Regarding perturbation budgets, we adopted values aligned with baselines to ensure a fair comparison. To further validate the effectiveness of BARPO, we train and test BARPO and baselines with larger budgets (0.1 for Hopper and Walker2d, 0.2 for Halfcheetah and Ant). The results demonstrate that BARPO consistently maintains a superior trade-off between robustness and performance.
>
> |Env|Return|SA|Radial|WocaR|BAR|
> |---|---|---|---|---|---|
> |Hopper|Natural|3163|3589|3653|**3661**|
> ||Worst|1032|1220|1090|**1265**|
> |Walker2d|Natural|4609|3619|4316|**4681**|
> ||Worst|1585|1319|1390|**2023**|
> |Halfcheetah|Natural|4226|**4574**|4273|4483|
> ||Worst|2905|1112|2589|**3014**|
> |Ant|Natural|4708|3897|4121|**4715**|
> ||Worst|**3860**|2930|3089|3841|
>
> ---
> **Q2: Sensitivity to inner optimization hyperparameters**
>
> **A:** We further compare the BARPO's performance with different inner steps (10, 15, 20) and varying step sizes (scaling from $1\times$ to $3\times$ the base ratio of $\epsilon$/steps). Across these variations, BARPO consistently maintained its performance, indicating it is not sensitive to specific parameter tuning.
>
> |Env|Return|10-steps|15-steps|20-steps|
> |---|---|---|---|---|
> |Hopper|Natural|3677|**3742**|3652|
> ||Worst|**1240**|1049|1194|
> |Walker2d|Natural|4401|**4733**|4519|
> ||Worst|**4140**|3536|3913|
> |Halfcheetah|Natural|**4623**|4408|4455|
> ||Worst|3453|3845|**3972**|
> |Ant|Natural|**5024**|4917|4852|
> ||Worst|4171|**4333**|4303|
>
>
>
> |Env|Return|$\epsilon$/steps|$2\epsilon$/steps|$3\epsilon$/steps|
> |---|---|---|---|---|
> |Hopper|Natural|**3677**|3573|3673|
> ||Worst|1240|1242|**1250**|
> |Walker2d|Natural|4401|**4424**|4219|
> ||Worst|4140|**4284**|3159|
> |Halfcheetah|Natural|**4623**|4307|4549|
> ||Worst|3453|3754|**3988**|
> |Ant|Natural|**5024**|4772|4853|
> ||Worst|4171|4124|**4226**|
>
>
> For the inner optimizer, standard gradient-based optimizers, such as PGD or FGSM, are not suitable in this context because the clean state acts as a stationary point with zero gradient for our objective function. We specifically utilize Stochastic Gradient Langevin Dynamics (SGLD) because it introduces the necessary noise to escape these stationary points. Currently, SGLD is the most effective tool for navigating this specific optimization landscape.
>
> ---
> **Q3: Quantitative comparison against strongest baselines**
>
> **A:** We conducted a direct comparison between BARPO (with guidance) and the top-performing baselines from Table 2 under identical attack settings. The results show that in Hopper and Halfcheetah tasks, BARPO achieves consistently higher natural and worst-case returns. For Walker2d and Ant, while BARPO shows a slight dip in natural performance compared to the top baselines, it delivers substantial gains in robustness.
>
>
> | Environment | Method | Natural Return | Random | Critic | MAD | RS | SA-RL | PA-AD | Worst | Worst-case Robustness |
> |-------------|--------|----------------|--------|--------|-----|----|-------|-------|-------|----------------------|
> | Hopper | WocaR | 3629| 3637 | 3657 | 3150 | 1171| 1452 | **2124** | 1171 | -0.677 |
> | | BARPO | **3684** | **3692** | **3678** | **3437** | **1340** | **1728** | 1908 | **1340** | **-0.636** |
> | Walker2d | SA | **4875** | **4907** | **5029** | **4833** | 2775 | 3356 | 997 | 997 | -0.795 |
> | | BARPO | 4732 | 4718| 4727| 4750 | **4646** | **4285** | **2699** | **2699** | **-0.436** |
> | Halfcheetah | WocaR | 4723 | **4798** | **4846** | 4543 | 3302 | 2270 | 2498 | 2270 | -0.519 |
> | | BARPO | **4837** | 4741 | 4803 | **4729** | **4265** | **3894** | **3181** | **3181** | **-0.342** |
> | Ant | SA | **5367** | **5217** | **5012** | **5114** | **4396** | **4227** | 2355 | 2355 | -0.539 |
> | | BARPO | 5024 | 4979 | 4777 | 4843 | 4171 | 3367 | **2825** | **2825** | **-0.438** |
>
> ---
> **Q4: Rationale for choosing SPO guidance over ARPO-style guidance**
>
> **A:** We selected SPO guidance to achieve an optimal balance between natural and robust performance. Our rationale is based on two key observations from our experiments (Figure 5 and Table 4):
>
> - Addressing natural performance gaps: While BAR-PPO without guidance significantly outperforms ARPO in both natural and robust returns, it still lags behind SPO in terms of natural performance, particularly in complex environments like HalfCheetah and Ant. We incorporate SPO guidance specifically to bridge this gap, helping the agent recover natural capabilities without compromising its learned robustness.
>
> - Trade-off with ARPO guidance: Utilizing ARPO-style guidance would impose stricter constraints focused solely on robustness. While this might yield marginal gains in adversarial robustness, it would likely come at the cost of significantly reduced natural performance.

---

### Official Review · Reviewer_wh98 · 2025-10-30

**Soundness:** 2
**Presentation:** 2
**Contribution:** 3
**Rating:** 4
**Confidence:** 3

**Summary:**

This work investigates a conflict between theoretical and practical consistency between reinforcement learning (RL) policies' optimality and adversarial robustness. A prior work proposes the concept of ISA-MDP, where the optimal robust policy (ORP) is the same as the nominal optimal policy under the Bellman optimality. However, in practice, such consistency can rarely be achieved. This works analyzes why the theoretical alignment cannot be practically realized, trying to close the gap. In particular, the authors analyze the optimization dynamic of standard policy optimization (SPO) and adversarially robust policy optimization (ARPO). They first prove that both SPO and ARPO converges to stationary policies rather than global optima. Then, they observe that SPO is vulnerable and ARPO is robust yet yielding significantly lower nominal returns, showing a critical tradeoff between robustness and optimality. To address this challenge, the authors analyze the optimization landscape and value geometry induced by SPO and ARPO. Motivated by their findings, a bilevel optimization framework is proposed to achieve a good balance, achieving robustness without sacrifice of nominal returns.

**Strengths:**

- To the best of reviewer's knowledge, the convergence analysis for the max-min optimization of ARPO is new.
- The geometry illustration (Figure 2) in Section 3.2 is informative and helpful.
- The experiments are comprehensive.
- The proposed bi-level optimization problem is intuitive (although the motivation and path reaching there are highly confusing).

**Weaknesses:**

- There is no interpretation or explanation for Proposition 3.1. $V^-$ is undefined. I am not sure why this result is useful. It only states that the gap between the nominal return and the return under attack can be much larger for ARPO than SPO polices. In fact, the whole subsection 3.1.2 feels disconnected from the rest of the work.
- I expect the author to draw insights from their theoretical analysis in 3.1.1 but they only provide empirical evidence. In addition, the empirical evidence seems to be conflicting with existing works and thus cannot provide strong evidence for the claimed tension between optimality and robustness.
- While Figure 2 of Section 3.2 is informative, the rest of it is confusing. In particular, $\Pi_{Robust}$ in proposition 3.2 is undefined. Moreover, the claims of proposition 3.2 is vague. The authors needs to explain the vague statement "ARPO tends to yield more isolated FOSPs than SPO" while Proposition 3.2 only proves that there exist an ISA-MDP with a cut point.
- Figure 3 is very confusing.
- After Section 3, I don't see a rigorously presented tension between optimality and robustness, contradicting the claim in the introduction that "We uncover a new form of optimality-robustness tension arising in policy optimization". I would recommend the authors to consider enhancing the presentation there and make the components of this work more connected.
- There is a same problem with Theorem 4.1. It appears disconnected from the proposed algorithm. The authors claim that "we need to specify a suitable inner objective G that both promotes policy learning and maintains strong robustness" and prove that KL divergence is an effective surrogate. But it is unclear why such surrogate leads to better policy learning and maintains robustness.
- While Figure 4 is helpful, it is disconnected from the theorem.
- Overall, the presentation of work needs considerable improvements, preventing the reviewer from evaluating the soundness of the proposed algorithm. While I see the technical contribution of this work, its lack in presentation coherence and smoothness severely undermines the quality of this work.

**Questions:**

- What is $\theta_0$ in Theorem 3.2?
- I think in the original ISA-MDP paper[1], the authors' implemented algorithms (CAR-PPO-SGLD/PGD) do not show significant natural/nominal performance drop without attacks, sometimes even outperforming SPO methods (PPO in [1]). This observation is consistent with empirical robust RL works such as [2]. Why the performance of ARPO drops this badly in Table 1?
- Could you explain what Figure 3 is trying the say?
- Why using a lower bound in Theorem 4.1 as the surrogate? If the goal is to minimize the gap between the nominal and under-attack returns, then shouldn't we use an upper bound?
- Where do Figure 4 comes from? Are the values in the figure true objective values (of SPO, ARPO and BARPO) for some RL problems, or are the values only for illustration purposes?
- Could you please explain in detail the connection between different parts of Section 4.2? Specifically, why "KL divergence serves as a valid first-order surrogate for minimizing the adversarial value" can lead to better policy learning? And why "This surrogate is both appropriate and reliable from an optimization perspective"? Lastly, why the lower bound in Theorem 4.1 leads to the proposed bi-level optimization problem? The reviewer fails to see the connections.
- While the proposed bi-level optimization problem is intuitive enough (restricting the perturbed policy to be close to the original policy), why this has the effect of changing land scape? In particular, how can it "reshapes the landscape by elevating low-value but robust regions"? I believe the authors state this claim without any justification.


[1] Towards Optimal Adversarial Robust Reinforcement Learning with Infinity Measurement Error
[2] Robust Adversarial Reinforcement Learning

---

> ### Author Response · Authors · 2025-11-21
>
> ---
> **W1. Clarification on Proposition, Figure, and Theorem Details**
>
> **A:** We sincerely appreciate the reviewer’s thoughtful and comprehensive feedback on the presentation and overall flow of our work. We apologize for any confusion caused by points that we intended to highlight but might not have highlighted sufficiently. We believe these issues are either already addressed or can be resolved with minor clarifications. We sincerely value the reviewer’s time and effort, and we kindly ask for consideration of the overall contributions of our work.
>
>
> 1. About Proposition 3.1 and subsection 3.1.2
>
>     - Definition of $V^-$: $V^-$ denotes the worst-case value achievable by any policy in the ISA-MDP, i.e., $V^-(\mu_0) = \min_\pi V^{\pi}(\mu_0)$.
>
>     - Usefulness of Proposition 3.1: The result in Proposition 3.1 is highly useful because it is the formal theoretical evidence of the policy optimization challenge.
>
>         - It rigorously proves that in an exact ISA-MDP, the FOSPs found by ARPO have significantly degraded returns compared to the FOSPs found by SPO.
>
>         - This result is not redundant with our empirical findings in MuJoCo. By proving this result on the exact, controllable ISA-MDP example, we isolate and confirm that the observed optimality-robustness tension is an inherent property that arises directly from the policy optimization process itself, rather than complex environmental factors or previously studied conflicting objectives.
>
>     - Role of Subsection 3.1.2 in the overall work: Subsection 3.1.2 plays a critical, connecting role in the paper.
>
>         - Subsection 3.1.1 establishes that both SPO and ARPO converge to their FOSPs.
>
>         - Subsection 3.1.2 then rigorously analyzes the properties of these FOSPs, thereby identifying the new form of optimality-robustness tension.
>
>         - Section 3.2 analyzes the underlying cause of this identified tension, attributing this to an intrinsic reshaping effect induced by the strongest adversary on the optimization landscape.
>
>         - Based on these analyses, Section 4 proposes BARPO to address the tension identified in Subsection 3.1.2.
>
>
> 2. About analysis in 3.1
>
>     - Theoretical insights in Section 3.1: The analysis in Section 3.1.2 is intentionally a combination of empirical demonstration and theoretical rigor, jointly establishing the new form of optimality-robustness tension.
>
>         - We provide empirical evidence in complex MuJoCo environments to demonstrate the phenomenon's prevalence.
>
>         - Crucially, we also provide rigorous theoretical analysis on a specific ISA-MDP (as seen in Proposition 3.1 and detailed in Appendix D) to prove that this tension is inherent to the policy optimization process.
>
>         - Furthermore, due to space limitations, a more detailed theoretical analysis of the learning dynamics of SPO and ARPO is included in Appendix C.3.
>
>     - Conflict with existing work: We clarify that our empirical evidence does not conflict with existing works. The perceived discrepancy regarding the drop in nominal performance is explained by the fundamental differences in problem settings and the use of stabilizing techniques in prior works. For a detailed explanation of why our empirical results are consistent with the true behavior of this class of algorithms, please refer to our response to Question 2.
>
>
> 3. About Proposition 3.2
>
>     - Definition of $\Pi_{Robust}$: $\Pi_{Robust} = \\{ \pi \in \Pi | V^\pi(s) = V^{\pi \circ \nu^* (\pi)}(s), \forall s \\}$ (detailed in Appendix D) is the robust policy set, which contains all policies $\pi$ whose values are invariant to their strongest adversary $\nu^* (\pi)$.
>
>     - Explanation of Proposition 3.2 and Isolated FOSPs: Proposition 3.2 formally establishes the geometric basis for the optimization difficulty caused by ARPO's distorted landscape.
>
>         - The proof demonstrates the existence of a cut point (a region of poor connectivity) on the path connecting two FOSP regions within the robust policy set of the simple ISA-MDP.
>
>         - The existence of such a cut point signifies that the optimization landscape is highly fractured. It confirms that the path between different FOSPs is either disconnected or requires traversing a barrier, thus hindering optimization trajectories.
>
>         - This theoretical proof directly supports our claim that ARPO tends to yield more isolated FOSPs than SPO. The intuitive visualization of this effect can be seen in Figure 4(b), which clearly shows the distinct, separated FOSP regions in the ARPO objective, separated by barriers and a cut point.

---

> ### Author Response · Authors · 2025-11-21
>
> 4. About Figure 3
>
>     Figure 3 visualizes the value function geometry of a specific two-state ISA-MDP (detailed in Appendix D) to illustrate the distorted landscape's impact on policy optimization.
>
>     The axes represent the value functions for the two states, i.e., $(V(s_1), V(s_2))$. The value space consists of the value functions corresponding to all policies, i.e., $\mathcal{V} = \\{ (V^\pi(s_1), V^\pi(s_2)) | \pi \in \Pi \\}$. The robust value space consists of the value functions corresponding to robust policies, i.e., $\mathcal{V} _ {Robust} = \\{ (V^\pi(s_1), V^\pi(s_2)) | \pi \in \Pi_{Robust}  \\}$, where $\Pi_{Robust} = \\{ \pi \in \Pi | V^\pi(s_i) = V^{\pi \circ \nu^*(\pi)}(s_i), \forall i=1,2 \\}$ is the robust policy set.
>
>     The left panel distinguishes between the standard value space and the robust value space. It highlights that the robust value space occupies a specific, connected subset of the value space.
>
>     The right panel demonstrates how ARPO leads to two distinct outcomes. ARPO with the initial policy in the yellow region successfully converges to the optimal robust policy, while ARPO with the initial policy in the green region gets trapped in a deceptive FOSP.
>
>     In summary, this figure provides concrete visual evidence that ARPO's distorted landscape impairs policy navigation. In this specific example, roughly one-third of potential starting policies lead to failure. This visually confirms that the reshaped landscape makes policy optimization significantly more difficult and prone to suboptimal convergence.
>
>
> 5. About the new form of optimality-robustness tension
>
>     The "new form of optimality-robustness tension" is the following observation, established in Section 3.1.2 and summarized in lines 234-236: FOSPs discovered by ARPO tend to exhibit robustness but generally achieve substantially reduced returns than those achieved by SPO.
>
>     The tension is considered "new" because we identify its source as arising from the policy optimization process itself, specifically the geometric distortion of the landscape, rather than the previously studied conflict between the standard and adversarial objective policies.
>
>     We rigorously demonstrate this crucial distinction through our analysis in Proposition 3.1, which proves that even in an exact, controlled ISA-MDP where the standard and robust objectives are not inherently conflicting, this severe performance penalty still exists at the convergence points.
>
>
> 6. About Theorem 4.1
>
>     Please refer to our response to Questions 6 and 7.
>
>
> 7. About Figure 4
>
>     Figure 4 serves as a crucial intuitive bridge between our abstract theoretical claims and the practical impact on optimization.
>
>     - Figure 4(b) provides a visual realization of our theoretical findings regarding landscape distortion (Proposition 3.2):
>
>         - The figure demonstrates the landscape of the ARPO objective, illustrating the vulnerable connectivity (i.e., the existence of barriers and cut points) between different policy convergence (FOSP) regions.
>
>         - This visualization makes the abstract claim of landscape distortion concrete, intuitively explaining why policy navigation is impaired and why ARPO often gets trapped in suboptimal FOSPs.
>
>     - Figure 4(c) illustrates the effect of our proposed BARPO:
>
>         - It shows how BARPO successfully reshapes the optimization landscape by strategically elevating the value of robust, low-performing policy regions.
>
>         - This transformation creates a much smoother and more favorable optimization trajectory, enabling the agent to maintain navigation in the robust policy space while avoiding the traps shown in Figure 4(b), thus leading to both strong performance and robustness.
>
> ---
> **Q1: Meaning of $\theta_0$**
>
> **A:** $\theta_0$ represents the initial parameters of the trained policy.

---

> ### Author Response · Authors · 2025-11-21
>
> ---
> **Q2: Clarify the natural performance drop of ARPO compared with other work**
>
> **A:** Thank you for this insightful question. We clarify why the performance drop of ARPO in natural environments does not contradict previous work [1,2].
>
> - First, the comparison with [1] requires careful distinction. The CAR-PPO algorithms presented in [1] utilize SPO guidance in their default implementation. This technique helps preserve performance in nominal settings. Therefore, their method is more analogous to the "ARPO (w g)" variant (discussed in Table 3), which also employs this guidance and performs well in natural returns, rather than the ARPO.
>
> To confirm this, we implement a version of CAR-PPO without SPO guidance. From the following table, we observe that this version also suffers a significant drop in natural performance, similar to the ARPO.
>
> |Env|Model|clean|random|critic|MAD|min RS|
> |---|---|---|---|---|---|---|
> |Hopper|SPO|3081|2923|2035|1763|756|
> ||CAR (w/o g, pgd)|2137|2409|3479|2186|1410|
> ||CAR (w/o g, sgld)|1375|1375|1625|1314|940|
> |Walker2d|SPO|4662|4628|4584|4507|1062|
> ||CAR (w/o g, pgd)|2680|2540|2174|2303|1772|
> ||CAR (w/o g, sgld)|1274|1345|897|1381|888|
> |Halfcheetah|SPO|5048|4463|3281|918|1049|
> ||CAR (w/o g, pgd)|2058|2068|2067|2070|2002|
> ||CAR (w/o g, sgld)|1836|1834|1804|1836|1755|
> |Ant|SPO|5381|5329|4696|1768|1097|
> ||CAR (w/o g, pgd)|1817|1801|1797|1768|1634|
> ||CAR (w/o g, sgld)|1798|1965|1834|1762|1665|
>
>
> - Second, the work in [2] addresses a different problem setting and uses a different methodology.
>
>     - Problem setting: [2] focuses on perturbations to the environment, i.e., the transition dynamics and reward functions. These perturbations are decoupled from the agent's policy. In contrast, our work focuses on state perturbations, which directly alter the information the agent uses for decision-making. This is a fundamentally different type of threat model.
>
>     - Methodology: [2] employs a multi-agent paradigm, where an agent and an adversary are trained alternately. Our approach follows the adversarial training paradigm from [3], where the adversary is explicitly computed by solving an inner optimization problem for a given agent policy, rather than being a co-trained agent.
>
> These key differences in problem formulation and algorithmic approach explain why the results are not directly comparable and why our observations do not conflict with those in [1] and [2].
>
> [1] Li et al., "Towards Optimal Adversarial Robust Reinforcement Learning with Infinity Measurement Error." arXiv:2502.16734
>
> [2] Pinto et al., "Robust adversarial reinforcement learning." ICML 2017
>
> [3] Madry et al., "Towards deep learning models resistant to adversarial attacks." ICLR 2018
>
> ---
> **Q3: Detailed interpretation of Figure 3**
>
> **A:** Please refer to our response to Weakness 1.4.
>
> ---
> **Q4: Use a lower bound as the surrogate in Theorem 4.1**
>
> **A:** We would like to clarify the perspective of the optimization in Theorem 4.1. This theorem is derived specifically for the adversary, not the agent.
>
> While the agent's ultimate goal is indeed to minimize the performance gap (to achieve robustness), the adversary's objective is the opposite: to maximize the difference between the nominal and under-attack returns. To effectively maximize a complex objective, it is standard practice to maximize its lower bound. Therefore, deriving and maximizing a lower bound is the mathematically correct approach for the adversary's optimization problem in this context.
>
> ---
> **Q5: Value source for Figure 4**
>
> **A:** The values presented in Figure 4 represent true objective values of the SPO, ARPO, and BARPO. They are not purely illustrative.
>
> The data is specifically drawn from the ISA-MDP example (detailed in Appendix D). The figure visualizes this loss landscape by plotting the objective values along a two-dimensional slice within the high-dimensional parameter space of the two-layer neural network policy. This provides a tangible geometric view of the differences between the three optimization objectives.

---

> ### Author Response · Authors · 2025-11-21
>
> ---
> **Q6: Connection between different parts of Section 4.2**
>
> **A:** We address this below to explain the structure and logic of our proposed method.
>
> - KL surrogate and better policy learning: The combination of the surrogate and the bilevel framework leads to improved policy learning through two complementary effects:
>
>     - The overall bilevel structure primarily works to correct the distorted policy landscape introduced by ARPO. This stabilization directly improves the agent's convergence and natural performance.
>
>     - The KL divergence serves as a valid surrogate to minimize the adversarial value. This ensures that while we improve the landscape, we maintain a high degree of robustness without excessive loss.
>
>     The synergy between a smoother landscape and preserved robustness results in overall better policy learning.
>
> - The appropriateness of the surrogate stems from the adversary’s objective in the inner optimization.
>
>     1. The adversary's objective is to minimize the adversarial value, which is equivalent to maximizing the gap between standard and adversarial values.
>
>     2. When maximizing a complex objective, it is standard practice to maximize its lower bound from an optimization perspective, as this provides a tractable guarantee.
>
>     3. The surrogate serves as a valid lower bound for this maximization problem, making it an appropriate and reliable proxy for the adversary's true objective.
>
> - Theorem 4.1 provides the theoretical justification for the inner optimization of the bilevel problem:
>
>     1. The theorem establishes a lower bound on the adversary’s objective function. Crucially, this lower bound is shown to be a monotonically increasing function of the KL divergence between the unperturbed and perturbed policies.
>
>     2. Therefore, maximizing the KL divergence is mathematically equivalent to maximizing the theoretical lower bound established in Theorem 4.1. This equivalence allows us to translate the theoretical guarantee into the practical, tractable inner maximization problem of the bilevel formulation.
>
> ---
> **Q7: BARPO's effect on landscape**
>
> **A:** We delve into a deeper understanding of how BARPO reshapes the optimization landscape from three complementary perspectives, which are provided in Appendix H.4 due to space limitations.
>
> -  Adversarial value improvement through landscape lifting: Let $\nu^\diamond$ denote the solution to BARPO's inner problem and $\nu^* $ denote the strongest adversary defined in ARPO. By definition, $V^{\pi\circ\nu^\diamond (\pi)} \ge V^{\pi\circ\nu^* (\pi)}$, indicating that BARPO tends to elevate low-value regions in ARPO's landscape. Furthermore, Theorem 4.1 shows that maximizing the KL surrogate aligns with minimizing adversarial value: when the KL surrogate is high, the adversarial value is correspondingly low. This provides a theoretical link between BARPO’s surrogate objective and ARPO’s robustness.
>
> - Smoothing via regularized maximin reformulation: Following prior work [4,5], when $\alpha$ is sufficiently large, BARPO can be reformulated as an equivalent maximin problem:
>     $$\max_{\theta,\nu^\diamond}\min_{\vartheta} L^\alpha (\theta,\nu^\diamond,\vartheta) = E_{s\sim\mu_0} [ V^{\pi_\theta\circ\nu^\diamond}(s) + \alpha(\operatorname{KL} \left(\pi_\theta (s) | \pi_\theta\circ\nu^\diamond (s) \right) - \operatorname{KL} \left(\pi_\theta (s) | \pi_\theta\circ\nu_\vartheta (s) \right) ) ].$$
>     Compared to ARPO’s objective, this formulation introduces a regularization term that has a smoothing effect on the outer landscape. Structurally, this resembles the use of Bregman divergences in defining the Moreau envelope [6], which is known to smooth the nonsmooth optimization problems.
>
> - Robustness under date-driven value approximation: In practice, ARPO optimizes a lower bound $L(\pi, \nu) \le V^{\pi\circ\nu}$ due to practical data-driven estimation for $V^{\pi\circ\nu}$, i.e., the practical objective becomes $\max_\pi \min_\nu L(\pi,\nu)$, which is no longer equivalent to the ideal adversarial objective. In contrast, BARPO's inner optimization minimizes an upper bound on the adversarial value, while the outer loop maximizes a lower bound. This formulation is more faithful to the original robust motivation and offers better resilience under value function approximation, thereby improving performance in practice.
>
> [4] Kwon et al., "A fully first-order method for stochastic bilevel optimization." ICML 2023
>
> [5] Lu and Mei, "First-order penalty methods for bilevel optimization." SIAM Journal on Optimization, 2024
>
> [6] Bauschke et al, "Regularizing with Bregman--Moreau envelopes." SIAM Journal on Optimization 2018

---

### Official Review · Reviewer_bepT · 2025-11-11

**Soundness:** 3
**Presentation:** 3
**Contribution:** 3
**Rating:** 6
**Confidence:** 2

**Summary:**

This paper introduced the Bilevel ARPO (BARPO) that balances the optimality-robustness trade-off, by adjusting the adversary strength to promote traversable optimization paths which smoothed the optimization landscape.

**Strengths:**

* The paper was well written and easy to follow. Assumptions and motivations were clearly justified.
* The insights on how the 'valleys' were formed and how to 'bridge' them were novel and interesting.
* Theoretical analyses were sound.
* In experiments BARPO was evaluated against extensive attack/robustness conditions.

**Weaknesses:**

* BARPO employs a KL-based surrogate for the inner minimization to approximate the strongest adversary.
  * While the paper shows that minimizing this surrogate aligns with minimizing the true adversarial value, the reviewer wonders whether the authors have additional theoretical or empirical insights regarding the convergence rate under this surrogate formulation. In particular, are there quantifiable bounds or guarantees on the degree of robustness potentially lost due to the surrogate approximation?
* From table 8 it showed that the performance could be fairly sensitive to $\kappa$.
  * Any findings around how it can be tuned would be appreciated.

**Questions:**

* The experiments mainly focused on continuous control tasks. How does BARPO generalizes to tasks with large discrete action space? Could the authors comment on potential challenges (e.g. combinatorial action spaces, function approximation issues)?
* Is there a way to tune BARPO (e.g., via a schedule on adversary strength) to balance return vs robustness, for different application needs?

---

> ### Author Response · Authors · 2025-11-21
>
> ---
> **W1: Convergence rate and robustness gap under the surrogate formulation**
>
> **A:** We thank the reviewer for their insightful questions regarding our KL-based surrogate. We address the convergence rate and the quantification of the robustness gap below.
>
> - On the Convergence Rate
>
>     - Theoretical insight: Our formulation's convergence can be analyzed by extending existing bilevel optimization frameworks. Specifically, adapting the analysis from [1], we can establish a theoretical convergence rate of $O(T^{-1/2})$ for our method.
>
>     - Empirical insight: We also refer the reviewer to Figure 6 in our paper. These results empirically demonstrate that our surrogate formulation achieves a convergence rate comparable to that of the original maximin formulation, aligning with our theoretical understanding.
>
> - On Quantifying the Robustness Gap
>
>     We analyzed the deviation between our surrogate objective and the true adversarial objective to quantify any potential robustness loss. Following [2,3], for sufficiently large $\alpha$, the surrogate formulation can be reformulated:
>     $$\max_{\theta,\nu^\diamond}\min_{\vartheta} L^\alpha (\theta,\nu^\diamond,\vartheta) = E_{s\sim\mu_0} [ V^{\pi_\theta\circ\nu^\diamond}(s) + \alpha(\operatorname{KL} \left(\pi_\theta (s) | \pi_\theta\circ\nu^\diamond (s) \right) - \operatorname{KL} \left(\pi_\theta (s) | \pi_\theta\circ\nu_\vartheta (s) \right) ) ].$$
>     Then, we quantify the deviation from the original objective using the bound:
>     $$|L^\alpha (s;\theta,\nu^\diamond,\vartheta) - V^{\pi_\theta\circ\nu_\vartheta}| \le |V^{\pi_\theta\circ\nu^\diamond}-V^{\pi_\theta\circ\nu_\vartheta}| + \alpha| \operatorname{KL} \left(\pi_\theta  \| \pi_\theta\circ\nu^\diamond  \right) - \operatorname{KL} \left(\pi_\theta  \| \pi_\theta\circ\nu_\vartheta  \right)|.$$
>     This gap consists of two main terms.
>
>     - The first term, the difference in value functions, is bounded by the square root of the KL divergence between the respective policies (based on Theorem 5 in [4]), i.e., $C \sqrt{\operatorname{KL}(\pi_\theta\circ\nu^\diamond \| \pi_\theta\circ\nu_\vartheta)}$.
>
>     - The second term, related to the difference in KL penalties, can also be bounded. Using (reverse) Pinsker's inequalities, this term is similarly controlled by the square root of the KL divergence, i.e., $O\left(\sqrt{\operatorname{KL}(\pi_\theta\circ\nu^\diamond \| \pi_\theta\circ\nu_\vartheta)}\right)$, especially when the policies are not too far apart.
>
>     This analysis shows that the total deviation is controlled by the square root of the KL divergence between $\pi_\theta\circ\nu^\diamond$ and $\pi_\theta\circ\nu_\vartheta$. This finding has a clear practical implication. If we utilize a state-Lipschitz policy class for function approximation (a common setup), this theoretical gap can be directly bounded by the perturbation budget, $O(\epsilon_{pert})$.
>
>
> [1] Ji et al., "Bilevel optimization: Convergence analysis and enhanced design." ICML 2021
>
> [2] Kwon et al., "A fully first-order method for stochastic bilevel optimization." ICML 2023
>
> [3] Lu and Mei, "First-order penalty methods for bilevel optimization." SIAM Journal on Optimization, 2024
>
> [4] Zhang et al., "Robust deep reinforcement learning against adversarial perturbations on state observations." NeurIPS 2020 spotlight
>
> ---
> **W2: About the tuning of $\kappa$**
>
> **A:** The hyperparameter $\kappa$ is indeed important for the trade-off between adversarial robustness and natural returns. Our findings illustrate the following relationships:
>
> - A larger $\kappa$ places a stronger emphasis on robustness, leading to a more resilient policy. However, setting $\kappa$ too high (e.g., $> 1.0$) can create an overly conservative policy, which may significantly reduce the natural return.
>
> - A smaller $\kappa$ prioritizes natural return. However, if $\kappa$ is set too low (e.g., $< 0.1$), the policy may not achieve a sufficient level of robustness.
>
> Based on our empirical results, we offer a practical guideline for tuning: selecting a $\kappa$ in the range $[0.1, 1.0]$ generally provides a good balance, ensuring a meaningful level of robustness without sacrificing excessive standard performance.

---

> ### Author Response · Authors · 2025-11-21
>
> ---
> **Q1: Extend BARPO to large discrete action spaces**
>
> **A:** We discuss this from three aspects.
>
> - Conceptual generalization: We confirm that the core framework of BARPO is not limited to continuous control and can be directly generalized to discrete action spaces. The primary adaptation would involve choosing the surrogate for the inner minimization. While our paper focused on a KL-based surrogate, a margin-based surrogate is also a proper alternative for discrete settings, i.e., $\nu^* (\pi) = \arg\max_\nu \max_{a\neq a^* (s)} (\pi \circ \nu)(a | s) - (\pi \circ \nu)(a^* (s) | s)$, where $a^* (s) = \arg\max_a \pi(a|s)$.
>
> - Potential challenges: The central challenge in large or combinatorial action spaces indeed lies in function approximation. Specifically, the practical challenge for BARPO is how to efficiently approximate the chosen surrogate (KL or margin-based) within the policy network. A practical implementation of BARPO in such settings would integrate our framework with established architectures designed for these complex spaces, such as action embeddings [5], autoregressive policies [6,7,8], embedded mixed integer programming [9], and value-based tree search [10].
>
> - New experimental validation: To validate this, we have already conducted new experiments on the Procgen benchmark [11]. Procgen is highly relevant as it features environments with combinatorial action spaces and is a standard for evaluating generalization. We trained agents with an attack budget of $1/255$, and tested robustness against four different attacks (combinations of PGD/FGSM and cross-entropy/margin losses) at increasing budgets ($1/255$, $3/255$, $5/255$). The results, summarized in the table below, show that BARPO achieves superior natural performance and robust returns in this challenging discrete setting.
>
> | Env     | Method            | Standard Normal      | $\epsilon = 1/255$       | $\epsilon = 3/255$       | $\epsilon = 5/255$       |
> | :------ | :---------------- | :-------------------- | :------------------ | :------------------ | :------------------ |
> | Fruitbot| SPO               | 25.0430 ± 0.3462     | 21.8010 ± 0.3696    | 14.1390 ± 0.3836    | 9.4050 ± 0.3537     |
> |         | RADIAL            | 26.1450 ± 0.2870     | 26.0740 ± 0.2901    | 25.9260 ± 0.2944    | 25.8860 ± 0.2988    |
> |         | ARPO              | 23.3500 ± 0.3672     | 23.5680 ± 0.3624    | 23.3420 ± 0.3670    | 23.6090 ± 0.3581    |
> |         | BARPO       | **26.9860 ± 0.3184** | **26.8040 ± 0.3213**| **26.4260 ± 0.3283**| **25.9460 ± 0.3347**|
> | Coinrun | SPO               | 5.3100 ± 0.1578      | 4.1000 ± 0.1555     | 3.2200 ± 0.1478     | 3.0600 ± 0.1457     |
> |         | RADIAL            | 4.8900 ± 0.1581      | 4.8600 ± 0.1581     | 4.8300 ± 0.1580     | 4.8700 ± 0.1581     |
> |         | ARPO              | 7.8900 ± 0.1290      | 7.6700 ± 0.1337     | 7.7700 ± 0.1316     | 7.5500 ± 0.1360     |
> |         | BARPO         | **7.9800 ± 0.1270**  | **8.0500 ± 0.1253** | **7.9300 ± 0.1281** | **7.6900 ± 0.1333** |
> | Jumper  | SPO               | 4.0900 ± 0.1555      | 3.4000 ± 0.1498     | 2.5400 ± 0.1377     | 2.3700 ± 0.1345     |
> |         | RADIAL            | 3.8600 ± 0.1539      | 3.8800 ± 0.1541     | 3.8100 ± 0.1536     | 3.8400 ± 0.1538     |
> |         | ARPO              | 4.4200 ± 0.1570      | 4.4200 ± 0.1570     | 4.3400 ± 0.1567     | 4.0700 ± 0.1554     |
> |         | BARPO         | **4.5100 ± 0.1574**     | **4.5700 ± 0.1575** | **4.5400 ± 0.1574** | **4.2600 ± 0.1564** |
>
>
> These findings confirm that BARPO's principles are effective beyond continuous control and can be successfully applied to tasks with discrete action spaces.
>
> [5] Dulac-Arnold et al., "Deep reinforcement learning in large discrete action spaces." arXiv:1512.07679
>
> [6] He et al., "Deep reinforcement learning with a combinatorial action space for predicting popular reddit threads." EMNLP 2016
>
> [7] Berto et al., "Parco: Learning parallel autoregressive policies for efficient multi-agent combinatorial optimization." NeurIPS 2025
>
> [8] Ma et al., "Reinforcement Learning with Discrete Diffusion Policies for Combinatorial Action Spaces." arXiv:2509.22963
>
> [9] Xu et al., "Reinforcement learning with combinatorial actions for coupled restless bandits." ICLR 2025
>
> [10] Landers et al., "BraVE: Offline Reinforcement Learning for Discrete Combinatorial Action Spaces." NeurIPS 2025
>
> [11] Cobbe et al., "Leveraging procedural generation to benchmark reinforcement learning." ICML 2020

---

> ### Author Response · Authors · 2025-11-21
>
> ---
> **Q2: Further tune BARPO for different application needs.**
>
> **A:** We thank the reviewer for this insightful question. Yes, BARPO is highly tunable, and we can adjust the balance between natural return and robustness to fit specific applications. Here are four primary mechanisms to achieve this:
>
> - Curriculum-based scheduling: This directly follows the reviewer's suggestion. Similar to the idea of curriculum learning, we can implement schedulers for key hyperparameters. For example, one can schedule the attack budget ($\epsilon$) or the regularization weight ($\kappa$), starting with lower values and gradually increasing them (e.g., linearly or exponentially). This allows the agent to first learn a high-performing policy before progressively focusing on adversarial robustness.
>
> - Adaptive regularization: Rather than a fixed schedule, the regularization coefficient $\kappa$ can be dynamically adapted during training. For instance, we could design a rule to automatically adjust $\kappa$ based on the agent's current performance, ensuring a consistent, pre-defined balance between standard return and robustness throughout the learning process.
>
> - Modifying the surrogate objective: The trade-off is also influenced by the choice of the surrogate used for the inner optimization. Our paper focuses on a KL-based surrogate. However, for different application domains, one could select a different objective. For example, for the discrete action spaces discussed in Q1, one could select a margin-based surrogate. This choice provides a high-level method for tailoring the robustness criteria.
>
> - Hybrid training phases: The training procedure itself can be modified. For example, one could train using our bilevel framework in a first phase to secure a good balance of natural and robust performance, and then switch to a pure maximin framework in a second phase to fine-tune and maximize robustness, if that becomes the priority.

---

> > ### Comment · Reviewer_bepT · 2025-11-28
> > **Q2 of wh98**
> >
> > The reviewer would like to thank the authors for the additional insights -- they addressed most of my outstanding concerns. After looking at the reviews from other reviewers, the reviewer thought it would be helpful to further clarify on wh98's Q2. The author mentioned in the response that this work focuses on state perturbation where the perturbed states are directly used by the policy so it would be more challenging compared to Pinto et al. which perturbs the transition dynamics and rewards. However, the reviewer is confused that if environment transition dynamics were perturbed, isn't it somewhat equivalent to perturbing the states directly (i.e., Pinto et al. covered the cases where the perturbed dynamics that can lead to state incorrectness which would have been taken by the policy too)? Since in the MDP step one can assume that there's a bijection between environmental dynamics and the states.

---

> > > ### Author Response · Authors · 2025-11-28
> > >
> > > We thank the reviewer for the positive feedback and for raising this interesting point regarding the relationship between state and dynamics perturbations. The decision-making under perturbations to dynamics is often formulated as RMDPs [1], and decision-making with perturbations to state observations is formulated as SA-MDPs [2]. They are inherently distinct mathematical problems, and one is not simply a subset of the other. We outline three key differences below:
> > >
> > > 1. The Mechanism of Influence: The two settings affect the decision-making process differently.
> > >
> > >     - State Perturbation: The adversary attacks the input to the policy. The agent must act based on a corrupted observation, meaning the policy's sensitivity (smoothness) with respect to the state is directly tied to robustness.
> > >
> > >     - Dynamics Perturbation: The adversary attacks the environment, which is relatively decoupled from the policy. The agent sees the correct state and acts, but the environment transitions to a worst-case next state. Because the decision is made before the perturbation takes effect, the "smoothness" required for state robustness does not translate to robustness against dynamics shifts. Therefore, a bijection between dynamics and states does not imply equivalence in the decision-making problem.
> > >
> > > 2. Existence of Optimal Policies: The theoretical properties of the two problems differ significantly.
> > >
> > >     - In the dynamics perturbation setting (RMDP), it has been proven that an optimal robust policy is guaranteed to exist under assumptions for s-rectangular ambiguity sets [1].
> > >
> > >     - In contrast, in the state perturbation setting (SA-MDP) with the rectangular assumption, an optimal robust policy may not exist [2]. This fundamental theoretical difference confirms they belong to different problem classes.
> > >
> > > 3. Geometry of the Value Function: The shape of the value geometry is distinct.
> > >
> > >     - As shown in [3], the value geometry in dynamics-perturbed settings is the intersections of hyperplanes, resulting in boundaries formed by straight line segments.
> > >
> > >     - In contrast, as illustrated in the left panel of Figure 3 in our paper, the value geometry for state perturbations is more complex, featuring boundaries consisting of both line segments and non-linear curves.
> > >
> > > These points collectively demonstrate that state perturbation presents a unique set of challenges distinct from the dynamics perturbation setting found.
> > >
> > > [1] Wiesemann et al, "Robust Markov decision processes." Mathematics of Operations Research, 2013
> > >
> > > [2] Zhang et al., "Robust deep reinforcement learning against adversarial perturbations on state observations." NeurIPS 2020 spotlight
> > >
> > > [3] Wang et al., "The geometry of robust value functions." ICML 2022

---

> > > > ### Comment · Reviewer_bepT · 2025-11-28
> > > >
> > > > Thanks for the clarifications. It would be helpful to maybe have a short discussion of this in the manuscript to clarify, as well as the other points in my review and from other reviewers. The overall responses sound good to me and I will increase to 8 when I can (score updates are now turned off by the conference for some reasons).

---

> > > > > ### Author Response · Authors · 2025-11-28
> > > > >
> > > > > Thanks for your feedback and thoughtful consideration! We will add a discussion in the related work to clarify. We sincerely appreciate your valuable comments, prompt response, and recognition of our work.

---

### Author Response · Authors · 2025-11-21
**General Response**

We sincerely appreciate the reviewers for their thoughtful reviews and constructive feedback! We are highly encouraged by the positive consensus regarding our work:

- Novelty and intuition: The paper tackles a central and underexplored issue with the trade-off between robustness and optimality (RuhM). The proposed geometric explanation is novel, informative, and intuitive (bepT, wh98, RuhM), and the method itself is well-motivated and intuitive (wh98, LLxi).

- Theoretical and empirical solidity: The theoretical analyses are sound (bepT, LLxi, RuhM) and new (wh98). The experimental evaluation to be comprehensive, demonstrating strong performance across diverse conditions (bepT, wh98, LLxi, RuhM).

- Clarity and presentation: The paper was well written and easy to follow (bepT, LLxi). Assumptions and motivations were clearly justified (bepT), and the theoretical development is clear (LLxi).

We hope the following responses could address reviewers' concerns.

---

### Meta-Review · Area_Chair_DDYK · 2026-01-05

**Summary:**

Overall, the reviewers raised minor concerns around the notation and presentation of the paper, though these concerns were easily resolved during the rebuttal phase. No significant concerns on the theory or experimental evaluation were raised. Most questions referred to possible extensions of the framework.

The reviewers seem to not have any major issues with the paper, and those that were able to participate in the discussion maintained or raised their score.

Overall, there is consensus that this is a good contribution. The authors can polish the presentation and incorporate the discussion before the camera ready version.

**Reviewer Concerns:**

Reviewer bepT had some minor questions about the theoretical analysis and experimental validation that the authors resolved positively. The reviewer mentioned they were intending to upgrade their score from 6 to 8. The authors promised to incorporate the discussion in the final version of the paper.

Reviewer wh98 had concerns with the presentation, concluding with "While I see the technical contribution of this work, its lack in presentation coherence and smoothness severely undermines the quality of this work". The other reviewers seem to not share the sentiment. The authors responded point by point to the reviewer. While there was no response from the reviewer, I believe the authors did a good job at addressing the review.

Reviewer LLxi had very few concerns (no weaknesses listed).

Reviewer RuhM had a couple of question about the overhead of the method, and relationship with other frameworks. The brief discussion with the authors ended with the reviewer saying "I do not have further questions and would like to keep my positive rating."

**Reviewer Scores:**

Reviewer bepT explicitly mentioned raising their score to 8.

Reviewer wh98 did not participate in the discussion.

Reviewer LLxi had very few concerns, with no weaknesses for the paper listed in their review.

Reviewer RuhM explicitly mentioned their desire of maintaining their positive score.

---

### Decision · Program_Chairs · 2026-01-26

Accept (Poster)